# CoCoA: A Minimum Bayes Risk Framework Bridging Confidence and Consistency for Uncertainty Quantification in LLMs

**Roman Vashurin**[*]    **Maiya Goloburda**[*]    **Albina Ilina**    **Aleksandr Rubashevskii**
**Preslav Nakov**        **Artem Shelmanov**
**Maxim Panov**

Mohamed bin Zayed University of Artificial Intelligence (MBZUAI)
{Roman.Vashurin, Maiya.Goloburda, Maxim.Panov}@mbzuai.ac.ae

## Abstract

Uncertainty quantification for Large Language Models (LLMs) encompasses a diverse range of approaches, with two major families being particularly prominent: (i) information-based, which estimate model confidence from token-level probabilities, and (ii) consistency-based, which assess the semantic agreement among multiple outputs generated using repeated sampling. While several recent methods have sought to combine these two paradigms to improve uncertainty quantification performance, they often fail to consistently outperform simpler baselines. In this work, we revisit the foundations of uncertainty estimation through the lens of Minimum Bayes Risk decoding, establishing a direct link between uncertainty and the optimal decision-making process of LLMs. Building on these findings, we propose CoCoA, a unified framework that integrates model confidence with output consistency, yielding a family of efficient and robust uncertainty quantification methods. We evaluate CoCoA across diverse tasks, including question answering, abstractive text summarization, and machine translation, and demonstrate sizable improvements over state-of-the-art uncertainty quantification approaches.

## 1 Introduction

Large Language Models (LLMs) have revolutionized natural language processing (NLP), enabling advances in information retrieval [Zhu et al., 2025], question answering [Kwiatkowski et al., 2019], machine translation [Kocmi and Federmann, 2023], and a broad range of other NLP applications. As these models become an integral part of our everyday life, ensuring the reliability of their outputs is crucial, especially in high-stakes scenarios where errors can have serious consequences. One way to address this challenge is through uncertainty quantification (UQ), which focuses on estimating the confidence of model predictions.

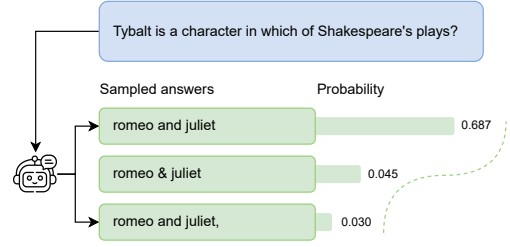

Figure 1: Example of inconsistent probabilities assigned to semantically identical answers by an LLM, demonstrating the limitation of relying solely on sequence-level information.

---

[*] These authors contributed equally.

39th Conference on Neural Information Processing Systems (NeurIPS 2025).

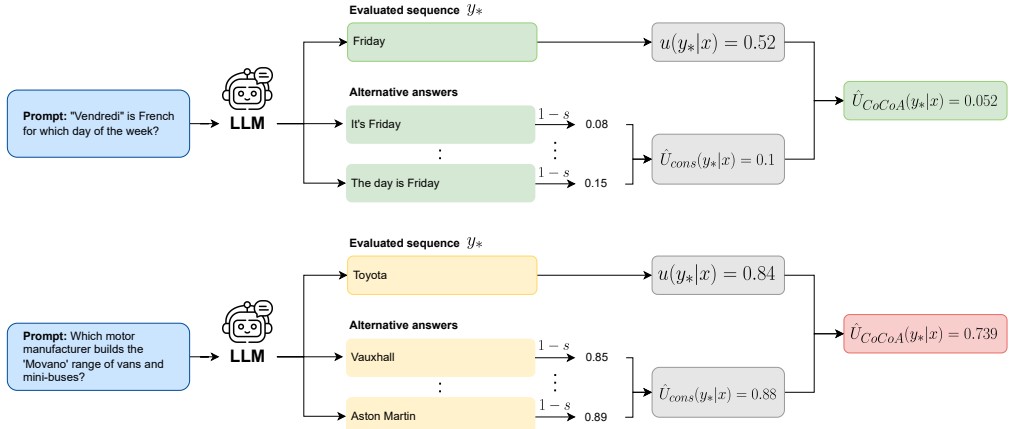

Figure 2: Illustration of our method: the LLM generates a response, evaluates the similarity to alternatives, computes the confidence, and finally combines the confidence with the similarity measure. High similarity to alternatives reduces the uncertainty, while low similarity keeps it high.

UQ for LLMs is a rapidly advancing research area, with new UQ methods emerging each year. Most novel techniques are based on two fundamental approaches: (a) information-theoretic analysis or (b) assessment of output consistency.

Information-theoretic methods quantify the confidence of a model by analyzing the probability distributions it induces for predictions [Malinin and Gales, 2021, Fomicheva et al., 2020]. A key limitation of these methods is that they cannot account for semantic variability across multiple possible responses for the same input. Specifically, the model may generate answers with the same meaning but with very different assigned probabilities; see Figure 1. LLMs are trained to predict the next token in a sequence based on patterns observed in vast amounts of data, resulting in varying probabilities for semantically equivalent output sequences.

In contrast, consistency-based methods directly analyze the semantic relationship between the sampled outputs [Fomicheva et al., 2020, Lin et al., 2024], capturing uncertainty as objective variability of meaning among the sampled outputs.

Information-theoretic and consistency-based methods have complementary strengths. For this reason, recent state-of-the-art methods aimed to unify these approaches [Kuhn et al., 2023, Duan et al., 2024]. We follow this direction and propose a way of quantifying risk as a combination of these basic measures. This results in a family of efficient and robust UQ techniques. Our approach, illustrated in Figure 2, combines the strengths of both information-based and consistency-based methods, providing a more comprehensive and accurate assessment of uncertainty.

Recently, it has been argued [Wang and Holmes, 2025, Daheim et al., 2025] that UQ in NLP tasks can be viewed through the lens of the Minimum Bayes Risk (MBR) framework. Following this, we formulate our methods as particular forms of risk functions under the MBR approach.

Our main contributions can be summarized as follows:

- We propose a new way of quantifying risk that combines information-theoretic and consistency-based measures, and derive a family of *Confidence and Consistency-based Approaches* (CoCoA) to uncertainty quantification in LLMs[2].

- We show that the consistency component can be approximated by a learned function trained on unlabeled held-out set with a negligible loss of performance, eliminating the need for costly repeated sampling from the LLM. We call this variation of our method CoCoA Light.

- We evaluate our approaches across a variety of NLP tasks, including question answering, summarization, and machine translation. Our experiments demonstrate sizable improvements in the reliability and the robustness of UQ compared to state-of-the-art methods.

---

[2]Our code is available publicly at https://github.com/stat-ml/llm_uncertainty_cocoa

## 2 Background

### 2.1 Language Model Decoding

LLMs define a probabilistic output distribution $p(\mathbf{y} \mid \mathbf{x})$ for a given input sequence $\mathbf{x}$. The standard ways to obtain a particular output $\mathbf{y}_*$ given $p(\mathbf{y} \mid \mathbf{x})$ include various variants of sampling $\mathbf{y}_* \sim p(\mathbf{y} \mid \mathbf{x})$ and greedy decoding, where

$$\mathbf{y}_* = \arg \max_{\mathbf{y}} p(\mathbf{y} \mid \mathbf{x}). \tag{1}$$

An alternative approach is to use *Minimum Bayes Risk (MBR)* decoding [Kumar and Byrne, 2004a]:

$$\mathbf{y}_* = \arg \min_{\mathbf{y} \in \mathcal{Y}} R(\mathbf{y} \mid \mathbf{x}), \tag{2}$$

where $\mathcal{Y}$ is the set of candidate sequences and $R(\mathbf{y} \mid \mathbf{x})$ is a risk function:

$$R(\mathbf{y} \mid \mathbf{x}) = \mathbb{E}_{\mathbf{y}' \sim p(\mathbf{y}|\mathbf{x})} \ r(\mathbf{y}, \mathbf{y}') \tag{3}$$

with $r(\mathbf{y}, \mathbf{y}')$ being a pairwise loss function.

The standard choice in the MBR literature is to take $r(\mathbf{y}, \mathbf{y}') \propto -s(\mathbf{y}, \mathbf{y}')$, where the so-called *utility function* $s(\mathbf{y}, \mathbf{y}')$ represents some notion of similarity between generated sequences $\mathbf{y}$ and $\mathbf{y}'$. Below we discuss how various decoding strategies and corresponding risk functions lead to well-grounded definitions of uncertainty.

### 2.2 From Risk to Uncertainty

Generally, an *uncertainty function* $U$ is a mapping that quantifies the level of uncertainty associated with the output of a model $\mathbf{y}$, conditioned on the input sequence $\mathbf{x}$, which we denote as $U(\mathbf{y} \mid \mathbf{x})$.

Bayesian uncertainty measures are well known to be strongly connected with minimum risks of various kinds [Xu and Raginsky, 2022, Kotelevskii et al., 2025]. Recently, this connection was revisited in the context of natural language generation [Wang and Holmes, 2025, Daheim et al., 2025], leading to the development of new MBR-based UQ methods. More precisely, one can consider the smallest achievable risk within $\mathcal{Y}$ or, equivalently, to maximum expected utility:

$$U_{\text{MBR}}(\mathbf{y}_* \mid \mathbf{x}) = \min_{\mathbf{y} \in \mathcal{Y}} R(\mathbf{y} \mid \mathbf{x}) = R(\mathbf{y}_* \mid \mathbf{x}), \tag{4}$$

where $\mathbf{y}_*$ is given by (2). Interestingly, MBR-based uncertainty (4) is applicable even to $\mathcal{Y}$ consisting of a single sequence $\mathbf{y}_*$.

Below, we discuss how various existing uncertainty measures can be seen as particular approximations of minimum Bayes risks for various choices of utility functions. We also derive new uncertainty measures based on the MBR framework.

**Single Sequence Information-Based Methods.** Information-based methods estimate the uncertainty of the generated sequence by aggregating the uncertainty scores of individual tokens. Within the framework of MBR decoding, consider $r(\mathbf{y}, \mathbf{y}') = \mathbf{1}\{\mathbf{y}' \neq \mathbf{y}\}$. In this case, the Bayes risk corresponds to the expected zero-one loss of decoding:

$$R_{0/1}(\mathbf{y} \mid \mathbf{x}) = \mathbb{E}_{\mathbf{y}' \sim p(\mathbf{y}|\mathbf{x})} \mathbf{1}\{\mathbf{y}' \neq \mathbf{y}\} = 1 - p(\mathbf{y} \mid \mathbf{x}). \tag{5}$$

This leads to one of the simplest information-based uncertainty measures, *Sequence Probability (SP)*:

$$U_{\text{SP}}(\mathbf{y}_* \mid \mathbf{x}) = 1 - p(\mathbf{y}_* \mid \mathbf{x}). \tag{6}$$

Although widely used as an uncertainty measure across various applications, including natural language generation, its connection to statistical risk has only recently been explored in the UQ literature [Kotelevskii et al., 2022, 2025, Aichberger et al., 2024].

Several other measures fall into this category, including *Perplexity* and *Mean Token Entropy* [Fomicheva et al., 2020]; see Appendix D.1 for details. While using only a single sample makes them computationally efficient, these techniques face three major challenges:

1. LLMs only give the probability of a specific answer, even though the same meaning can often be conveyed in multiple ways. To obtain a proper probability for the meaning of an answer, we need to marginalize over its various possible rephrasings. However, this is not feasible if we generate only a single sample.

2. The reliability of such methods depends heavily on the underlying LLM being well-calibrated, which is a quality that is hard to define and harder to ensure.

3. These methods are point estimates that do not provide information about the shape of the output distribution.

**Semantic Consistency-Based Methods.** The aforementioned issues lead to the development of consistency-based methods based on repetitive sampling from the LLM. Consider that we have sampled a set of outputs $\left\{\mathbf{y}^{(i)}\right\}_{i=1}^{M}$, where $\mathbf{y}^{(i)} \sim p(\mathbf{y} \mid \mathbf{x})$. Consistency-based UQ methods rely on the diversity of the answers $\mathbf{y}^{(i)}$ sampled from the LLM. The idea is that if the model outputs similar answers for the same prompt over and over again, it is confident in its predictions; otherwise, it is uncertain. These techniques do not require a probability distribution estimated by an LLM and can be applied in a black-box setting, where only the generated tokens are available. This case is quite common when LLMs are deployed as a service and are accessible through a limited API.

Formally, consistency-based methods start from defining some similarity function $s\left(\mathbf{y}, \mathbf{y}'\right) \in [0, 1]$ between arbitrary LLM generations $\mathbf{y}$ and $\mathbf{y}'$. The value $s\left(\mathbf{y}, \mathbf{y}'\right) = 1$ indicates the complete equivalence between $\mathbf{y}$ and $\mathbf{y}'$, and $s\left(\mathbf{y}, \mathbf{y}'\right) = 0$ indicates that there is no similarity. The similarity could be computed in various ways. For instance, the *Lexical Similarity* method [Fomicheva et al., 2020] relies on surface-form similarity, measuring the degree of word-level or phrase-level overlap between the generated texts. More advanced techniques propose various methods for taking into account the semantic similarity between the generated answers by means of hard or soft clustering [Lin et al., 2024].

Finally, given $M$ samples from the model, standard consistency-based methods compute a similarity matrix $S$, where $s_{ij} = s\left(\mathbf{y}^{(i)}, \mathbf{y}^{(j)}\right)$. Then, various statistics of $S$ are computed in order to estimate the uncertainty [Lin et al., 2024]; see Appendix D.2 for more detail. Alternatively, one can focus on a particular output $\mathbf{y}_*$ and compute uncertainty measures based on its similarity $s_{*i} = s\left(\mathbf{y}_*, \mathbf{y}^{(i)}\right)$ with other samples $\mathbf{y}^{(i)}$; see [Lin et al., 2024] for more detail.

Interestingly, it was recently shown [Wang and Holmes, 2025, Daheim et al., 2025] that the latter approach has direct relation to MBR decoding. Building on the MBR framework, one can define the utility function as $r(\mathbf{y}, \mathbf{y}') = 1 - s(\mathbf{y}, \mathbf{y}')$ and derive the following consistency-based uncertainty score:

$$U_{\text{cons}}(\mathbf{y}_* \mid \mathbf{x}) = \mathbb{E}_{\mathbf{y}' \sim p(\mathbf{y}|\mathbf{x})}\left[1 - s(\mathbf{y}_*, \mathbf{y}')\right]. \tag{7}$$

Then the Monte Carlo approximation of the minimum Bayes risk is given by

$$\widehat{U}_{\text{cons}}(\mathbf{y}_* \mid \mathbf{x}) = \frac{1}{M}\sum_{i=1}^{M}(1 - s_{*i}), \tag{8}$$

where $s_{*i} = s\left(\mathbf{y}_*, \mathbf{y}^{(i)}\right)$.

In our ablation study below, we show that such an uncertainty measure reliably outperforms consistency-based measures that aggregate the pairwise similarities of all samples (see Appendix C.2).

A key strength of consistency-based techniques is that, by generating multiple samples and analyzing their semantic similarity, they can estimate empirical probabilities over *meanings* rather than over individual answers. Their main drawback is that they discard the useful information that comes from the probability distribution represented by the LLM, including estimates of the probabilities of the specific answers.

## 3 CoCoA: Bridging Confidence and Consistency for Better Uncertainty Quantification

Both information-based and (semantic) consistency-based methods provide grounded and useful uncertainty quantification measures. There exist approaches that bridge the gap between information-based and consistency-based methods that show great promise but lack the fundamental base; see discussion in Section 4. In what follows, we present a family of *Confidence and Consistency-based Approaches* (CoCoA) for UQ, offering a new way to merge information- and consistency-based measures for uncertainty quantification in LLMs via the unifying MBR-based framework.

If we only consider the consistency measure, e.g., as given in (8), we miss the information that is contained in the model confidence. Thus, we want the resulting uncertainty measure to explicitly consider both the semantic consistency and the model-based uncertainty. Let us consider a risk of the following form:

$$r(\mathbf{y}, \mathbf{y}' \mid \mathbf{x}) = u(\mathbf{y} \mid \mathbf{x}) \cdot \big(1 - s(\mathbf{y}, \mathbf{y}')\big), \tag{9}$$

where

- $s(\mathbf{y}, \mathbf{y}') \in [0, 1]$ is any utility function used in the standard MBR decoding (usually representing semantic similarity);

- $u(\mathbf{y} \mid \mathbf{x}) \geq 0$ is the model-based uncertainty measure for the output sequence $\mathbf{y}$. For example, we can use $u(\mathbf{y} \mid \mathbf{x}) = 1 - p(\mathbf{y} \mid \mathbf{x})$ (similarly to (6)) or $u(\mathbf{y} \mid \mathbf{x}) = -\log p(\mathbf{y} \mid \mathbf{x})$.

Then, the corresponding Bayes risk is $R_{\mathrm{CoCoA}}(\mathbf{y} \mid \mathbf{x}) = u(\mathbf{y} \mid \mathbf{x}) \cdot \mathbb{E}_{\mathbf{y}' \sim p(\mathbf{y} \mid \mathbf{x})} \big(1 - s(\mathbf{y}, \mathbf{y}')\big)$ and for an output sequence $\mathbf{y}_*$, the resulting uncertainty measure becomes

$$U_{\mathrm{CoCoA}}(\mathbf{y}_* \mid \mathbf{x}) = u(\mathbf{y}_* \mid \mathbf{x}) \cdot \mathbb{E}_{\mathbf{y}' \sim p(\mathbf{y} \mid \mathbf{x})} \big(1 - s(\mathbf{y}_*, \mathbf{y}')\big). \tag{10}$$

Finally, given a set of samples $\big\{\mathbf{y}^{(i)}\big\}_{i=1}^{M}$ we obtain an empirical uncertainty estimate:

$$\widehat{U}_{\mathrm{CoCoA}}(\mathbf{y}_* \mid \mathbf{x}) = u(\mathbf{y}_* \mid \mathbf{x}) \cdot \frac{1}{M} \sum_{i=1}^{M} (1 - s_{*i}) = u(\mathbf{y}_* \mid \mathbf{x}) \cdot \widehat{U}_{\mathrm{cons}}(\mathbf{y}_* \mid \mathbf{x}), \tag{11}$$

where $s_{*i} = s\big(\mathbf{y}_*, \mathbf{y}^{(i)}\big)$.

The resulting uncertainty measure integrates both global (semantic) and local (model-specific) uncertainty signals. It ensures that uncertainty is amplified for sequences $\mathbf{y}_*$ that are both intrinsically uncertain (high $u(\mathbf{y}_* \mid \mathbf{x})$) and semantically inconsistent with respect to the other samples (high $\frac{1}{M} \sum_{i=1}^{M} (1 - s_{*i})$), while keeping it low for the opposite scenario; see Figure 2 for an example.

**CoCoA Light.** Computing the consistency-based uncertainty measure $\widehat{U}_{\mathrm{cons}}(\mathbf{y}_* \mid \mathbf{x})$ within $\widehat{U}_{\mathrm{CoCoA}}(\mathbf{y}_* \mid \mathbf{x})$ requires multiple samples, whoch poses significant computational overhead. To address this, we propose to approximate the behavior of $\widehat{U}_{\mathrm{cons}}(\mathbf{x})$ with a learned function. Our method closely matches the original sampling-based measure in quality while requiring only greedy output generation during inference, eliminating the need for additional sampling. Notably, this approximation can be learned without access to ground-truth labels, relying solely on the input data and the associated uncertainty values.

Thus, the CoCoA uncertainty measure that incorporates an approximation of the consistency uncertainty $\widehat{U}_{\mathrm{cons}}^{L}(\mathbf{y}_* \mid \mathbf{x}) \approx \widehat{U}_{\mathrm{cons}}(\mathbf{y}_* \mid \mathbf{x})$ can be defined as follows:

$$\widehat{U}_{\mathrm{CoCoA}}^{L}(\mathbf{y}_* \mid \mathbf{x}) = u(\mathbf{y}_* \mid \mathbf{x}) \cdot \widehat{U}_{\mathrm{cons}}^{L}(\mathbf{y}_* \mid \mathbf{x}), \tag{12}$$

We name this approach `CoCoA Light`. Below, we describe how this approximation can be obtained.

Assume access to a held-out set of input sequences $\mathbf{x}_j, j = 1, \ldots, n$. We emphasize that this held-out set is not labeled. For each sequence, we extract the embeddings $\{e(\mathbf{y}_j \mid \mathbf{x}_j)\}_{j=1}^{n}$ of the corresponding greedy model generations $\mathbf{y}_j$. The corresponding targets are the consistency uncertainty scores $\widehat{U}_{\mathrm{cons}}(\mathbf{y}_j \mid \mathbf{x}_j), j = 1, \ldots, n$, computed via Monte Carlo estimation of the minimum Bayes risk; see equation (8).

A lightweight auxiliary model, $\widehat{g}\big(e(\mathbf{y}_* \mid \mathbf{x})\big)$, is trained in a supervised fashion to map model embeddings to uncertainty scores. During inference, the model generates a greedy output $\mathbf{y}_{\mathrm{test}}$ for a test input $\mathbf{x}_{\mathrm{test}}$, from which the embeddings $e(\mathbf{y}_{\mathrm{test}} \mid \mathbf{x}_{\mathrm{test}})$ are obtained and passed to the auxiliary model to predict the uncertainty score:

$$\widehat{U}_{\mathrm{cons}}^{L}(\mathbf{y}_{\mathrm{test}} \mid \mathbf{x}_{\mathrm{test}}) = \widehat{g}\big(e(\mathbf{y}_{\mathrm{test}} \mid \mathbf{x}_{\mathrm{test}})\big). \tag{13}$$

We note that a similar approach [Kossen et al., 2024] has been previously proposed to approximate semantic entropy using the hidden states from the model.

# 4 Related Work

In this section, we review existing approaches to uncertainty quantification, as well as prior work on Minimum Bayes Risk in application to natural language generation.

**Information-based methods.**    These methods are one of the most commonly used. They quantify uncertainty by analyzing the probability distributions of the tokens within a given output. The simplest of these methods, *Sequence Probability (SP)*, measures the probability of the sequence given a specific input. *Perplexity* is another common uncertainty measure: it gauges how well the model predicts each token in a sequence and is formally defined as the exponential of the mean negative log-likelihood of those tokens [Fomicheva et al., 2020]. The *Mean Token Entropy* method evaluates the token-level predictions across the entire sequence by computing the average entropy of the token probability distributions at each position [Fomicheva et al., 2020]. While these methods provide useful uncertainty estimates, they do not address the broader uncertainty inherent in the generative tasks, as they rely on a single sequence and fail to capture the diversity of the possible outputs that the model could generate for the same input.

**Consistency-Based Methods.**    These methods estimate uncertainty by generating multiple sequences from the model's output distribution and analyzing the variability among the sampled outputs. These methods are particularly relevant to NLP tasks, as they account for scenarios in which multiple plausible outputs may exist for a given input. Among the simplest consistency-based methods are the *Number of Semantic Sets* and the *Sum of Eigenvalues of the Graph Laplacian* [Lin et al., 2024], which quantify uncertainty by how many distinct "meanings" the model produces. While effective at capturing global variation, they do not yield uncertainty scores for individual responses. To overcome this, the diagonal of the *Degree Matrix* was proposed to assess how similar each output is to the rest, enabling per-response uncertainty quantification [Lin et al., 2024]. A related approach, *Lexical Similarity*, operates by calculating the average similarity of words or phrases across every pair of responses in the sample [Fomicheva et al., 2020].

**Information-Based Methods with Repeated Sampling.**    More recent methods attempt to reconcile output consistency with information-based signals—i.e., they not only look at how varied outputs are, but also how probable each output is under the model's own generation process. For instance, *Semantic Entropy* [Kuhn et al., 2023] groups outputs into semantically homogeneous clusters—capturing distinct "meanings" that might just differ in phrasing—and calculates entropy across these clusters. *SentenceSAR* refines this idea by defining "relevance" as the sum of pairwise similarities between a given sentence and others weighted by each sentence's model-assigned generation probability [Duan et al., 2024]. *SAR* combines the sentence-level relevance of SentenceSAR with token-level probability adjustments, attempting a more fine-grained balance of semantic and token-level uncertainty [Duan et al., 2024]. *Semantic Density* [Qiu and Miikkulainen, 2024] evaluates uncertainty by evaluating how densely a model's generated response is situated within the semantic space of all possible outputs, with lower density indicating higher uncertainty. One limitation of sampling-based approaches is their computational cost: generating multiple outputs from the model can be expensive. Recent work has sought to address this by learning to predict semantic uncertainty directly from hidden states, allowing for the approximation of semantic entropy without repeated sampling [Kossen et al., 2024]. Finally, BSDetector [Chen and Mueller, 2024] proposes an additive composition of self-reported confidence with observed consistency, however, it relies on ability of the model to assess its own confidence in text form. Another limitation is that it requires selection of a trade-off coefficient between confidence and consistency parts.

While these probability-weighted measures can provide deeper insights, they also sometimes underperform in practice [Vashurin et al., 2025], especially when the model's probability estimates are unreliable, or when important nuances of semantic diversity are lost in the weighting process. Proper balancing raw output consistency with generation likelihood remains an open problem.

**Minimum Bayes Risk for LLMs.**    Minimum Bayes Risk (MBR) decoding, originally used in machine translation [Kumar and Byrne, 2004b], has recently been applied to LLMs by incorporating a posterior over model parameters to enable uncertainty-aware generation [Daheim et al., 2025]. Complementing this, a Bayesian framework for subjective uncertainty quantification was developed yielding improved calibration in generation tasks [Wang and Holmes, 2025].

# 5 Experiments

In this section, we present the experimental setup, the results, and the ablations.

## 5.1 Experimental Setup

To evaluate the effectiveness of our proposed framework, we extended the `LM-Polygraph` library [Vashurin et al., 2025, Fadeeva et al., 2023]. Since it already includes tools for calculating other uncertainty scores, it provided a convenient and efficient environment for setting up and running experiments. The primary objective of our experiments is to evaluate whether our method offers improved performance in key tasks such as question answering (QA), text summarization (SUM), and machine translation (MT), compared to existing baselines.

**Datasets.** For QA, we selected diverse datasets to capture a variety of challenges: TriviaQA [Joshi et al., 2017], an open-domain factual QA dataset; CoQA [Reddy et al., 2019], a conversational QA benchmark requiring multi-turn contextual understanding; MMLU [Hendrycks et al., 2021], a multi-task dataset spanning 57 topics to test broad knowledge; and GSM8k [Cobbe et al., 2021], which focuses on grade-school math problems requiring logical reasoning. For translation, we evaluated our method on WMT14 French-English [Bojar et al., 2014] and WMT19 German-English [Barrault et al., 2019]. Finally, for summarization, we used XSUM [Narayan et al., 2018], a dataset of complex documents paired with concise abstractive summaries. For all datasets, we follow [Vashurin et al., 2025] in subset selection, prompt formatting, and few-shot example sourcing.

**Models.** We evaluated our method on the base versions of three open-weight language models: LLaMA 3.1 8B [Touvron et al., 2023], Mistral 7B [Jiang et al., 2023], and Falcon 3 7B [Team, 2024]. The open-weight nature of these models enables direct access to token probabilities, which is crucial for implementing our UQ method. All experiments were conducted using the base (non–instruction-tuned) variants. We additionally report results on the larger Gemma 3 12B-Base model [Team et al., 2025], with detailed analysis provided in Appendix J.

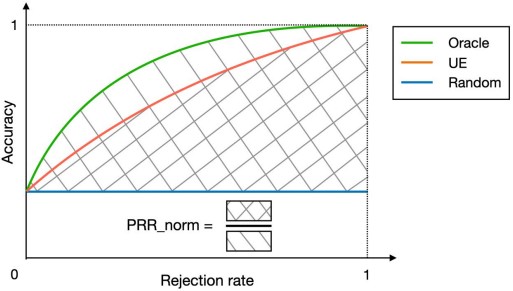

Figure 3: *Prediction-Rejection Ratio (PRR) Curve* illustrating the quality of the non-rejected predictions as a function of the rejection rate. *Oracle* represents the optimal rejection strategy, *Random* is a random rejection, and *UQ* is rejection based on the evaluated uncertainty quantification method.

**Similarity Function.** To measure the similarity between two generations, we use the RoBERTa-large cross-encoder model, fine-tuned on the Semantic Textual Similarity benchmark dataset [Liu et al., 2019, Reimers and Gurevych, 2019, Cer et al., 2017]. This model is widely regarded as one of the most reliable and commonly used approaches for evaluating sentence similarity. The cross-encoder processes two sequences jointly and directly outputs a similarity score ranging from 0 to 1, providing a nuanced measure. Appendix C.1 contains comparative experiments with cross-encoder and other choices of the similarity function, substantiating this choice.

**Baselines.** We compare the performance of the proposed method against a diverse set of baselines and state-of-the-art UQ scores, including confidence-based, consistency-based, and hybrid approaches. For information-based approaches, we evaluate Sequence Probability (SP), Perplexity (PPL), Mean Token Entropy (MTE), Monte Carlo Sequence Entropy (MCSE), and Monte Carlo Normalized Sequence Entropy (MCNSE). In the consistency-based category, we consider the Degree Matrix (DegMat) and the Sum of Eigenvalues of the Graph Laplacian (EigValLaplacian). Finally, we include hybrid methods Semantic Entropy and SAR, as well as verbalized confidence method P(true) [Kadavath et al., 2022]. All formulations for these baselines can be found in Appendix D. Finally, we evaluate the performance of $\widehat{U}_{\text{cons}}$ (Consistency) and $\widehat{U}_{\text{cons}}^{L}$ (Consistency Light) as an uncertainty measure.

**Evaluation Measure.** As our evaluation measure, we chose the *Prediction Rejection Ratio (PRR)*, which measures the effectiveness of the uncertainty scores for identifying high-quality predictions [Malinin and Gales, 2021]. PRR operates by progressively rejecting predictions with uncertainty scores above a threshold $a$ and observing how the average quality of the remaining predictions changes (see Figure 3). It is calculated as the ratio of two areas: the area between the Prediction Rejection (PR) curves for the evaluated uncertainty score and a random baseline, and the area between the oracle (the ideal uncertainty score that perfectly ranks instances by quality) and the random baseline. Formally, PRR is defined as follows:

$$PRR = \frac{\text{AUC}_{\text{unc}} - \text{AUC}_{\text{rnd}}}{\text{AUC}_{\text{oracle}} - \text{AUC}_{\text{rnd}}}. \tag{14}$$

Higher PRR values indicate better alignment of uncertainty scores with prediction quality, approaching the performance of an oracle. To ensure practical applicability, we compute PRR only up to a rejection threshold of 50%, preventing cases where excessive rejection artificially inflates the quality measures.

For the QA datasets, we further report AUROC (see Appendix G).

**Quality Measures.** PRR requires an appropriate quality measure for each specific task to effectively evaluate the model output. For question–answering tasks, we use *Accuracy* to directly evaluate whether the generated answers match the ground truth in short-form QA tasks (e.g., MMLU), and we use the *AlignScore* between the correct answer and generated sequence for assessing the performance for long-form QA tasks [Zha et al., 2023]. For summarization tasks, we use *AlignScore* to measure the alignment between the output summary and the input document. It serves as a quality indicator by evaluating the relevance and the overlap between the generated content and the source text. For translation tasks, we use *COMET*, as it captures both semantic adequacy and fluency, ensuring that translations are accurate and linguistically appropriate [Rei et al., 2020].

To further improve the comprehensiveness of the reported results, we considered alternative choices of quality measures, and report the corresponding PRR scores in Appendix F.

**Generation Setup.** We discuss the generation parameters, the decoding strategy, and the sample selection procedure in depth in Appendix A. In short, we report the evaluation results in two distinct setups: greedy decoding and stochastic sampling with focus on the most probable sequence among the generated outputs (*best-sample*). These two setups offer the highest-quality outputs and are the most reasonable generation approaches in practice.

**CoCoA Light Details.** We use a multilayer perceptron network as an auxiliary model. The features used are embeddings from the middle layers of the base LLM, which are the most informative according to recent work [Chen et al., 2024]. In particular, for Llama 3.1 8B and Mistral 7B models, we take embeddings from the 16th layer, and for Falcon 3 7B from the 14th layer. More details on the training are available in Appendix I.

## 5.2 Results

Table 1 shows the PRR scores under the *greedy* generation setup (See Appendix B for Best Sample and MBR Sample). We report aggregated PRR for each type of task – question answering, neural machine translation (NMT), and summarization (SUM) – by averaging the results across all relevant datasets (e.g., TriviaQA, MMLU, CoQA, GSM8k for QA). This aggregated score provides a concise measure of the performance for each model for each task. Detailed results for each dataset separately can be found in Appendix E.

We can see that our CoCoA methods are *the best* across all tasks and models. They outperform existing consistency-based and hybrid state-of-the-art approaches, like Semantic Entropy and SAR. In addition, the proposed CoCoA approach consistently surpasses the baseline UQ measures: for example, CoCoA$_{PPL}$ outperforms standard Perplexity, illustrating the advantage of combining token-level confidence with semantic consistency. This pattern holds for other information-based metrics as well, demonstrating that using the consistency between multiple sampled outputs reliably enhances uncertainty quantification.

| Metric | Llama8b-Base | | | Mistral7b-Base | | | Falcon7b-Base | | |
|---|---|---|---|---|---|---|---|---|---|
| | **QA** | **NMT** | **SUM** | **QA** | **NMT** | **SUM** | **QA** | **NMT** | **SUM** |
| MCSE | 0.310 | 0.323 | 0.033 | 0.389 | 0.304 | 0.007 | 0.414 | 0.317 | 0.159 |
| MCNSE | 0.309 | 0.393 | 0.022 | 0.384 | 0.410 | 0.009 | 0.405 | 0.422 | 0.108 |
| Semantic Entropy | 0.356 | 0.343 | 0.033 | 0.423 | 0.327 | 0.008 | 0.439 | 0.348 | 0.164 |
| DegMat | 0.406 | 0.302 | 0.081 | 0.423 | 0.305 | 0.137 | 0.483 | 0.353 | 0.201 |
| EigValLaplacian | 0.375 | 0.238 | 0.079 | 0.391 | 0.267 | 0.132 | 0.459 | 0.312 | 0.201 |
| SAR | 0.414 | 0.455 | 0.077 | 0.462 | 0.435 | 0.094 | 0.481 | 0.458 | 0.144 |
| P(True) | -0.064 | 0.042 | 0.058 | -0.029 | 0.075 | 0.179 | 0.118 | 0.155 | -0.159 |
| Consistency | 0.437 | 0.421 | 0.024 | 0.471 | 0.392 | 0.051 | 0.494 | 0.416 | 0.226 |
| Consistency Light | 0.390 | 0.458 | -0.022 | 0.444 | 0.387 | -0.006 | 0.427 | 0.476 | 0.232 |
| SP | 0.409 | 0.399 | 0.328 | 0.475 | 0.383 | 0.287 | 0.475 | 0.356 | 0.201 |
| CoCoA$_{SP}$ | 0.451 ↑ | **0.519** ↑ | 0.378 ↑ | **0.509** ↑ | **0.497** ↑ | **0.330** ↑ | 0.511 ↑ | 0.505 ↑ | **0.257** ↑ |
| CoCoA$_{SP}$ Light | 0.449 ↑ | 0.502 ↑ | 0.358 ↑ | 0.503 ↑ | 0.480 ↑ | 0.309 ↑ | 0.492 ↑ | 0.514 ↑ | 0.242 ↑ |
| PPL | 0.381 | 0.386 | 0.369 | 0.424 | 0.427 | 0.204 | 0.456 | 0.450 | 0.155 |
| CoCoA$_{PPL}$ | **0.454** ↑ | 0.481 ↑ | **0.387** ↑ | 0.494 ↑ | 0.472 ↑ | 0.286 ↑ | 0.523 ↑ | 0.508 ↑ | 0.229 ↑ |
| CoCoA$_{PPL}$ Light | 0.445 ↑ | 0.487 ↑ | 0.382 ↑ | 0.479 ↑ | 0.480 ↑ | 0.260 ↑ | 0.512 ↑ | 0.528 ↑ | 0.234 ↑ |
| MTE | 0.353 | 0.382 | 0.357 | 0.417 | 0.438 | 0.182 | 0.456 | 0.473 | 0.152 |
| CoCoA$_{MTE}$ | 0.447 ↑ | 0.478 ↑ | 0.380 ↑ | 0.492 ↑ | 0.469 ↑ | 0.288 ↑ | **0.527** ↑ | 0.508 ↑ | 0.228 ↑ |
| CoCoA$_{MTE}$ Light | 0.428 ↑ | 0.494 ↑ | 0.372 ↑ | 0.475 ↑ | 0.482 ↑ | 0.254 ↑ | 0.507 ↑ | **0.533** ↑ | 0.234 ↑ |

Table 1: Results for Evaluated Sequence – Greedy Sample: Mean PRR across datasets for each task. The best-performing method is shown in bold, and the second-best is underscored. The arrows indicate improvement in CoCoA over the base version.

## 5.3 Ablations

**Similarity Function.** We investigate the impact of different similarity measures (see Appendix C.1). On average, the cross-encoder provides strong performance. However, for summarization tasks, AlignScore yields better results, while in long-form QA, similarity based on Natural Language Inference (NLI) sometimes outperforms the cross-encoder. We hypothesize that for generation with shorter outputs (1–2 sentences), any capable NLI or cross-encoder model is sufficient. For longer generations, approaches that estimate similarity over chunks and then aggregate scores (e.g., AlignScore) may be more effective. Nevertheless, across all tasks, the cross-encoder achieves the best overall performance and, in scenarios where tuning is not feasible, serves as a strong default choice.

**Alternative Formulations.** The next section of our ablation study focuses on alternative forms of combining model confidence $u(\mathbf{y}_* \mid \mathbf{x})$ and consistency $\widehat{U}_{\text{cons}}(\mathbf{y}_* \mid \mathbf{x})$; see Appendix C.2. First, we consider an additive form of combining them:

$$\widehat{U}_{\text{AdditiveCoCoA}}(\mathbf{y}_* \mid \mathbf{x}) = u(\mathbf{y}_* \mid \mathbf{x}) + \widehat{U}_{\text{cons}}(\mathbf{y}_* \mid \mathbf{x}). \qquad (15)$$

The results show that this additive formulation does not perform as well as the multiplicative one. The additive form tends to underemphasize the interaction between the two components, which is critical for capturing the nuanced relationships between confidence and consistency.

We also consider an alternative formulation of the consistency term $\widehat{U}_{\text{cons}}(\mathbf{y}_* \mid \mathbf{x})$, as the average of the full pairwise dissimilarity. In this formulation, $\widehat{U}_{\text{cons}}(\mathbf{y}_* \mid \mathbf{x})$ represents the average inconsistency across all samples rather than focusing solely on the dissimilarity of the evaluated sequence with the other samples. Our experiments demonstrate that this formulation is not very strong. By distributing the consistency computation across all samples, it loses focus on the specific sequence being evaluated.

Lastly, in Appendix C.2, we also consider alternative formulations of the information-based metric that do not rely on logarithmic transformations. While we primarily use logarithms due to their numerical stability, we explore an alternative approach by converting these values back to probabilities and analyzing their impact on uncertainty quantification. Our findings indicate that both formulations exhibit consistent performance and yield similar results. This suggests that while logarithmic transformations enhance numerical stability, the choice between log-based and probability-based formulations does not affect much the overall performance.

# 6   Limitations

While our proposed CoCoA approach demonstrates robust empirical performance, several important considerations remain.

**Task and Domain Dependency.** CoCoA relies on an information-based confidence score and a semantic similarity function, whose effectiveness can vary across models, tasks, and domains. In open-ended tasks like creative generation, producing diverse outputs is expected; on the other hand, tasks requiring precise reasoning can be sensitive to subtle errors that generic similarity metrics may miss. Adapting these components to specific domains remains an important direction for future work.

**Limited Sample Size.** CoCoA estimates consistency by sampling multiple outputs. Generating many samples is computationally expensive, and thus our experiments, e.g., most sampling-based methods, use relatively small sets. While even a few samples can yield meaningful estimates, they may miss the full diversity of model outputs for complex prompts.

**Quality Metric.**   Finally, the CoCoA's performance assessment depends on quality metrics (e.g., COMET for machine translation, and Accuracy for QA) that may not capture every nuance of textual outputs. Further refining or extending quality metrics to account for deeper reasoning, factual faithfulness, and stylistic appropriateness would better align uncertainty scores with real-world perceptions of model correctness.

# 7   Conclusion and Future Work

We presented CoCoA, a unified approach that integrates **Co**nfidence and **Co**nsistency for uncertainty quantification in LLMs. By combining confidence scores with semantic similarity between multiple sampled outputs, CoCoA offers a more holistic view of uncertainty than either approach alone. In extensive evaluations on question answering, summarization, and translation, our approach outperformed existing baselines and state-of-the-art UQ methods. Moreover, CoCoA's flexible design allows easy adaptation to a variety of tasks and settings.

Moving forward, several directions are open for further exploration. These include incorporating more adaptive sampling strategies that efficiently capture the model output space, refining the semantic similarity functions for domain-specific tasks, like code generation or commonsense reasoning.

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

# A    Decoding Strategy and Sample Selection

Modern LLMs are capable of producing output using a wide range of decoding strategies, and it is not readily apparent which one to use as a foundation for UQ experiments. On top of that, when sampling multiple outputs stochastically, one has to decide which sample to select for comparison with the target sequence and UQ purposes.

To facilitate the choice of decoding and sample selection strategies for our experiments, we conducted an evaluation of model performance with different approaches to both. Table 2 shows average values of corresponding quality metrics for all combinations of models and datasets. We considered 4 approaches for the selection of output that subsequently is used to calculate the quality of generation:

- **Greedy decoding** produces single output by selecting top-1 candidate token at each generation step, thus no further selection of sample is needed.

- **Random sample** corresponds to the case where random output is selected among the number of samples produced by repeatedly prompting the model with the same question. In practice we use first generated sample, highlighting model performance when stochastic decoding is done only once.

- **Most Probable sample** selects the output with highest model-assigned probability among several sampled outputs.

- **MBR sample** selects the output with highest average consistency with respect to other sampled outputs.

We note that selecting a random sample from the model outputs incurs a significant drop in the quality of results on several datasets, most prominently on GSM8k. Based on these observations, we evaluate the efficacy of UQ in three setups: greedy decoding, stochastic sampling with a focus on the most probable sample and MBR decoding.

Regardless of the way the main model response was obtained, responses that were used to quantify consistency in (8) were generated via repeated stochastic sampling. In all the experiments, where stochastic sampling is involved, it was performed with temperature $T = 1.0$, top-k equal to 50, and top-p equal to $1.0$.

| Sample | Metric | Greedy | Random Sample | Most Probable Sample | MBR Sample |
|---|---|---|---|---|---|
| | | | Falcon7b-Base | | |
| CoQA | Align Score | 0.793 | 0.706 | 0.783 | 0.793 |
| GSM8k | Accuracy | 0.776 | 0.313 | 0.205 | 0.419 |
| MMLU | Accuracy | 0.715 | 0.638 | 0.715 | 0.699 |
| TriviaQA | Align Score | 0.557 | 0.473 | 0.568 | 0.559 |
| WMT14FrEn | Comet | 0.867 | 0.833 | 0.857 | 0.855 |
| WMT19DeEn | Comet | 0.846 | 0.807 | 0.826 | 0.828 |
| XSUM | Align Score | 0.842 | 0.734 | 0.782 | 0.801 |
| | | | Llama8b-Base | | |
| CoQA | Align Score | 0.756 | 0.671 | 0.743 | 0.766 |
| GSM8k | Accuracy | 0.548 | 0.234 | 0.261 | 0.346 |
| MMLU | Accuracy | 0.570 | 0.368 | 0.577 | 0.469 |
| TriviaQA | Align Score | 0.686 | 0.625 | 0.687 | 0.699 |
| WMT14FrEn | Comet | 0.863 | 0.819 | 0.852 | 0.844 |
| WMT19DeEn | Comet | 0.870 | 0.816 | 0.854 | 0.845 |
| XSUM | Align Score | 0.848 | 0.608 | 0.825 | 0.703 |
| | | | Mistral7b-Base | | |
| CoQA | Align Score | 0.793 | 0.695 | 0.777 | 0.790 |
| GSM8k | Accuracy | 0.382 | 0.169 | 0.190 | 0.259 |
| MMLU | Accuracy | 0.632 | 0.552 | 0.632 | 0.612 |
| TriviaQA | Align Score | 0.743 | 0.655 | 0.750 | 0.739 |
| WMT14FrEn | Comet | 0.863 | 0.812 | 0.830 | 0.841 |
| WMT19DeEn | Comet | 0.864 | 0.805 | 0.836 | 0.837 |
| XSUM | Align Score | 0.803 | 0.578 | 0.775 | 0.665 |

Table 2: Base quality metrics for models for different evaluated sequence choice.

## B    Results Summary for Most Probable Sample and MBR Sample strategies

Tables 3 and 4 report the PRR scores under the *Most Probable Sample* and *MBR Sample* generation setups, as discussed in Section 5.2. We observe that CoCoA-family methods, and their Supervised versions, consistently improve base uncertainty estimators.

| Metric | Llama8b-Base | | | Mistral7b-Base | | | Falcon7b-Base | | |
|---|---|---|---|---|---|---|---|---|---|
| | QA | NMT | SUM | QA | NMT | SUM | QA | NMT | SUM |
| MCSE | 0.356 | 0.380 | 0.192 | 0.453 | 0.406 | 0.162 | 0.460 | 0.409 | 0.128 |
| MCNSE | 0.380 | 0.429 | 0.186 | 0.466 | 0.489 | 0.196 | 0.530 | 0.424 | 0.153 |
| Semantic Entropy | 0.396 | 0.411 | 0.194 | 0.482 | 0.438 | 0.164 | 0.479 | 0.440 | 0.134 |
| DegMat | 0.422 | 0.342 | 0.191 | 0.465 | 0.425 | 0.205 | 0.543 | 0.386 | 0.177 |
| EigValLaplacian | 0.388 | 0.274 | 0.190 | 0.426 | 0.366 | 0.197 | 0.498 | 0.336 | 0.174 |
| SAR | 0.478 | 0.506 | 0.159 | 0.542 | 0.576 | 0.175 | 0.590 | 0.488 | 0.193 |
| P(True) | -0.071 | 0.066 | 0.058 | 0.051 | 0.371 | 0.207 | 0.281 | 0.245 | 0.022 |
| Consistency | 0.535 | 0.536 | 0.030 | 0.572 | 0.689 | 0.071 | 0.626 | 0.571 | 0.282 |
| SP | 0.395 | 0.376 | 0.464 | 0.444 | 0.252 | 0.330 | 0.343 | 0.381 | 0.099 |
| CoCoA$_{SP}$ | 0.484 ↑ | 0.607 ↑ | **0.484** ↑ | 0.526 ↑ | 0.721 ↑ | 0.366 ↑ | 0.529 ↑ | 0.631 ↑ | 0.210 ↑ |
| PPL | 0.532 | 0.563 | 0.458 | 0.587 | 0.686 | 0.365 | 0.627 | 0.589 | 0.275 |
| CoCoA$_{PPL}$ | **0.571** ↑ | **0.617** ↑ | 0.450 | **0.613** ↑ | **0.745** ↑ | 0.372 ↑ | **0.647** ↑ | **0.648** ↑ | **0.310** ↑ |
| MTE | 0.476 | 0.469 | 0.449 | 0.559 | 0.637 | 0.350 | 0.602 | 0.492 | 0.186 |
| CoCoA$_{MTE}$ | 0.547 ↑ | 0.579 ↑ | 0.451 ↑ | 0.600 ↑ | 0.720 ↑ | **0.373** ↑ | 0.641 ↑ | 0.614 ↑ | 0.289 ↑ |

Table 3: Results for Evaluated Sequence – Most Probable Sample: Mean PRR across datasets for each task. The best performing method is in bold, and the second-best is underscored. Arrows indicate improvement in CoCoA over the base version.

| Metric | Llama8b-Base | | | Mistral7b-Base | | | Falcon7b-Base | | |
|---|---|---|---|---|---|---|---|---|---|
| | QA | NMT | SUM | QA | NMT | SUM | QA | NMT | SUM |
| MCSE | 0.402 | 0.297 | 0.158 | 0.474 | 0.330 | 0.237 | 0.437 | 0.346 | 0.076 |
| MCNSE | 0.392 | 0.389 | 0.124 | 0.441 | 0.435 | 0.192 | 0.446 | 0.440 | 0.078 |
| Semantic Entropy | 0.443 | 0.316 | 0.159 | 0.507 | 0.352 | 0.236 | 0.458 | 0.373 | 0.077 |
| DegMat | 0.478 | 0.290 | 0.047 | 0.471 | 0.322 | 0.124 | 0.499 | 0.371 | -0.018 |
| EigValLaplacian | 0.434 | 0.224 | 0.037 | 0.437 | 0.279 | 0.115 | 0.460 | 0.326 | -0.022 |
| SAR | 0.509 | 0.456 | 0.187 | 0.532 | 0.448 | 0.266 | 0.531 | 0.478 | 0.083 |
| P(True) | -0.049 | 0.094 | -0.014 | -0.044 | 0.113 | 0.079 | 0.160 | 0.178 | -0.124 |
| Consistency | 0.504 | 0.365 | 0.235 | 0.518 | 0.367 | 0.284 | 0.538 | 0.410 | 0.099 |
| SP | 0.415 | 0.404 | 0.266 | 0.447 | 0.433 | 0.312 | 0.405 | 0.376 | **0.170** |
| CoCoA$_{SP}$ | 0.527 ↑ | 0.496 ↑ | **0.293** ↑ | 0.553 ↑ | **0.515** ↑ | **0.348** ↑ | 0.500 ↑ | 0.526 ↑ | 0.154 |
| PPL | 0.408 | 0.421 | 0.164 | 0.432 | 0.452 | 0.273 | 0.445 | 0.438 | 0.100 |
| CoCoA$_{PPL}$ | 0.532 ↑ | **0.504** ↑ | 0.215 ↑ | 0.554 ↑ | 0.514 ↑ | 0.328 ↑ | 0.552 ↑ | **0.537** ↑ | 0.109 ↑ |
| MTE | 0.467 | 0.461 | 0.148 | 0.515 | 0.496 | 0.219 | 0.490 | 0.521 | 0.069 |
| CoCoA$_{MTE}$ | **0.542** ↑ | 0.479 ↑ | 0.201 ↑ | **0.574** ↑ | 0.486 | 0.290 ↑ | **0.562** ↑ | 0.527 ↑ | 0.081 ↑ |

Table 4: Results for Evaluated Sequence - MBR Decoding: Mean PRR across datasets for each task. The best performing method is in bold, and the second-best is underscored. Arrows indicate improvement in CoCoA over the base version.

# C Ablation

## C.1 Choice of Similarity Function

For sample consistency estimation, one could come up with a variety of similarity functions $s(\mathbf{y}, \mathbf{y}')$. We perform a comparison of the effectiveness of `CoCoA`-family methods using several such functions. We consider the following functions:

- AlignScore [Zha et al., 2023] with `AlignScore-large` model;
- RougeL [Lin, 2004];
- NLI [He et al., 2021] based on `microsoft/deberta-large-mnli` model;
- CrossEncoder [Liu et al., 2019] based on `cross-encoder/stsb-roberta-large` model.

Tables 5, 6 and 7 report these results. There exists a considerable variation of relative effectiveness of proposed methods with various similarity function choices, depending on a task at hand. We opt to report all results in other sections with CrossEncoder-based similarity as it by itself provides a significant improvement over baselines, and for consistency and ease of comparison reasons. However, when applying these methods to a particular task, we encourage users to select appropriate underlying similarity function for best results.

| Method | Dataset | | | | | | |
|---|---|---|---|---|---|---|---|
| | XSUM | WMT14FrEn | WMT19DeEn | CoQA | TriviaQA | MMLU | GSM8k |
| **Mistral7b-Base** | | | | | | | |
| $CoCoA_{SP}$ | | | | | | | |
| AlignScore | **0.334** | 0.293 | 0.445 | 0.354 | 0.655 | 0.466 | 0.550 |
| RougeL | 0.289 | 0.358 | 0.546 | 0.369 | 0.649 | 0.466 | 0.536 |
| NLI | 0.308 | 0.313 | 0.477 | **0.403** | 0.677 | **0.470** | **0.568** |
| CrossEncoder | 0.327 | **0.397** | **0.595** | 0.381 | 0.671 | 0.466 | 0.505 |
| $CoCoA_{PPL}$ | | | | | | | |
| AlignScore | 0.307 | 0.308 | 0.489 | 0.373 | 0.666 | 0.466 | 0.536 |
| RougeL | 0.226 | 0.369 | 0.531 | 0.352 | 0.653 | 0.466 | 0.466 |
| NLI | 0.233 | 0.316 | 0.501 | 0.376 | **0.682** | **0.470** | 0.480 |
| CrossEncoder | 0.281 | 0.371 | 0.565 | 0.365 | 0.674 | 0.466 | 0.465 |
| $CoCoA_{MTE}$ | | | | | | | |
| AlignScore | 0.302 | 0.299 | 0.477 | 0.366 | 0.664 | 0.450 | 0.555 |
| RougeL | 0.212 | 0.377 | 0.528 | 0.345 | 0.652 | 0.449 | 0.497 |
| NLI | 0.219 | 0.313 | 0.488 | 0.362 | 0.681 | 0.453 | 0.490 |
| CrossEncoder | 0.282 | 0.368 | 0.560 | 0.351 | 0.673 | 0.448 | 0.486 |
| **Llama8b-Base** | | | | | | | |
| $CoCoA_{SP}$ | | | | | | | |
| AlignScore | 0.367 | 0.331 | 0.452 | 0.308 | 0.596 | **0.484** | 0.401 |
| RougeL | 0.336 | 0.393 | 0.545 | 0.321 | 0.563 | 0.474 | 0.375 |
| NLI | 0.344 | 0.352 | 0.467 | **0.364** | 0.606 | 0.478 | 0.419 |
| CrossEncoder | 0.375 | **0.454** | **0.583** | 0.350 | 0.598 | 0.480 | 0.367 |
| $CoCoA_{PPL}$ | | | | | | | |
| AlignScore | **0.422** | 0.346 | 0.450 | 0.337 | 0.596 | 0.453 | 0.446 |
| RougeL | 0.370 | 0.408 | 0.486 | 0.319 | 0.552 | 0.441 | 0.418 |
| NLI | 0.374 | 0.354 | 0.438 | 0.348 | 0.600 | 0.446 | 0.409 |
| CrossEncoder | 0.380 | 0.444 | 0.514 | 0.339 | 0.593 | 0.447 | 0.429 |
| $CoCoA_{MTE}$ | | | | | | | |
| AlignScore | 0.419 | 0.340 | 0.438 | 0.339 | 0.605 | 0.411 | **0.459** |
| RougeL | 0.362 | 0.417 | 0.481 | 0.319 | 0.560 | 0.390 | 0.440 |
| NLI | 0.366 | 0.342 | 0.428 | 0.340 | **0.612** | 0.396 | 0.420 |
| CrossEncoder | 0.374 | 0.441 | 0.511 | 0.337 | 0.601 | 0.394 | 0.444 |
| **Falcon7b-Base** | | | | | | | |
| $CoCoA_{SP}$ | | | | | | | |
| AlignScore | **0.278** | 0.306 | 0.475 | 0.361 | 0.677 | 0.528 | 0.470 |
| RougeL | 0.205 | 0.394 | 0.499 | 0.378 | 0.678 | 0.527 | 0.417 |
| NLI | 0.236 | 0.361 | 0.511 | 0.407 | 0.684 | **0.532** | 0.532 |
| CrossEncoder | 0.253 | 0.436 | 0.577 | 0.396 | 0.685 | 0.529 | 0.428 |
| $CoCoA_{PPL}$ | | | | | | | |
| AlignScore | 0.252 | 0.340 | 0.523 | **0.410** | 0.678 | 0.528 | 0.521 |
| RougeL | 0.170 | 0.409 | 0.537 | 0.389 | 0.668 | 0.527 | 0.439 |
| NLI | 0.193 | 0.364 | 0.531 | 0.408 | 0.680 | **0.532** | 0.499 |
| CrossEncoder | 0.226 | 0.437 | **0.579** | 0.405 | 0.677 | 0.529 | 0.474 |
| $CoCoA_{MTE}$ | | | | | | | |
| AlignScore | 0.253 | 0.337 | 0.519 | 0.403 | 0.683 | 0.515 | **0.554** |
| RougeL | 0.170 | 0.426 | 0.540 | 0.382 | 0.673 | 0.514 | 0.472 |
| NLI | 0.190 | 0.364 | 0.525 | 0.398 | **0.687** | 0.521 | 0.514 |
| CrossEncoder | 0.223 | **0.438** | 0.575 | 0.395 | 0.685 | 0.517 | 0.505 |

Table 5: Comparison of PRRs of CoCoA-family methods with different choices of the similarity function. Main model response obtained by greedy decoding.

| Method | Dataset | | | | | | |
|---|---|---|---|---|---|---|---|
| | XSUM | WMT14FrEn | WMT19DeEn | CoQA | TriviaQA | MMLU | GSM8k |
| Mistral7b-Base | | | | | | | |
| CoCoA$_{MSP}$ | | | | | | | |
| AlignScore | 0.393 | 0.448 | 0.491 | 0.399 | 0.626 | 0.467 | 0.476 |
| RougeL | 0.344 | 0.602 | 0.597 | 0.420 | 0.622 | 0.466 | 0.538 |
| NLI | 0.340 | 0.615 | 0.604 | **0.445** | 0.651 | **0.470** | 0.456 |
| CrossEncoder | 0.366 | 0.712 | 0.730 | 0.430 | 0.644 | 0.466 | 0.562 |
| CoCoA$_{PPL}$ | | | | | | | |
| AlignScore | 0.474 | 0.619 | 0.657 | 0.408 | 0.638 | 0.467 | 0.910 |
| RougeL | 0.362 | 0.710 | 0.717 | 0.391 | 0.627 | 0.466 | 0.950 |
| NLI | 0.370 | 0.677 | 0.684 | 0.414 | **0.657** | **0.470** | 0.941 |
| CrossEncoder | 0.372 | **0.735** | **0.755** | 0.402 | 0.648 | 0.466 | 0.937 |
| CoCoA$_{MTE}$ | | | | | | | |
| AlignScore | **0.492** | 0.547 | 0.590 | 0.383 | 0.633 | 0.449 | 0.914 |
| RougeL | 0.355 | 0.695 | 0.684 | 0.366 | 0.624 | 0.448 | **0.959** |
| NLI | 0.364 | 0.656 | 0.658 | 0.387 | 0.656 | 0.453 | 0.918 |
| CrossEncoder | 0.373 | 0.708 | 0.732 | 0.373 | 0.645 | 0.447 | 0.935 |
| Llama8b-Base | | | | | | | |
| CoCoA$_{MSP}$ | | | | | | | |
| AlignScore | 0.520 | 0.332 | 0.491 | 0.354 | 0.587 | **0.457** | 0.401 |
| RougeL | 0.471 | 0.470 | 0.588 | 0.362 | 0.551 | 0.446 | 0.499 |
| NLI | 0.466 | 0.442 | 0.577 | **0.386** | 0.597 | 0.446 | 0.470 |
| CrossEncoder | 0.484 | 0.529 | 0.685 | 0.384 | 0.587 | 0.452 | 0.513 |
| CoCoA$_{PPL}$ | | | | | | | |
| AlignScore | 0.546 | 0.406 | 0.561 | 0.376 | 0.577 | 0.429 | 0.875 |
| RougeL | 0.452 | 0.518 | 0.639 | 0.352 | 0.532 | 0.417 | **0.931** |
| NLI | 0.458 | 0.466 | 0.597 | 0.365 | 0.583 | 0.418 | 0.912 |
| CrossEncoder | 0.450 | **0.544** | **0.689** | 0.364 | 0.573 | 0.422 | 0.925 |
| CoCoA$_{MTE}$ | | | | | | | |
| AlignScore | **0.561** | 0.325 | 0.497 | 0.365 | 0.589 | 0.380 | 0.821 |
| RougeL | 0.448 | 0.496 | 0.598 | 0.336 | 0.539 | 0.361 | 0.921 |
| NLI | 0.449 | 0.446 | 0.565 | 0.344 | **0.598** | 0.359 | 0.881 |
| CrossEncoder | 0.451 | 0.520 | 0.638 | 0.346 | 0.582 | 0.363 | 0.900 |
| Falcon7b-Base | | | | | | | |
| CoCoA$_{MSP}$ | | | | | | | |
| AlignScore | 0.181 | 0.378 | 0.473 | 0.410 | 0.654 | 0.528 | 0.239 |
| RougeL | 0.122 | 0.531 | 0.581 | 0.420 | 0.655 | 0.528 | 0.426 |
| NLI | 0.120 | 0.496 | 0.607 | 0.437 | 0.658 | **0.533** | 0.458 |
| CrossEncoder | 0.210 | 0.564 | 0.698 | 0.428 | **0.659** | 0.530 | 0.498 |
| CoCoA$_{PPL}$ | | | | | | | |
| AlignScore | **0.384** | 0.454 | 0.586 | **0.440** | 0.648 | 0.528 | 0.994 |
| RougeL | 0.280 | 0.565 | 0.668 | 0.410 | 0.637 | 0.528 | **1.000** |
| NLI | 0.283 | 0.515 | 0.671 | 0.424 | 0.647 | **0.533** | 0.998 |
| CrossEncoder | 0.310 | **0.579** | **0.717** | 0.415 | 0.644 | 0.530 | **1.000** |
| CoCoA$_{MTE}$ | | | | | | | |
| AlignScore | 0.292 | 0.386 | 0.498 | 0.435 | 0.648 | 0.515 | 0.972 |
| RougeL | 0.222 | 0.545 | 0.607 | 0.400 | 0.633 | 0.515 | 0.998 |
| NLI | 0.201 | 0.498 | 0.636 | 0.415 | 0.645 | 0.521 | 0.987 |
| CrossEncoder | 0.289 | 0.551 | 0.678 | 0.402 | 0.646 | 0.517 | 0.998 |

Table 6: Comparison of PRRs of CoCoA-family methods with different choices of similarity function. Main model response obtained by selecting the most probable sample.

| Method | Dataset | | | | | | |
|---|---|---|---|---|---|---|---|
| | XSUM | WMT14FrEn | WMT19DeEn | CoQA | TriviaQA | MMLU | GSM8k |
| | | | | Mistral7b-Base | | | |
| | | | | $CoCoA_{MSP}$ | | | |
| AlignScore | 0.287 | 0.326 | 0.460 | 0.312 | 0.575 | **0.515** | 0.747 |
| RougeL | 0.259 | 0.420 | 0.562 | 0.304 | 0.502 | **0.515** | 0.750 |
| NLI | 0.278 | 0.375 | 0.506 | 0.350 | 0.589 | 0.512 | 0.766 |
| CrossEncoder | **0.295** | 0.441 | **0.589** | 0.336 | 0.597 | 0.514 | 0.766 |
| | | | | $CoCoA_{PPL}$ | | | |
| AlignScore | 0.283 | 0.333 | 0.482 | 0.330 | 0.596 | **0.515** | 0.738 |
| RougeL | 0.219 | 0.427 | 0.561 | 0.300 | 0.517 | **0.515** | 0.744 |
| NLI | 0.219 | 0.390 | 0.508 | 0.334 | 0.602 | 0.512 | 0.723 |
| CrossEncoder | 0.258 | **0.451** | 0.577 | 0.322 | 0.612 | 0.514 | 0.768 |
| | | | | $CoCoA_{MTE}$ | | | |
| AlignScore | 0.269 | 0.314 | 0.447 | **0.357** | 0.663 | 0.497 | 0.746 |
| RougeL | 0.181 | 0.446 | 0.571 | 0.327 | 0.622 | 0.497 | **0.786** |
| NLI | 0.182 | 0.370 | 0.482 | 0.348 | **0.686** | 0.499 | 0.704 |
| CrossEncoder | 0.236 | 0.419 | 0.553 | 0.337 | 0.680 | 0.494 | 0.783 |
| | | | | Llama8b-Base | | | |
| | | | | $CoCoA_{MSP}$ | | | |
| AlignScore | 0.221 | 0.309 | 0.462 | 0.261 | 0.467 | 0.622 | 0.717 |
| RougeL | 0.195 | 0.401 | 0.542 | 0.250 | 0.361 | **0.631** | 0.721 |
| NLI | 0.222 | 0.311 | 0.463 | 0.311 | 0.480 | 0.580 | 0.769 |
| CrossEncoder | **0.225** | 0.403 | **0.590** | 0.304 | 0.487 | 0.586 | 0.731 |
| | | | | $CoCoA_{PPL}$ | | | |
| AlignScore | **0.225** | 0.325 | 0.484 | 0.304 | 0.476 | 0.603 | 0.747 |
| RougeL | 0.169 | 0.416 | 0.537 | 0.274 | 0.370 | 0.612 | 0.758 |
| NLI | 0.180 | 0.334 | 0.460 | 0.319 | 0.488 | 0.558 | 0.761 |
| CrossEncoder | 0.198 | 0.428 | 0.580 | 0.320 | 0.496 | 0.562 | 0.751 |
| | | | | $CoCoA_{MTE}$ | | | |
| AlignScore | 0.209 | 0.293 | 0.460 | 0.320 | 0.563 | 0.556 | 0.749 |
| RougeL | 0.148 | **0.436** | 0.559 | 0.314 | 0.497 | 0.587 | **0.785** |
| NLI | 0.156 | 0.298 | 0.445 | 0.340 | **0.592** | 0.460 | 0.740 |
| CrossEncoder | 0.189 | 0.398 | 0.561 | **0.345** | 0.587 | 0.472 | 0.765 |
| | | | | Falcon7b-Base | | | |
| | | | | $CoCoA_{MSP}$ | | | |
| AlignScore | **0.279** | 0.307 | 0.473 | 0.312 | 0.594 | 0.574 | 0.460 |
| RougeL | 0.202 | 0.424 | 0.487 | 0.307 | 0.566 | 0.574 | 0.447 |
| NLI | 0.231 | 0.362 | 0.508 | 0.355 | 0.618 | 0.577 | 0.531 |
| CrossEncoder | 0.248 | 0.461 | 0.590 | 0.336 | 0.617 | 0.575 | 0.472 |
| | | | | $CoCoA_{PPL}$ | | | |
| AlignScore | 0.278 | 0.320 | 0.534 | 0.350 | 0.607 | 0.574 | 0.617 |
| RougeL | 0.190 | 0.428 | 0.556 | 0.333 | 0.571 | 0.574 | 0.632 |
| NLI | 0.216 | 0.376 | 0.549 | 0.368 | 0.623 | **0.578** | 0.661 |
| CrossEncoder | 0.242 | 0.462 | **0.613** | 0.345 | 0.624 | 0.575 | **0.662** |
| | | | | $CoCoA_{MTE}$ | | | |
| AlignScore | 0.277 | 0.302 | 0.522 | 0.382 | 0.657 | 0.556 | 0.598 |
| RougeL | 0.175 | **0.472** | 0.573 | 0.364 | 0.642 | 0.556 | 0.617 |
| NLI | 0.196 | 0.359 | 0.530 | **0.400** | **0.671** | 0.562 | 0.639 |
| CrossEncoder | 0.237 | 0.454 | 0.600 | 0.373 | 0.666 | 0.557 | 0.650 |

Table 7: Comparison of PRRs of CoCoA-family methods with different choices of similarity function. Main model response obtained by MBR decoding.

## C.2 Different Ways of Combining Confidence and Consistency

We justify the particular form of equation (11) by considering alternative ways to combine sample-focused confidence with consistency estimation. Results are presented in Tables 8, 9 and 10. In particular, we investigate the performance of the additive approach (AdditiveCoCoA):

$$U_{\text{AdditiveCoCoA}}(\mathbf{y}_* \mid \mathbf{x}) = u(\mathbf{y}_* \mid \mathbf{x}) + \widehat{U}_{\text{cons}}(\mathbf{y}_* \mid \mathbf{x}), \tag{16}$$

and the same multiplicative combination, replacing sample-focused dissimilarity from (8) with the average of the full pairwise dissimilarity matrix (24):

$$U_{\text{FullSampleCoCoA}}(\mathbf{y}_* \mid \mathbf{x}) = u(\mathbf{y}_* \mid \mathbf{x}) \cdot U_{\text{DegMat}}(\mathbf{x}). \tag{17}$$

In addition, we present results where we compute $u(\mathbf{y}_* \mid \mathbf{x})$ using probability-based scores instead of log-likelihood, see equation (6). We refer to this variant as $U_{\text{ProbCoCoA}}(\mathbf{y}_* \mid \mathbf{x})$. While the results are generally similar, the log-likelihood-based formulation offers greater numerical stability. Therefore, we adopt it as the default in the main paper.

It is evident that on average the multiplicative form proposed in equation (10) with both confidence and consistency terms focused on a single sample is the better performing variant.

| Method | Dataset | | | | | | |
|---|---|---|---|---|---|---|---|
| | XSUM | WMT14FrEn | WMT19DeEn | CoQA | TriviaQA | MMLU | GSM8k |
| | | | | Mistral7b-Base | | | |
| AdditiveCoCoA$_{SP}$ | 0.290 | 0.319 | 0.459 | 0.351 | 0.654 | **0.471** | 0.472 |
| FullSampleCoCoA$_{SP}$ | 0.319 | 0.385 | 0.590 | 0.357 | 0.668 | 0.467 | 0.505 |
| ProbCoCoA$_{SP}$ | 0.059 | 0.302 | 0.520 | **0.390** | 0.671 | 0.461 | 0.435 |
| CoCoA$_{SP}$ | **0.330** | **0.396** | **0.598** | 0.383 | 0.670 | 0.466 | **0.517** |
| AdditiveCoCoA$_{PPL}$ | 0.262 | 0.392 | 0.564 | 0.369 | 0.671 | 0.464 | 0.494 |
| FullSampleCoCoA$_{PPL}$ | 0.277 | 0.373 | 0.551 | 0.334 | 0.672 | 0.467 | 0.435 |
| ProbCoCoA$_{PPL}$ | 0.297 | 0.369 | 0.566 | 0.373 | **0.674** | 0.464 | 0.475 |
| CoCoA$_{PPL}$ | 0.286 | 0.375 | 0.568 | 0.369 | **0.674** | 0.466 | 0.467 |
| AdditiveCoCoA$_{MTE}$ | -0.279 | -0.058 | -0.072 | 0.098 | 0.312 | 0.079 | 0.187 |
| FullSampleCoCoA$_{MTE}$ | 0.274 | 0.368 | 0.543 | 0.309 | 0.668 | 0.442 | 0.456 |
| CoCoA$_{MTE}$ | 0.288 | 0.374 | 0.564 | 0.355 | 0.673 | 0.447 | 0.491 |
| | | | | Llama8b-Base | | | |
| AdditiveCoCoA$_{SP}$ | 0.330 | 0.345 | 0.462 | 0.301 | 0.566 | **0.502** | 0.326 |
| FullSampleCoCoA$_{SP}$ | 0.358 | 0.434 | 0.564 | 0.333 | 0.589 | 0.488 | 0.354 |
| ProbCoCoA$_{SP}$ | 0.031 | 0.405 | 0.471 | **0.371** | **0.612** | 0.461 | 0.368 |
| CoCoA$_{SP}$ | 0.378 | **0.456** | **0.582** | 0.349 | 0.597 | 0.485 | 0.372 |
| AdditiveCoCoA$_{PPL}$ | 0.368 | 0.431 | 0.504 | 0.336 | 0.595 | 0.455 | 0.437 |
| FullSampleCoCoA$_{PPL}$ | **0.389** | 0.420 | 0.487 | 0.314 | 0.580 | 0.450 | 0.399 |
| ProbCoCoA$_{PPL}$ | 0.381 | 0.445 | 0.513 | 0.345 | 0.599 | 0.446 | 0.438 |
| CoCoA$_{PPL}$ | 0.387 | 0.448 | 0.514 | 0.338 | 0.593 | 0.452 | 0.433 |
| AdditiveCoCoA$_{MTE}$ | -0.331 | -0.042 | -0.122 | 0.089 | 0.321 | -0.122 | 0.117 |
| FullSampleCoCoA$_{MTE}$ | 0.383 | 0.410 | 0.481 | 0.308 | 0.588 | 0.363 | 0.414 |
| CoCoA$_{MTE}$ | 0.380 | 0.446 | 0.511 | 0.337 | 0.601 | 0.402 | **0.447** |
| | | | | Falcon7b-Base | | | |
| AdditiveCoCoA$_{SP}$ | 0.203 | 0.318 | 0.409 | 0.350 | 0.674 | **0.533** | 0.379 |
| FullSampleCoCoA$_{SP}$ | 0.225 | 0.423 | 0.571 | 0.388 | 0.678 | **0.533** | 0.404 |
| ProbCoCoA$_{SP}$ | 0.226 | 0.367 | 0.515 | **0.416** | 0.680 | 0.526 | 0.426 |
| CoCoA$_{SP}$ | **0.257** | 0.433 | 0.578 | 0.396 | 0.684 | 0.529 | 0.436 |
| AdditiveCoCoA$_{PPL}$ | 0.222 | 0.433 | **0.580** | 0.413 | 0.677 | 0.525 | 0.489 |
| FullSampleCoCoA$_{PPL}$ | 0.204 | 0.425 | 0.565 | 0.393 | 0.669 | **0.533** | 0.437 |
| ProbCoCoA$_{PPL}$ | 0.235 | 0.433 | 0.576 | 0.410 | 0.680 | 0.528 | 0.482 |
| CoCoA$_{PPL}$ | 0.229 | 0.436 | **0.580** | 0.406 | 0.677 | 0.529 | 0.478 |
| AdditiveCoCoA$_{MTE}$ | 0.001 | -0.103 | -0.106 | 0.114 | 0.041 | 0.138 | 0.221 |
| FullSampleCoCoA$_{MTE}$ | 0.201 | 0.425 | 0.557 | 0.377 | 0.675 | 0.519 | 0.470 |
| CoCoA$_{MTE}$ | 0.228 | **0.439** | 0.577 | 0.395 | **0.685** | 0.517 | **0.510** |

Table 8: Comparison of PRRs of CoCoA-family methods with alternative formulations. Main model response obtained via greedy decoding.

| Method | Dataset | | | | | | |
|---|---|---|---|---|---|---|---|
| | XSUM | WMT14FrEn | WMT19DeEn | CoQA | TriviaQA | MMLU | GSM8k |
| Mistral7b-Base | | | | | | | |
| AdditiveCoCoA$_{SP}$ | 0.333 | 0.239 | 0.310 | 0.406 | 0.631 | **0.472** | 0.311 |
| FullSampleCoCoA$_{SP}$ | 0.354 | 0.543 | 0.565 | 0.412 | 0.643 | 0.468 | 0.428 |
| ProbCoCoA$_{SP}$ | 0.076 | 0.684 | 0.721 | 0.428 | 0.643 | 0.464 | 0.846 |
| CoCoA$_{SP}$ | 0.366 | 0.712 | 0.730 | **0.430** | 0.644 | 0.466 | 0.562 |
| AdditiveCoCoA$_{PPL}$ | 0.368 | 0.737 | 0.751 | 0.406 | 0.644 | 0.465 | 0.939 |
| FullSampleCoCoA$_{PPL}$ | **0.383** | 0.714 | 0.723 | 0.379 | **0.649** | 0.468 | 0.933 |
| ProbCoCoA$_{PPL}$ | 0.369 | **0.738** | **0.756** | 0.401 | **0.649** | 0.467 | 0.935 |
| CoCoA$_{PPL}$ | 0.372 | 0.735 | 0.755 | 0.402 | 0.648 | 0.466 | 0.937 |
| AdditiveCoCoA$_{MTE}$ | 0.368 | 0.723 | 0.702 | 0.332 | 0.643 | 0.452 | **0.942** |
| FullSampleCoCoA$_{MTE}$ | 0.380 | 0.661 | 0.653 | 0.331 | 0.643 | 0.442 | 0.929 |
| CoCoA$_{MTE}$ | 0.373 | 0.708 | 0.732 | 0.373 | 0.645 | 0.447 | 0.935 |
| Llama8b-Base | | | | | | | |
| AdditiveCoCoA$_{SP}$ | 0.466 | 0.349 | 0.425 | 0.333 | 0.555 | **0.473** | 0.285 |
| FullSampleCoCoA$_{SP}$ | 0.476 | 0.462 | 0.619 | 0.363 | 0.574 | 0.464 | 0.379 |
| ProbCoCoA$_{SP}$ | 0.035 | 0.491 | 0.617 | **0.398** | **0.598** | 0.433 | 0.795 |
| CoCoA$_{SP}$ | **0.484** | 0.529 | 0.685 | 0.384 | 0.587 | 0.452 | 0.513 |
| AdditiveCoCoA$_{PPL}$ | 0.454 | 0.536 | 0.673 | 0.358 | 0.575 | 0.425 | 0.923 |
| FullSampleCoCoA$_{PPL}$ | 0.459 | 0.525 | 0.649 | 0.343 | 0.556 | 0.430 | 0.914 |
| ProbCoCoA$_{PPL}$ | 0.438 | **0.547** | **0.689** | 0.364 | 0.574 | 0.419 | 0.923 |
| CoCoA$_{PPL}$ | 0.450 | 0.544 | **0.689** | 0.364 | 0.573 | 0.422 | **0.925** |
| AdditiveCoCoA$_{MTE}$ | 0.457 | 0.496 | 0.579 | 0.304 | 0.561 | 0.361 | 0.901 |
| FullSampleCoCoA$_{MTE}$ | 0.455 | 0.464 | 0.577 | 0.313 | 0.563 | 0.341 | 0.878 |
| CoCoA$_{MTE}$ | 0.451 | 0.520 | 0.638 | 0.346 | 0.582 | 0.363 | 0.900 |
| Falcon7b-Base | | | | | | | |
| AdditiveCoCoA$_{SP}$ | 0.100 | 0.397 | 0.394 | 0.393 | 0.649 | **0.534** | -0.156 |
| FullSampleCoCoA$_{SP}$ | 0.144 | 0.531 | 0.607 | 0.416 | 0.654 | 0.533 | 0.189 |
| ProbCoCoA$_{SP}$ | 0.282 | 0.522 | 0.670 | **0.434** | 0.658 | 0.529 | 0.978 |
| CoCoA$_{SP}$ | 0.210 | 0.564 | 0.698 | 0.428 | **0.659** | 0.530 | 0.498 |
| AdditiveCoCoA$_{PPL}$ | 0.297 | 0.582 | 0.706 | 0.417 | 0.643 | 0.526 | **1.000** |
| FullSampleCoCoA$_{PPL}$ | 0.297 | 0.560 | 0.670 | 0.405 | 0.641 | 0.533 | **1.000** |
| ProbCoCoA$_{PPL}$ | **0.311** | **0.587** | **0.718** | 0.414 | 0.648 | 0.531 | **1.000** |
| CoCoA$_{PPL}$ | 0.310 | 0.579 | 0.717 | 0.415 | 0.644 | 0.530 | **1.000** |
| AdditiveCoCoA$_{MTE}$ | 0.253 | 0.554 | 0.634 | 0.383 | 0.630 | 0.523 | 0.997 |
| FullSampleCoCoA$_{MTE}$ | 0.237 | 0.502 | 0.554 | 0.383 | 0.636 | 0.519 | 0.989 |
| CoCoA$_{MTE}$ | 0.289 | 0.551 | 0.678 | 0.402 | 0.646 | 0.517 | 0.998 |

Table 9: Comparison of PRRs of CoCoA-family methods with alternative formulations. Main model response obtained by selecting the most probable sample.

| Method | Dataset | | | | | | |
|---|---|---|---|---|---|---|---|
| | XSUM | WMT14FrEn | WMT19DeEn | CoQA | TriviaQA | MMLU | GSM8k |
| *Mistral7b-Base* | | | | | | | |
| AdditiveCoCoA$_{SP}$ | 0.259 | 0.382 | 0.488 | 0.231 | 0.388 | 0.512 | 0.711 |
| FullSampleCoCoA$_{SP}$ | 0.287 | 0.447 | **0.591** | 0.318 | 0.570 | **0.516** | 0.748 |
| ProbCoCoA$_{SP}$ | 0.258 | 0.374 | 0.473 | 0.149 | 0.099 | -0.265 | 0.710 |
| CoCoA$_{MTE}$ | **0.295** | 0.441 | 0.589 | 0.336 | 0.597 | 0.514 | 0.766 |
| AdditiveCoCoA$_{PPL}$ | 0.234 | 0.430 | 0.560 | 0.255 | 0.517 | 0.507 | 0.721 |
| FullSampleCoCoA$_{PPL}$ | 0.251 | 0.458 | 0.571 | 0.305 | 0.587 | **0.516** | 0.745 |
| ProbCoCoA$_{PPL}$ | 0.094 | -0.123 | -0.143 | -0.202 | -0.404 | -0.423 | 0.027 |
| CoCoA$_{PPL}$ | 0.258 | 0.451 | 0.577 | 0.322 | 0.612 | 0.514 | 0.768 |
| AdditiveCoCoA$_{MTE}$ | 0.201 | **0.468** | 0.585 | 0.296 | 0.669 | 0.501 | 0.760 |
| FullSampleCoCoA$_{MTE}$ | 0.232 | 0.426 | 0.544 | 0.318 | 0.673 | 0.496 | 0.772 |
| CoCoA$_{MTE}$ | 0.236 | 0.419 | 0.553 | **0.337** | **0.680** | 0.494 | **0.783** |
| *Llama8b-Base* | | | | | | | |
| AdditiveCoCoA$_{SP}$ | 0.197 | 0.345 | 0.465 | 0.177 | 0.287 | 0.571 | 0.676 |
| FullSampleCoCoA$_{SP}$ | 0.214 | 0.409 | 0.589 | 0.289 | 0.449 | **0.589** | 0.724 |
| ProbCoCoA$_{SP}$ | 0.196 | 0.337 | 0.449 | 0.102 | 0.053 | 0.434 | 0.673 |
| CoCoA$_{MTE}$ | **0.225** | 0.403 | **0.590** | 0.304 | 0.487 | 0.586 | 0.731 |
| AdditiveCoCoA$_{PPL}$ | 0.182 | 0.406 | 0.500 | 0.229 | 0.401 | 0.540 | 0.715 |
| FullSampleCoCoA$_{PPL}$ | 0.186 | 0.433 | 0.563 | 0.304 | 0.461 | 0.565 | 0.746 |
| ProbCoCoA$_{PPL}$ | 0.101 | -0.097 | -0.152 | -0.175 | -0.367 | 0.099 | 0.053 |
| CoCoA$_{PPL}$ | 0.198 | 0.428 | 0.580 | 0.320 | 0.496 | 0.562 | 0.751 |
| AdditiveCoCoA$_{MTE}$ | 0.160 | **0.437** | 0.554 | 0.302 | 0.560 | 0.477 | 0.748 |
| FullSampleCoCoA$_{MTE}$ | 0.174 | 0.400 | 0.543 | 0.334 | 0.568 | 0.475 | 0.762 |
| CoCoA$_{MTE}$ | 0.189 | 0.398 | 0.561 | **0.345** | **0.587** | 0.472 | **0.765** |
| *Falcon7b-Base* | | | | | | | |
| AdditiveCoCoA$_{SP}$ | 0.188 | 0.356 | 0.401 | 0.228 | 0.488 | **0.578** | 0.368 |
| FullSampleCoCoA$_{SP}$ | 0.237 | 0.468 | 0.586 | 0.325 | 0.593 | 0.576 | 0.453 |
| ProbCoCoA$_{SP}$ | 0.184 | 0.339 | 0.378 | 0.144 | 0.350 | -0.345 | 0.364 |
| CoCoA$_{MTE}$ | **0.248** | 0.461 | 0.590 | 0.336 | 0.617 | 0.575 | 0.472 |
| AdditiveCoCoA$_{PPL}$ | 0.206 | 0.404 | 0.565 | 0.271 | 0.537 | 0.572 | 0.625 |
| FullSampleCoCoA$_{PPL}$ | 0.231 | 0.464 | 0.606 | 0.337 | 0.598 | 0.577 | 0.650 |
| ProbCoCoA$_{PPL}$ | 0.014 | -0.167 | -0.265 | -0.187 | -0.237 | -0.416 | -0.206 |
| CoCoA$_{PPL}$ | 0.242 | 0.462 | **0.613** | 0.345 | 0.624 | 0.575 | **0.662** |
| AdditiveCoCoA$_{MTE}$ | 0.191 | **0.488** | 0.612 | 0.353 | 0.644 | 0.564 | 0.617 |
| FullSampleCoCoA$_{MTE}$ | 0.219 | 0.452 | 0.591 | 0.368 | 0.654 | 0.556 | 0.629 |
| CoCoA$_{MTE}$ | 0.237 | 0.454 | 0.600 | **0.373** | **0.666** | 0.557 | 0.650 |

Table 10: Comparison of PRRs of CoCoA-family methods with alternative formulations. Main model response obtained by MBR decoding.

# D  Detailed Description of Uncertainty Quantification Methods

In this section, we provide a detailed description of the uncertainty quantification methods used in this study.

## D.1  Information-Based Methods

Information-based methods are commonly used to estimate uncertainty by analyzing the probability distributions of tokens within a given output. These methods examine different levels of model generation, such as the model's confidence in producing a specific sequence, its ability to predict individual tokens at each generation step, and the variability in the token-level predictions across the sequence.

*Sequence Probability (SP)* is one of the simplest and most direct methods for estimating uncertainty. It measures the probability of the most likely output sequence given a specific input. Thus, uncertainty is quantified by calculating the probability of the sequence with the highest likelihood, under the assumption that the model is most confident in this output. In its simplest form it is given by (6). An equivalent (in terms of ordering of predictions for different inputs) but more numerically stable formulation is logarithmic. It is defined as:

$$U_{\text{SP}}(\mathbf{y} \mid \mathbf{x}) = -\log p(\mathbf{y} \mid \mathbf{x}). \tag{18}$$

*Perplexity (PPL)* is another widely used metric for estimating uncertainty in language models [Fomicheva et al., 2020]. It measures the model's confidence by evaluating the average likelihood of generating the sequence tokens:

$$U_{\text{PPL}}(\mathbf{y} \mid \mathbf{x}) = -\frac{1}{L}\log p(\mathbf{y} \mid \mathbf{x}). \tag{19}$$

*Mean Token Entropy* takes a broader view of uncertainty by considering the token-level predictions across the entire sequence [Fomicheva et al., 2020]. Instead of evaluating the model's confidence in a single output or individual token predictions, Mean Token Entropy calculates the average entropy of the token probability distributions for each token in the sequence:

$$U_{\mathcal{H}_T}(\mathbf{y} \mid \mathbf{x}) = \frac{1}{L}\sum_{l=1}^{L}\mathcal{H}(y_l \mid \mathbf{y}_{<l}, \mathbf{x}), \tag{20}$$

where $\mathcal{H}(y_l \mid \mathbf{y}_{<l}, \mathbf{x})$ is an entropy of the token distribution $p(y_l \mid \mathbf{y}_{<l}, \mathbf{x})$.

The *TokenSAR* method, introduced in [Duan et al., 2024], generalizes length-normalized log probability by computing a weighted average of the negative log probabilities of generated tokens, where weights are based on token relevance to the overall text. Using a similarity function $s(\cdot, \cdot)$ and token relevance function $R_T(y_k, \mathbf{y}, \mathbf{x}) = 1 - s(\mathbf{x} \cup \mathbf{y}, \mathbf{x} \cup \mathbf{y} \setminus y_k)$, the uncertainty estimate is calculated as:

$$U_{\text{TokenSAR}}(\mathbf{y} \mid \mathbf{x}) = -\sum_{l=1}^{L}\tilde{R}_T(y_l, \mathbf{y}, \mathbf{x})\log p(y_l \mid \mathbf{y}_{<l}, \mathbf{x}), \tag{21}$$

where

$$\tilde{R}_T(y_k, \mathbf{y} \mid \mathbf{x}) = \frac{R_T(y_k, \mathbf{y}, \mathbf{x})}{\sum_{l=1}^{L}R_T(y_l, \mathbf{y}, \mathbf{x})}. \tag{22}$$

This measure is central for computing *SAR* uncertainty measure.

## D.2  Consistency-Based Methods

Consistency-based methods assess the uncertainty of a language model by evaluating the semantic consistency of its predictions across multiple outputs for the same prompt. The core idea is that semantically similar outputs indicate higher confidence, while diverse or conflicting outputs suggest greater uncertainty. Since language models can express the same meaning in different surface forms, these methods construct a semantic similarity matrix $S = (s_{ij})$, where each entry represents the degree of similarity between pairs of responses. By clustering responses into groups with equivalent meanings, these methods provide a semantic measure of the model's consistency.

The work [Lin et al., 2024] offers two similarity measures to evaluate the similarity of sequences. The first is the Jaccard similarity, which treats sequences as sets of words and calculates the proportion of shared words to the total number of unique words in both sequences: $s(\mathbf{y}, \mathbf{y}') = |\mathbf{y} \cap \mathbf{y}'| / |\mathbf{y} \cup \mathbf{y}'|$.

Natural Language Inference (NLI) provides another method for computing similarity between sequences. We use the DeBERTa-large NLI model [He et al., 2021], following [Kuhn et al., 2023]. For each pair of sequences, an NLI model predicts two probabilities: $p_{\text{entail}}(\mathbf{y}, \mathbf{y}')$, indicating entailment, and $p_{\text{contra}}(\mathbf{y}, \mathbf{y}')$, indicating contradiction. Similarity is then defined as either $s_{\text{entail}}(\mathbf{y}, \mathbf{y}') = p_{\text{entail}}(\mathbf{y}, \mathbf{y}')$ or $s_{\text{contra}}(\mathbf{y}, \mathbf{y}') = 1 - p_{\text{contra}}(\mathbf{y}, \mathbf{y}')$.

Among the simplest consistency-based approaches are the *Number of Semantic Sets* and the *Sum of Eigenvalues of the Graph Laplacian* [Lin et al., 2024]. *Number of Semantic Sets* estimates how many distinct "meanings" the model produces by clustering its outputs with an NLI model. The number of semantic sets is initially equal to the total number of generated answers, $M$. Two sentences are grouped into the same cluster if the following conditions are satisfied: $p_{\text{entail}}(\mathbf{y}^{(i)}, \mathbf{y}^{(j)}) > p_{\text{contra}}(\mathbf{y}^{(i)}, \mathbf{y}^{(j)})$ and $p_{\text{entail}}(\mathbf{y}^{(j)}, \mathbf{y}^{(i)}) > p_{\text{contra}}(\mathbf{y}^{(j)}, \mathbf{y}^{(i)})$. This computation is performed for all pairs of answers, and the final number of distinct clusters is denoted by $U_{\text{NumSemSets}}$.

*Sum of Eigenvalues of the Graph Laplacian* examines global diversity: it constructs a similarity matrix among the sampled outputs and computes a continuous uncertainty score from the eigenvalues of the Laplacian of that similarity graph. The work [Lin et al., 2024] proposes computing an averaged similarity matrix as $s_{ij} = \left(s(\mathbf{y}^{(i)}, \mathbf{y}^{(j)}) + s(\mathbf{y}^{(j)}, \mathbf{y}^{(i)})\right)/2$. The Laplacian for the matrix $S$ is defined as $L = I - D^{-\frac{1}{2}} S D^{-\frac{1}{2}}$, where $D$ is a (diagonal) degree matrix with elements $D_{ii} = \sum_{j=1}^{M} s_{ij}$. Consequently, the following formula is derived:

$$U_{\text{EigV}}(\mathbf{x}) = \sum_{i=1}^{M} \max(0, 1 - \lambda_i(\mathbf{x})). \tag{23}$$

Both *Number of Semantic Sets* and *Sum of Eigenvalues of the Graph Laplacian* effectively capture overall variation in generated text but cannot produce an individual uncertainty score for each output. To address this, the work [Lin et al., 2024] proposes to use the diagonal *Degree Matrix* $D(\mathbf{x})$ which represents the total similarity of each answer with all others. The corrected trace of $D(\mathbf{x})$ provides an average pairwise distance between answers, and uncertainty is computed as:

$$U_{\text{DegMat}}(\mathbf{x}) = 1 - \text{trace}(D(\mathbf{x}))/M^2. \tag{24}$$

### D.3  Information-Based Methods with Repeated Sampling

The natural idea is to somehow benefit from having multiple samples from the model while using important information contained in the output probabilities estimated by an LLM. Below, we examine several approaches that have sought to achieve this.

**Averaging uncertainties.**

We can compute the entropy on the sequence level $\mathbb{E}[-\log p(\mathbf{y} \mid \mathbf{x})]$, where the expectation is taken over the sequences $\mathbf{y}$ randomly generated from the distribution $p(\mathbf{y} \mid \mathbf{x})$. Unfortunately, while for token level, we have an exact way of computing the entropy, for the sequence level, we need to adhere to some approximations. In practice, we can use Monte-Carlo integration, i.e. generate several sequences $\mathbf{y}^{(i)}$, $i = 1, \ldots, M$ via random sampling and compute *Monte Carlo Sequence Entropy* [Kuhn et al., 2023]:

$$U_{\mathcal{H}_S}(\mathbf{x}) = -\frac{1}{M} \sum_{i=1}^{M} \log p(\mathbf{y}^{(i)} \mid \mathbf{x}). \tag{25}$$

We can replace $p(\mathbf{y}^{(i)} \mid \mathbf{x})$ with its length-normalized version $\bar{p}(\mathbf{y}^{(i)} \mid \mathbf{x})$ leading to a more reliable uncertainty measure in some cases.

While simple averaging represents a natural way to aggregate uncertainties, it has certain issues related to the nature of LLMs. First of all, in the vast majority of applications, an LLM-based system should produce a single output $\mathbf{y}_*$ for an input query. When we consider $U_{\mathcal{H}_S}(\mathbf{x})$ or other similar measure, we essentially perform averaging of uncertainties of different sequences, thus somewhat assessing the

uncertainty related to the entire generative distribution $p(\mathbf{y} \mid \mathbf{x})$ for the input $\mathbf{x}$, but not for a particular generated sequence $\mathbf{y}_*$. This averaged uncertainty might not be adequate for this particular sequence and, remarkably, often performs worse than the uncertainty $u_* = U(\mathbf{y}_* \mid \mathbf{x})$, which is related solely to the output $\mathbf{y}_*$. Moreover, although intuitive, this naïve aggregation method assumes that all outputs contribute equally to the final uncertainty estimate, regardless of their semantic relationships. This can lead to inconsistencies when semantically equivalent outputs have varying uncertainty scores or when outputs with low similarity are treated as equally important.

**Semantically weighted averaging.** *Semantic Entropy* [Kuhn et al., 2023] addresses the issue of generated sequences with similar meanings but differing probabilities according to the model, which can heavily influence the resulting entropy value (25). The method clusters generated sequences $\mathbf{y}^{(i)}$, $i = 1, \ldots, M$ into semantically homogeneous groups $\mathcal{C}_k$, $k = 1, \ldots, K$ (where $K \leq M$) using a bi-directional entailment algorithm. Probabilities of sequences are averaged within each cluster. The entropy estimate is then defined as:

$$U_{\text{SE}}(\mathbf{x}) = -\sum_{k=1}^{K} \frac{|\mathcal{C}_k|}{M} \log \hat{p}_k(\mathbf{x}), \tag{26}$$

where $\hat{p}_k(\mathbf{x}) = \sum_{\mathbf{y} \in \mathcal{C}_k} p(\mathbf{y} \mid \mathbf{x})$ represents the aggregated probability for cluster $\mathcal{C}_k$.

*SentenceSAR* [Duan et al., 2024] enhances the probability of sentences that are more relevant. It uses a sentence relevance measure $s(\mathbf{y}^{(j)}, \mathbf{y}^{(k)})$ to evaluate the relevance of $\mathbf{y}^{(j)}$ with respect to $\mathbf{y}^{(k)}$. SentenceSAR is calculated as:

$$U_{\text{SentSAR}}(\mathbf{x}) = -\frac{1}{M} \sum_{i=1}^{M} \log\Big(p(\mathbf{y}^{(i)} \mid \mathbf{x}) + \frac{1}{t} \mathrm{R}_S(\mathbf{y}^{(i)}, \mathbf{x})\Big), \tag{27}$$

where $t$ is a temperature parameter used to control the scale of shifting to relevance, and

$$\mathrm{R}_S(\mathbf{y}^{(j)}, \mathbf{x}) = \sum_{k \neq j} s(\mathbf{y}^{(j)}, \mathbf{y}^{(k)}) p(\mathbf{y}^{(k)} \mid \mathbf{x}). \tag{28}$$

The combination of SentenceSAR and TokenSAR results in a unified method called *SAR* [Duan et al., 2024]. In this approach, the generative probability $p(\mathbf{y} \mid \mathbf{x})$ in the SentenceSAR formula is replaced with the token-shifted probability $p'(\mathbf{y} \mid \mathbf{x}) = \exp\big\{-\text{TokenSAR}(\mathbf{y}, \mathbf{x})\big\}$, creating a comprehensive measure that integrates both sentence- and token-level adjustments.

The aggregation approaches like Semantic Entropy [Kuhn et al., 2023] or SAR [Duan et al., 2024] can be unified into a semantically-aware Generalized Monte Carlo uncertainty estimate, defined as

$$U_{\text{GMCU}}(\mathbf{x}) = \frac{1}{M} \sum_{i=1}^{M} h\left(\sum_{j=1}^{M} s(\mathbf{y}^{(i)}, \mathbf{y}^{(j)}) p(\mathbf{y}^{(j)} \mid \mathbf{x})\right). \tag{29}$$

Here, the inner summation aggregates sequence probabilities $p(\mathbf{y}^{(j)} \mid \mathbf{x})$ weighted by their semantic similarity to the $i$-th output, and the outer summation averages these contributions across all samples. The function $h(\cdot)$ provides an additional layer of flexibility, transforming the reweighted uncertainty scores, making the method a generalized framework for uncertainty quantification.

Unfortunately, methods that fall under GMCU, while offering benefits, also inherit the aforementioned issues from both categories of methods. In particular, the outer summation in (29), similarly to the case of simple Monte Carlo averaging, often fails to outperform the uncertainty $U_*(\mathbf{x}) = U(\mathbf{y}_* \mid \mathbf{x})$ of a single generated sequence $\mathbf{y}_*$.

### D.4 Verbalized Uncertainty

Verbalized uncertainty methods refer to approaches that prompt a model to explicitly express its confidence. In our experiment, we utilize the P(True) method, implemented following the description by [Kadavath et al., 2022]. Specifically, the model is presented with the original question and its answer, and then prompted to indicate whether the answer is True or False. We use the negative log-probability of the token "True" as the uncertainty score.

# E  Detailed Experimental Results

Tables 3 and 1 presented PRR scores averaged over datasets corresponding to each task. Here we present expanded results for each dataset, see Tables 11, 12 and 13.

| Method | Dataset | | | | | | |
|---|---|---|---|---|---|---|---|
| | XSUM | WMT14FrEn | WMT19DeEn | CoQA | TriviaQA | MMLU | GSM8k |
| Mistral7b-Base | | | | | | | |
| MCSE | 0.007 | 0.257 | 0.350 | 0.247 | 0.496 | 0.337 | 0.475 |
| MCNSE | 0.009 | 0.342 | 0.478 | 0.238 | 0.540 | 0.356 | 0.401 |
| Semantic Entropy | 0.008 | 0.271 | 0.382 | 0.271 | 0.562 | 0.387 | 0.472 |
| DegMat | 0.137 | 0.229 | 0.382 | 0.336 | 0.646 | 0.410 | 0.299 |
| EigValLaplacian | 0.132 | 0.207 | 0.328 | 0.301 | 0.624 | 0.398 | 0.241 |
| SAR | 0.094 | 0.353 | 0.517 | 0.313 | 0.644 | 0.419 | 0.471 |
| P(True) | 0.179 | 0.118 | 0.033 | -0.061 | -0.128 | 0.068 | 0.005 |
| Consistency Light | -0.006 | 0.317 | 0.457 | 0.306 | 0.597 | 0.432 | 0.441 |
| Consistency | 0.051 | 0.285 | 0.500 | 0.379 | 0.647 | 0.423 | 0.435 |
| SP | 0.287 | 0.315 | 0.451 | 0.326 | 0.628 | 0.474 | 0.471 |
| CoCoA$_{SP}$ | **0.330** | 0.396 | **0.598** | **0.383** | 0.670 | 0.466 | **0.517** |
| CoCoA$_{SP}$ Light | 0.309 | 0.380 | 0.580 | 0.351 | 0.674 | **0.480** | 0.506 |
| PPL | 0.204 | 0.365 | 0.489 | 0.281 | 0.632 | 0.474 | 0.311 |
| CoCoA$_{PPL}$ | 0.286 | 0.375 | 0.568 | 0.369 | 0.674 | 0.466 | 0.467 |
| CoCoA$_{PPL}$ Light | 0.260 | 0.404 | 0.557 | 0.326 | **0.684** | **0.480** | 0.428 |
| MTE | 0.182 | 0.392 | 0.484 | 0.243 | 0.619 | 0.456 | 0.350 |
| CoCoA$_{MTE}$ | 0.288 | 0.374 | 0.564 | 0.355 | 0.673 | 0.447 | 0.491 |
| CoCoA$_{MTE}$ Light | 0.254 | **0.415** | 0.550 | 0.303 | 0.672 | 0.465 | 0.458 |
| Llama8b-Base | | | | | | | |
| MCSE | 0.033 | 0.293 | 0.354 | 0.237 | 0.482 | 0.171 | 0.351 |
| MCNSE | 0.022 | 0.370 | 0.415 | 0.219 | 0.501 | 0.170 | 0.344 |
| Semantic Entropy | 0.033 | 0.297 | 0.389 | 0.272 | 0.549 | 0.229 | 0.375 |
| DegMat | 0.081 | 0.250 | 0.355 | 0.353 | **0.622** | 0.342 | 0.309 |
| EigValLaplacian | 0.079 | 0.198 | 0.278 | 0.332 | 0.604 | 0.292 | 0.273 |
| SAR | 0.077 | 0.427 | 0.483 | 0.311 | 0.595 | 0.352 | 0.398 |
| P(True) | 0.058 | 0.047 | 0.037 | -0.037 | -0.066 | -0.180 | 0.026 |
| Consistency Light | -0.022 | 0.475 | 0.441 | 0.353 | 0.552 | 0.279 | 0.378 |
| Consistency | 0.024 | 0.389 | 0.453 | **0.375** | 0.614 | 0.391 | 0.368 |
| SP | 0.328 | 0.342 | 0.456 | 0.277 | 0.526 | **0.508** | 0.324 |
| CoCoA$_{SP}$ | 0.378 | 0.456 | **0.582** | 0.349 | 0.597 | 0.485 | 0.372 |
| CoCoA$_{SP}$ Light | 0.358 | 0.448 | 0.556 | 0.340 | 0.598 | 0.504 | 0.353 |
| PPL | 0.369 | 0.351 | 0.422 | 0.253 | 0.507 | 0.461 | 0.303 |
| CoCoA$_{PPL}$ | **0.387** | 0.448 | 0.514 | 0.338 | 0.593 | 0.452 | 0.433 |
| CoCoA$_{PPL}$ Light | 0.382 | 0.483 | 0.491 | 0.319 | 0.597 | 0.467 | 0.398 |
| MTE | 0.357 | 0.357 | 0.408 | 0.239 | 0.497 | 0.349 | 0.326 |
| CoCoA$_{MTE}$ | 0.380 | 0.446 | 0.511 | 0.337 | 0.601 | 0.401 | **0.447** |
| CoCoA$_{MTE}$ Light | 0.372 | **0.501** | 0.487 | 0.316 | 0.599 | 0.387 | 0.412 |
| Falcon7b-Base | | | | | | | |
| MCSE | 0.159 | 0.297 | 0.337 | 0.258 | 0.549 | 0.420 | 0.427 |
| MCNSE | 0.108 | 0.371 | 0.474 | 0.293 | 0.586 | 0.442 | 0.299 |
| Semantic Entropy | 0.164 | 0.307 | 0.389 | 0.294 | 0.581 | 0.463 | 0.418 |
| DegMat | 0.201 | 0.274 | 0.431 | 0.407 | 0.651 | 0.480 | 0.395 |
| EigValLaplacian | 0.201 | 0.229 | 0.394 | 0.381 | 0.645 | 0.454 | 0.358 |
| SAR | 0.144 | 0.398 | 0.517 | 0.381 | 0.649 | 0.508 | 0.387 |
| P(True) | -0.159 | 0.175 | 0.135 | 0.036 | 0.275 | 0.027 | 0.133 |
| Consistency Light | 0.232 | 0.483 | 0.468 | 0.327 | 0.657 | 0.362 | 0.363 |
| Consistency | 0.226 | 0.337 | 0.496 | **0.408** | 0.656 | 0.485 | 0.426 |
| SP | 0.201 | 0.312 | 0.400 | 0.321 | 0.662 | **0.539** | 0.377 |
| CoCoA$_{SP}$ | **0.257** | 0.433 | 0.578 | 0.396 | 0.684 | 0.529 | 0.436 |
| CoCoA$_{SP}$ Light | 0.242 | 0.463 | 0.566 | 0.349 | 0.691 | 0.522 | 0.405 |
| PPL | 0.155 | 0.375 | 0.525 | 0.316 | 0.644 | **0.539** | 0.326 |
| CoCoA$_{PPL}$ | 0.229 | 0.436 | **0.580** | 0.406 | 0.677 | 0.529 | 0.478 |
| CoCoA$_{PPL}$ Light | 0.234 | 0.497 | 0.558 | 0.364 | **0.694** | 0.522 | 0.468 |
| MTE | 0.152 | 0.409 | 0.537 | 0.291 | 0.633 | 0.533 | 0.367 |
| CoCoA$_{MTE}$ | 0.228 | 0.439 | 0.577 | 0.395 | 0.685 | 0.517 | **0.510** |
| CoCoA$_{MTE}$ Light | 0.234 | **0.518** | 0.549 | 0.345 | 0.693 | 0.491 | 0.500 |

Table 11: Detailed experimental results. Main model response obtained by greedy decoding.

| Method | Dataset | | | | | | |
|---|---|---|---|---|---|---|---|
| | XSUM | WMT14FrEn | WMT19DeEn | CoQA | TriviaQA | MMLU | GSM8k |
| Mistral7b-Base | | | | | | | |
| MCSE | 0.162 | 0.406 | 0.407 | 0.289 | 0.492 | 0.339 | 0.693 |
| MCNSE | 0.196 | 0.471 | 0.507 | 0.277 | 0.529 | 0.358 | 0.700 |
| Semantic Entropy | 0.164 | 0.434 | 0.442 | 0.312 | 0.554 | 0.389 | 0.675 |
| DegMat | 0.205 | 0.439 | 0.410 | 0.376 | 0.618 | 0.410 | 0.454 |
| EigValLaplacian | 0.197 | 0.388 | 0.344 | 0.342 | 0.600 | 0.399 | 0.361 |
| SAR | 0.175 | 0.563 | 0.590 | 0.347 | 0.620 | 0.421 | 0.780 |
| P(True) | 0.207 | 0.472 | 0.269 | -0.058 | -0.084 | 0.068 | 0.278 |
| Consistency | 0.071 | 0.670 | 0.708 | $\underline{0.405}$ | 0.614 | 0.423 | 0.846 |
| SP | 0.330 | 0.212 | 0.291 | 0.388 | 0.607 | $\underline{0.476}$ | 0.307 |
| CoCoA$_{SP}$ | 0.366 | $\underline{0.712}$ | 0.730 | **0.430** | 0.644 | 0.466 | 0.562 |
| PPL | 0.365 | 0.695 | 0.676 | 0.327 | 0.615 | $\underline{0.476}$ | 0.931 |
| CoCoA$_{PPL}$ | $\underline{0.372}$ | **0.735** | **0.755** | 0.402 | 0.648 | 0.466 | **0.937** |
| MTE | 0.350 | 0.668 | 0.606 | 0.254 | 0.594 | 0.457 | 0.932 |
| CoCoA$_{MTE}$ | **0.373** | 0.708 | $\underline{0.732}$ | 0.373 | 0.645 | 0.447 | $\underline{0.935}$ |
| Llama8b-Base | | | | | | | |
| MCSE | 0.192 | 0.366 | 0.395 | 0.258 | 0.465 | 0.158 | 0.545 |
| MCNSE | 0.186 | 0.377 | 0.480 | 0.239 | 0.484 | 0.165 | 0.631 |
| Semantic Entropy | 0.194 | 0.371 | 0.451 | 0.286 | 0.528 | 0.213 | 0.557 |
| DegMat | 0.191 | 0.274 | 0.409 | 0.366 | **0.606** | 0.320 | 0.396 |
| EigValLaplacian | 0.190 | 0.216 | 0.333 | 0.339 | 0.587 | 0.274 | 0.351 |
| SAR | 0.159 | 0.441 | 0.571 | 0.327 | 0.578 | 0.340 | 0.667 |
| P(True) | 0.058 | 0.075 | 0.056 | -0.011 | -0.071 | -0.120 | -0.084 |
| Consistency | 0.030 | 0.473 | 0.598 | **0.394** | $\underline{0.600}$ | 0.353 | 0.793 |
| SP | $\underline{0.464}$ | 0.339 | 0.413 | 0.304 | 0.514 | **0.483** | 0.280 |
| CoCoA$_{SP}$ | **0.484** | 0.529 | $\underline{0.685}$ | $\underline{0.384}$ | 0.587 | 0.452 | 0.513 |
| PPL | 0.458 | 0.504 | 0.622 | 0.293 | 0.483 | 0.441 | 0.911 |
| CoCoA$_{PPL}$ | 0.450 | **0.544** | **0.689** | 0.364 | 0.573 | 0.422 | $\underline{0.924}$ |
| MTE | 0.449 | 0.437 | 0.501 | 0.238 | 0.458 | 0.326 | 0.883 |
| CoCoA$_{MTE}$ | 0.451 | 0.520 | 0.638 | 0.345 | 0.582 | 0.363 | 0.899 |
| Falcon7b-Base | | | | | | | |
| MCSE | 0.128 | 0.399 | 0.419 | 0.285 | 0.535 | 0.421 | 0.598 |
| MCNSE | 0.153 | 0.395 | 0.452 | 0.318 | 0.588 | 0.443 | 0.771 |
| Semantic Entropy | 0.134 | 0.420 | 0.460 | 0.319 | 0.566 | 0.463 | 0.567 |
| DegMat | 0.177 | 0.350 | 0.422 | $\underline{0.422}$ | 0.637 | 0.480 | 0.633 |
| EigValLaplacian | 0.174 | 0.289 | 0.382 | 0.393 | 0.622 | 0.454 | 0.522 |
| SAR | 0.193 | 0.455 | 0.521 | 0.385 | 0.642 | 0.509 | 0.826 |
| P(True) | 0.022 | 0.245 | 0.245 | 0.038 | 0.244 | 0.028 | 0.815 |
| Consistency | 0.282 | 0.491 | 0.651 | 0.416 | 0.627 | 0.484 | 0.979 |
| SP | 0.099 | 0.385 | 0.378 | 0.369 | 0.638 | **0.540** | -0.175 |
| CoCoA$_{SP}$ | 0.210 | 0.564 | $\underline{0.698}$ | **0.428** | 0.659 | 0.530 | 0.498 |
| PPL | 0.275 | 0.541 | 0.637 | 0.353 | 0.614 | **0.540** | **1.000** |
| CoCoA$_{PPL}$ | **0.310** | $\underline{0.579}$ | **0.717** | 0.415 | 0.644 | 0.530 | **1.000** |
| MTE | 0.186 | 0.475 | 0.510 | 0.317 | 0.573 | $\underline{0.534}$ | 0.984 |
| CoCoA$_{MTE}$ | 0.289 | 0.551 | 0.678 | 0.402 | 0.646 | 0.517 | $\underline{0.998}$ |

Table 12: Detailed experimental results. Main model response obtained by selecting most probable candidate among stochastically sampled responses.

| Method | Dataset | | | | | | |
|---|---|---|---|---|---|---|---|
| | XSUM | WMT14FrEn | WMT19DeEn | CoQA | TriviaQA | MMLU | GSM8k |
| Mistral7b-Base | | | | | | | |
| MCSE | 0.123 | 0.297 | 0.363 | 0.245 | 0.515 | 0.385 | 0.751 |
| MCNSE | 0.105 | 0.385 | 0.485 | 0.256 | 0.552 | 0.406 | 0.548 |
| Semantic Entropy | 0.127 | 0.310 | 0.395 | 0.272 | 0.580 | 0.439 | 0.735 |
| DegMat | 0.244 | 0.280 | 0.365 | **0.358** | 0.673 | 0.461 | 0.392 |
| EigValLaplacian | 0.243 | 0.238 | 0.319 | 0.334 | 0.654 | 0.443 | 0.318 |
| SAR | 0.204 | 0.392 | 0.503 | 0.334 | 0.658 | 0.478 | 0.656 |
| P(True) | 0.079 | 0.142 | 0.084 | -0.056 | -0.084 | 0.047 | -0.084 |
| Consistency | 0.211 | 0.297 | 0.437 | 0.355 | 0.659 | 0.458 | 0.603 |
| SP | 0.259 | 0.381 | 0.486 | 0.216 | 0.346 | **0.514** | 0.711 |
| CoCoA$_{MTE}$ | **0.295** | 0.441 | **0.589** | 0.336 | 0.597 | **0.514** | 0.766 |
| PPL | 0.205 | 0.391 | 0.513 | 0.186 | 0.358 | **0.514** | 0.672 |
| CoCoA$_{PPL}$ | 0.258 | **0.451** | 0.577 | 0.322 | 0.612 | **0.514** | 0.768 |
| MTE | 0.156 | 0.443 | 0.550 | 0.234 | 0.598 | 0.508 | 0.717 |
| CoCoA$_{MTE}$ | 0.236 | 0.419 | 0.553 | 0.337 | **0.680** | 0.494 | **0.783** |
| Llama8b-Base | | | | | | | |
| MCSE | 0.089 | 0.240 | 0.354 | 0.220 | 0.444 | 0.297 | 0.647 |
| MCNSE | 0.055 | 0.334 | 0.443 | 0.247 | 0.470 | 0.306 | 0.545 |
| Semantic Entropy | 0.089 | 0.250 | 0.382 | 0.245 | 0.514 | 0.361 | 0.650 |
| DegMat | 0.217 | 0.218 | 0.361 | 0.368 | **0.611** | 0.391 | 0.539 |
| EigValLaplacian | 0.217 | 0.167 | 0.280 | 0.347 | 0.592 | 0.305 | 0.492 |
| SAR | 0.125 | 0.393 | 0.520 | 0.345 | 0.578 | 0.472 | 0.642 |
| P(True) | -0.014 | 0.098 | 0.091 | -0.014 | -0.054 | -0.210 | 0.083 |
| Consistency | 0.146 | 0.290 | 0.440 | **0.387** | 0.603 | 0.428 | 0.599 |
| SP | 0.197 | 0.344 | 0.463 | 0.161 | 0.247 | 0.577 | 0.675 |
| CoCoA$_{MTE}$ | **0.225** | 0.403 | **0.590** | 0.304 | 0.487 | **0.586** | 0.731 |
| PPL | 0.168 | 0.382 | 0.460 | 0.161 | 0.253 | 0.549 | 0.668 |
| CoCoA$_{PPL}$ | 0.198 | **0.428** | 0.580 | 0.320 | 0.496 | 0.562 | 0.751 |
| MTE | 0.130 | 0.412 | 0.509 | 0.245 | 0.466 | 0.459 | 0.699 |
| CoCoA$_{MTE}$ | 0.189 | 0.398 | 0.561 | 0.345 | 0.587 | 0.472 | **0.765** |
| Falcon7b-Base | | | | | | | |
| MCSE | 0.188 | 0.331 | 0.362 | 0.274 | 0.560 | 0.455 | 0.460 |
| MCNSE | 0.126 | 0.387 | 0.494 | 0.306 | 0.605 | 0.479 | 0.396 |
| Semantic Entropy | 0.192 | 0.332 | 0.414 | 0.304 | 0.594 | 0.497 | 0.438 |
| DegMat | 0.246 | 0.271 | 0.470 | **0.392** | **0.672** | 0.516 | 0.415 |
| EigValLaplacian | 0.238 | 0.220 | 0.432 | 0.365 | 0.660 | 0.476 | 0.339 |
| SAR | 0.184 | 0.412 | 0.543 | 0.370 | 0.664 | 0.553 | 0.536 |
| P(True) | -0.124 | 0.194 | 0.161 | 0.016 | 0.279 | 0.061 | 0.284 |
| Consistency | 0.235 | 0.323 | 0.498 | 0.376 | 0.665 | 0.523 | 0.588 |
| SP | 0.187 | 0.355 | 0.398 | 0.213 | 0.458 | **0.583** | 0.367 |
| CoCoA$_{SP}$ | **0.248** | 0.461 | 0.590 | 0.336 | 0.617 | 0.575 | 0.472 |
| PPL | 0.177 | 0.371 | 0.504 | 0.201 | 0.439 | **0.583** | 0.556 |
| CoCoA$_{PPL}$ | 0.242 | **0.462** | **0.613** | 0.345 | 0.624 | 0.575 | **0.662** |
| MTE | 0.159 | **0.462** | 0.580 | 0.290 | 0.588 | 0.565 | 0.517 |
| CoCoA$_{MTE}$ | 0.237 | 0.454 | 0.600 | 0.373 | 0.666 | 0.557 | 0.650 |

Table 13: Detailed experimental results. Main model response obtained by MBR decoding.

# F Alternative Performance Metrics for PRR

The choice of PRR as a UQ quality metric of choice is dictated by its ability to handle both continuous performance metrics, like Comet and AlignScore without the need for selecting arbitrary thresholds, as well as relative robustness to class imbalance in case of binary performance metric. However, PRR scores are calculated for a particular choice of underlying performance metric. For a truly comprehensive evaluation, we perform the same evaluation as in our main experimental run, but with alternative choice of performance metrics.

Tables 14 and 15 report PRRs for these metrics on all models for NMT and QA tasks. MetricX [Juraska et al., 2024] was used for NMT and GPT-as-a-judge (gpt-4o-2024-08-06) was used to score QA datasets. The following is the prompt used to facilitate GPT QA scoring:

```
You are a text evaluator. The model was asked the following question:
{question}
The 'Generated' text is a model's response. The 'Target' is the correct answer.
If the generated answer correctly answers the question based on the target, return 1.
If it is wrong, return 0.
Respond ONLY with a single digit: 1 or 0.

Generated: {model output}
Target: {target sequence}
```

| | NMT (MetricX) | | QA (GPT-4o) | | | |
|---|---|---|---|---|---|---|
| Method | WMT14FrEn | WMT19DeEn | CoQA/Gpt | TriviaQA/Gpt | MMLU/Gpt | GSM8k/Gpt |
| Mistral7b-Base | | | | | | |
| MCSE | 0.127 | 0.212 | 0.226 | 0.478 | 0.338 | 0.453 |
| MCNSE | 0.267 | 0.417 | 0.246 | 0.535 | 0.357 | 0.388 |
| Semantic Entropy | 0.158 | 0.261 | 0.254 | 0.554 | 0.387 | 0.454 |
| CEDegMat | 0.270 | 0.402 | 0.336 | 0.659 | 0.402 | 0.356 |
| SAR | 0.302 | 0.451 | 0.333 | 0.660 | 0.419 | 0.456 |
| Semantic Density | 0.291 | 0.366 | 0.056 | **0.688** | 0.335 | 0.195 |
| SP | 0.173 | 0.287 | 0.287 | 0.640 | **0.473** | 0.485 |
| $CoCoA_{SP}$ | 0.314 | 0.462 | **0.367** | 0.681 | 0.465 | **0.537** |
| PPL | 0.316 | 0.448 | 0.220 | 0.645 | **0.473** | 0.345 |
| $CoCoA_{PPL}$ | 0.361 | **0.512** | 0.331 | 0.687 | 0.465 | 0.499 |
| MTE | 0.350 | 0.455 | 0.190 | 0.631 | 0.456 | 0.387 |
| $CoCoA_{MTE}$ | **0.366** | **0.512** | 0.323 | 0.684 | 0.447 | 0.524 |
| Llama8b-Base | | | | | | |
| MCSE | 0.145 | 0.181 | 0.200 | 0.486 | 0.158 | 0.332 |
| MCNSE | 0.319 | 0.361 | 0.183 | 0.530 | 0.170 | 0.324 |
| Semantic Entropy | 0.157 | 0.235 | 0.239 | 0.552 | 0.203 | 0.361 |
| CEDegMat | 0.287 | 0.367 | **0.350** | 0.621 | 0.299 | 0.300 |
| SAR | 0.354 | 0.420 | 0.302 | 0.610 | 0.319 | 0.382 |
| Semantic Density | 0.252 | 0.328 | 0.069 | **0.650** | 0.335 | 0.299 |
| SP | 0.176 | 0.274 | 0.228 | 0.564 | **0.469** | 0.313 |
| $CoCoA_{SP}$ | 0.329 | 0.451 | 0.332 | 0.625 | 0.449 | 0.365 |
| PPL | 0.327 | 0.388 | 0.217 | 0.549 | 0.462 | 0.309 |
| $CoCoA_{PPL}$ | **0.395** | 0.469 | 0.329 | 0.623 | 0.443 | 0.445 |
| MTE | 0.353 | 0.388 | 0.214 | 0.543 | 0.358 | 0.329 |
| $CoCoA_{MTE}$ | 0.394 | **0.472** | 0.333 | 0.634 | 0.393 | **0.455** |
| Falcon7b-Base | | | | | | |
| MCSE | 0.168 | 0.218 | 0.228 | 0.587 | 0.420 | 0.443 |
| MCNSE | 0.304 | 0.399 | 0.265 | 0.640 | 0.442 | 0.323 |
| Semantic Entropy | 0.201 | 0.276 | 0.262 | 0.623 | 0.463 | 0.448 |
| CEDegMat | 0.287 | 0.428 | **0.380** | 0.712 | 0.476 | 0.347 |
| SAR | 0.339 | 0.457 | 0.367 | 0.715 | 0.508 | 0.410 |
| Semantic Density | 0.323 | 0.422 | 0.076 | 0.746 | 0.396 | 0.349 |
| SP | 0.176 | 0.275 | 0.279 | 0.737 | **0.539** | 0.391 |
| $CoCoA_{SP}$ | 0.335 | 0.483 | 0.364 | **0.763** | 0.529 | 0.459 |
| PPL | 0.322 | 0.454 | 0.245 | 0.722 | **0.539** | 0.341 |
| $CoCoA_{PPL}$ | 0.394 | **0.531** | 0.345 | 0.754 | 0.529 | 0.513 |
| MTE | 0.370 | 0.455 | 0.228 | 0.707 | 0.533 | 0.385 |
| $CoCoA_{MTE}$ | **0.403** | 0.529 | 0.341 | 0.762 | 0.516 | **0.551** |

Table 14: PRRs for all models on QA and NMT tasks with alternative choice of performance metrics. Main model response obtained by greedy decoding.

|  | NMT (MetricX) | | QA (GPT-4o) | | | |
|---|---|---|---|---|---|---|
| Method | WMT14FrEn | WMT19DeEn | CoQA/Gpt | TriviaQA/Gpt | MMLU/Gpt | GSM8k/Gpt |
| Mistral7b-Base | | | | | | |
| MCSE | 0.305 | 0.324 | 0.244 | 0.464 | 0.339 | 0.392 |
| MCNSE | 0.360 | 0.466 | 0.252 | 0.516 | 0.358 | 0.188 |
| Semantic Entropy | 0.336 | 0.366 | 0.272 | 0.536 | 0.389 | 0.420 |
| CEDegMat | 0.468 | 0.485 | 0.339 | 0.623 | 0.404 | 0.239 |
| SAR | 0.455 | 0.527 | 0.336 | 0.628 | 0.421 | 0.298 |
| Semantic Density | 0.565 | 0.554 | 0.043 | 0.646 | 0.306 | 0.421 |
| SP | 0.169 | 0.252 | 0.311 | 0.608 | **0.474** | 0.460 |
| CoCoA$_{SP}$ | 0.607 | 0.623 | **0.376** | 0.647 | 0.466 | **0.550** |
| PPL | 0.510 | 0.552 | 0.265 | 0.619 | **0.474** | 0.096 |
| CoCoA$_{PPL}$ | **0.620** | **0.667** | 0.346 | **0.655** | 0.466 | 0.091 |
| MTE | 0.487 | 0.499 | 0.211 | 0.604 | 0.457 | 0.167 |
| CoCoA$_{MTE}$ | 0.581 | 0.638 | 0.324 | 0.653 | 0.447 | 0.151 |
| Llama8b-Base | | | | | | |
| MCSE | 0.284 | 0.271 | 0.191 | 0.451 | 0.155 | 0.302 |
| MCNSE | 0.354 | 0.419 | 0.197 | 0.493 | 0.171 | 0.251 |
| Semantic Entropy | 0.299 | 0.333 | 0.230 | 0.515 | 0.201 | 0.341 |
| CEDegMat | 0.343 | 0.451 | **0.369** | 0.598 | 0.301 | 0.241 |
| SAR | 0.412 | 0.497 | 0.315 | 0.584 | 0.323 | 0.311 |
| Semantic Density | 0.403 | 0.452 | 0.065 | **0.638** | 0.286 | 0.396 |
| SP | 0.247 | 0.289 | 0.225 | 0.523 | **0.467** | 0.332 |
| CoCoA$_{SP}$ | 0.473 | 0.568 | 0.346 | 0.597 | 0.440 | **0.418** |
| PPL | 0.437 | 0.517 | 0.238 | 0.498 | 0.459 | 0.185 |
| CoCoA$_{PPL}$ | **0.510** | **0.614** | 0.335 | 0.588 | 0.433 | 0.199 |
| MTE | 0.387 | 0.439 | 0.201 | 0.485 | 0.345 | 0.226 |
| CoCoA$_{MTE}$ | 0.495 | 0.589 | 0.330 | 0.604 | 0.372 | 0.237 |
| Falcon7b-Base | | | | | | |
| MCSE | 0.269 | 0.341 | 0.261 | 0.553 | 0.421 | 0.159 |
| MCNSE | 0.325 | 0.403 | 0.312 | 0.620 | 0.443 | 0.064 |
| Semantic Entropy | 0.308 | 0.397 | 0.296 | 0.588 | 0.463 | 0.172 |
| CEDegMat | 0.381 | 0.478 | 0.398 | 0.686 | 0.477 | 0.083 |
| SAR | 0.389 | 0.483 | 0.398 | 0.685 | 0.509 | 0.152 |
| Semantic Density | 0.434 | 0.518 | 0.047 | 0.661 | 0.377 | 0.299 |
| SP | 0.269 | 0.311 | 0.340 | 0.683 | **0.540** | 0.104 |
| CoCoA$_{SP}$ | 0.465 | 0.621 | **0.407** | **0.705** | 0.530 | **0.300** |
| PPL | 0.430 | 0.553 | 0.308 | 0.663 | **0.540** | 0.156 |
| CoCoA$_{PPL}$ | **0.526** | **0.655** | 0.376 | 0.691 | 0.530 | 0.128 |
| MTE | 0.418 | 0.452 | 0.275 | 0.615 | 0.535 | 0.054 |
| CoCoA$_{MTE}$ | 0.512 | 0.620 | 0.370 | 0.692 | 0.517 | 0.050 |

Table 15: PRRs for all models on QA and NMT tasks with alternative choice of performance metrics. Main model response obtained by selecting the most probable candidate among stochastically sampled responses.

# G AUROC

We strongly believe that due to the considerations presented in Appendix F, PRR is a superior metric to AUROC when comparing relative performance of UQ methods. However, we acknowledge that AUROC is widely used in modern UQ literature, so we opt to include AUROC results on QA datasets. The results are presented in Tables 16 and 17. All results here were obtained with GPT-as-a-judge correctness scoring.

| | CoQA/Gpt | TriviaQA/Gpt | MMLU/Gpt | GSM8k/Gpt |
|---|---|---|---|---|
| Mistral7b-Base | | | | |
| MCSE | 0.642 | 0.784 | 0.737 | 0.716 |
| MCNSE | 0.653 | 0.806 | 0.744 | 0.684 |
| Semantic Entropy | 0.653 | 0.817 | 0.756 | 0.716 |
| CEDegMat | 0.690 | 0.859 | 0.759 | 0.682 |
| SAR | 0.694 | 0.862 | 0.770 | 0.717 |
| Semantic Density | 0.531 | **0.871** | 0.713 | 0.576 |
| SP | 0.674 | 0.849 | **0.794** | 0.684 |
| CoCoA$_{SP}$ | **0.708** | 0.867 | 0.790 | **0.729** |
| PPL | 0.636 | 0.855 | **0.794** | 0.628 |
| CoCoA$_{PPL}$ | 0.690 | **0.871** | 0.790 | 0.714 |
| MTE | 0.620 | 0.849 | 0.787 | 0.640 |
| CoCoA$_{MTE}$ | 0.687 | 0.870 | 0.783 | 0.727 |
| Llama8b-Base | | | | |
| MCSE | 0.635 | 0.779 | 0.620 | 0.705 |
| MCNSE | 0.630 | 0.799 | 0.629 | 0.689 |
| Semantic Entropy | 0.655 | 0.815 | 0.652 | 0.713 |
| CEDegMat | **0.708** | 0.845 | 0.703 | 0.691 |
| SAR | 0.694 | 0.842 | 0.715 | 0.724 |
| Semantic Density | 0.540 | **0.855** | 0.680 | 0.655 |
| SP | 0.647 | 0.816 | **0.787** | 0.669 |
| CoCoA$_{SP}$ | 0.700 | 0.847 | 0.776 | 0.718 |
| PPL | 0.640 | 0.815 | 0.778 | 0.661 |
| CoCoA$_{PPL}$ | 0.699 | 0.848 | 0.770 | 0.745 |
| MTE | 0.633 | 0.817 | 0.726 | 0.667 |
| CoCoA$_{MTE}$ | 0.702 | 0.854 | 0.746 | **0.753** |
| Falcon7b-Base | | | | |
| MCSE | 0.648 | 0.807 | 0.770 | 0.766 |
| MCNSE | 0.660 | 0.831 | 0.779 | 0.709 |
| Semantic Entropy | 0.669 | 0.830 | 0.788 | 0.761 |
| CEDegMat | 0.718 | 0.870 | 0.784 | 0.722 |
| SAR | 0.713 | 0.870 | 0.805 | 0.757 |
| Semantic Density | 0.544 | 0.881 | 0.746 | 0.716 |
| SP | 0.681 | 0.866 | **0.820** | 0.731 |
| CoCoA$_{SP}$ | **0.719** | 0.884 | 0.815 | 0.774 |
| PPL | 0.645 | 0.860 | **0.820** | 0.700 |
| CoCoA$_{PPL}$ | 0.705 | 0.882 | 0.815 | 0.800 |
| MTE | 0.630 | 0.854 | 0.818 | 0.721 |
| CoCoA$_{MTE}$ | 0.704 | **0.886** | 0.810 | **0.814** |

Table 16: AUROC for all models on QA tasks. Main model response was obtained by greedy decoding.

| | CoQA/Gpt | TriviaQA/Gpt | MMLU/Gpt | GSM8k/Gpt |
|---|---|---|---|---|
| | Mistral7b-Base | | | |
| MCSE | 0.650 | 0.777 | 0.738 | 0.676 |
| MCNSE | 0.656 | 0.798 | 0.745 | 0.602 |
| Semantic Entropy | 0.661 | 0.808 | 0.757 | 0.692 |
| CEDegMat | 0.693 | 0.845 | 0.760 | 0.625 |
| SAR | 0.695 | 0.849 | 0.771 | 0.643 |
| Semantic Density | 0.523 | 0.854 | 0.696 | 0.633 |
| SP | 0.684 | 0.836 | **0.795** | 0.663 |
| $CoCoA_{SP}$ | **0.713** | 0.853 | 0.790 | **0.720** |
| PPL | 0.657 | 0.843 | **0.795** | 0.592 |
| $CoCoA_{PPL}$ | 0.697 | **0.858** | 0.790 | 0.615 |
| MTE | 0.627 | 0.838 | 0.788 | 0.606 |
| $CoCoA_{MTE}$ | 0.689 | **0.858** | 0.783 | 0.632 |
| | Llama8b-Base | | | |
| MCSE | 0.632 | 0.763 | 0.617 | 0.653 |
| MCNSE | 0.633 | 0.783 | 0.629 | 0.629 |
| Semantic Entropy | 0.653 | 0.800 | 0.650 | 0.669 |
| CEDegMat | **0.716** | 0.836 | 0.703 | 0.633 |
| SAR | 0.698 | 0.831 | 0.715 | 0.659 |
| Semantic Density | 0.534 | **0.849** | 0.655 | 0.666 |
| SP | 0.647 | 0.801 | **0.787** | 0.627 |
| $CoCoA_{SP}$ | 0.706 | 0.836 | 0.771 | **0.686** |
| PPL | 0.648 | 0.796 | 0.776 | 0.635 |
| $CoCoA_{PPL}$ | 0.702 | 0.835 | 0.765 | 0.658 |
| MTE | 0.624 | 0.794 | 0.719 | 0.644 |
| $CoCoA_{MTE}$ | 0.701 | 0.842 | 0.736 | 0.673 |
| | Falcon7b-Base | | | |
| MCSE | 0.662 | 0.797 | 0.770 | 0.589 |
| MCNSE | 0.682 | 0.826 | 0.779 | 0.571 |
| Semantic Entropy | 0.684 | 0.821 | 0.788 | 0.593 |
| CEDegMat | 0.726 | 0.863 | 0.785 | 0.580 |
| SAR | 0.727 | 0.862 | 0.806 | 0.611 |
| Semantic Density | 0.525 | 0.856 | 0.734 | **0.684** |
| SP | 0.706 | 0.850 | **0.820** | 0.555 |
| $CoCoA_{SP}$ | **0.738** | **0.869** | 0.815 | 0.667 |
| PPL | 0.677 | 0.844 | **0.820** | 0.657 |
| $CoCoA_{PPL}$ | 0.719 | 0.866 | 0.815 | 0.657 |
| MTE | 0.654 | 0.825 | 0.818 | 0.603 |
| $CoCoA_{MTE}$ | 0.717 | 0.867 | 0.810 | 0.628 |

Table 17: AUROC for all models on QA tasks. Main model response obtained by selecting the most probable candidate among stochastically sampled responses

## H Computational Budget

Total available computational resources used to produce results in this paper amounted to 12 compute nodes each having 4xA100 40Gb GPUs. Total computational budget spent to produce results was around 400 GPU-days. Each individual combination of model and dataset amounted roughly to 16 GPU-days on average to obtain all model outputs and hidden states needed for computing the results.

# I    CoCoA Light Training Details

The model architecture consisted of a simple multilayer perceptron (MLP) with a single hidden layer:

- **Input dimension:** equal to the embedding size of the corresponding base model (4096 for Mistral7b-Base and Llama8b-Base, 3072 for Falcon7b-Base).
- **Hidden dimension:** 2048.
- **Output dimension:** 4096.
- **Dropout:** 0.1 applied between layers.

We trained the MLP on mean pooled hidden-layer embeddings extracted from the middle layer of an LLM. Table 18 details the size of a train set for each model and dataset.

| Dataset | LLaMA | Falcon | Mistral |
|---|---|---|---|
| CoQA | 10,000 | 10,000 | 10,000 |
| GSM8K | 3,000 | 3,000 | 2,500 |
| MMLU | 1,461 | 1,461 | 1,461 |
| TriviaQA | 10,000 | 10,000 | 10,000 |
| WMT14 Fr-En | 6,000 | 6,000 | 6,000 |
| WMT19 De-En | 6,000 | 6,000 | 6,000 |
| XSum | 7,500 | 5,000 | 6,500 |

Table 18: Training set sizes (number of examples) for Llama8b-Base, Falcon7b-Base, and Mistral7b-Base across several datasets.

Training was conducted for 20 epochs with the following hyperparameters:

- **Batch size:** 4 (per device) with gradient accumulation of 7 steps (effective batch size of 28).
- **Learning rate:** $1 \times 10^{-5}$ with AdamW optimizer.
- **Weight decay:** 0.1.
- **Warmup ratio:** 0.05.
- **Gradient clipping:** 1.0.

## J   Larger Model Experiments

We additionally evaluate our method using the Gemma 3 12B-Base model to assess its performance on a larger-scale architecture. As shown in Table 19, CoCoA continues to achieve the best average performance among all methods.

| Method | Dataset | | | | | | |
|---|---|---|---|---|---|---|---|
| | XSUM | WMT14FrEn | WMT19DeEn | CoQA | TriviaQA | MMLU | GSM8k |
| MCSE | -0.014 | 0.277 | 0.408 | 0.262 | 0.556 | 0.449 | 0.386 |
| MCNSE | -0.031 | 0.393 | 0.442 | 0.280 | 0.595 | 0.468 | 0.344 |
| Semantic Entropy | -0.011 | 0.273 | 0.430 | 0.287 | 0.621 | 0.531 | 0.407 |
| DegMat | 0.130 | 0.246 | 0.370 | 0.356 | 0.692 | 0.561 | 0.359 |
| EigValLaplacian | 0.130 | 0.195 | 0.307 | 0.323 | 0.667 | 0.526 | 0.312 |
| SAR | 0.087 | 0.429 | 0.494 | 0.335 | 0.688 | 0.582 | 0.418 |
| P(True) | -0.022 | 0.012 | -0.020 | -0.006 | 0.195 | 0.046 | 0.145 |
| Consistency Light | -0.006 | 0.328 | 0.483 | 0.345 | 0.709 | 0.408 | 0.421 |
| Consistency | -0.163 | 0.452 | 0.496 | 0.311 | 0.657 | 0.293 | 0.387 |
| MSP | 0.257 | 0.315 | 0.499 | 0.319 | 0.672 | **0.630** | 0.335 |
| CoCoA$_{MSP}$ | 0.308 | 0.424 | **0.641** | **0.372** | 0.723 | 0.617 | 0.391 |
| CoCoA$_{MSP}$ Light | 0.287 | 0.416 | 0.634 | 0.356 | 0.720 | 0.613 | 0.360 |
| PPL | 0.303 | 0.369 | 0.451 | 0.288 | 0.681 | **0.630** | 0.275 |
| CoCoA$_{PPL}$ | **0.339** | 0.427 | 0.539 | 0.362 | **0.728** | 0.617 | 0.439 |
| CoCoA$_{PPL}$ Light | 0.327 | 0.489 | 0.518 | 0.334 | 0.727 | 0.613 | 0.353 |
| MTE | 0.284 | 0.372 | 0.413 | 0.269 | 0.664 | 0.617 | 0.310 |
| CoCoA$_{MTE}$ | 0.326 | 0.422 | 0.527 | 0.344 | 0.728 | 0.592 | **0.468** |
| CoCoA$_{MTE}$ Light | 0.308 | **0.501** | 0.498 | 0.313 | 0.719 | 0.565 | 0.386 |

Table 19: Detailed experimental results for Gemma 3 12B-Base model. Model response obtained by greedy decoding

| Method | Dataset | | | | | | |
|---|---|---|---|---|---|---|---|
| | XSUM | WMT14FrEn | WMT19DeEn | CoQA | TriviaQA | MMLU | GSM8k |
| MCSE | 0.051 | 0.346 | 0.447 | 0.273 | 0.531 | 0.447 | 0.495 |
| MCNSE | 0.088 | 0.441 | 0.456 | 0.319 | 0.580 | 0.465 | 0.533 |
| Semantic Entropy | 0.054 | 0.348 | 0.492 | 0.294 | 0.597 | 0.527 | 0.505 |
| DegMat | 0.165 | 0.294 | 0.372 | 0.366 | 0.659 | 0.558 | 0.457 |
| EigValLaplacian | 0.164 | 0.228 | 0.306 | 0.327 | 0.634 | 0.522 | 0.415 |
| SAR | 0.109 | 0.474 | 0.526 | 0.353 | 0.661 | 0.577 | 0.593 |
| P(True) | -0.029 | 0.037 | -0.015 | 0.019 | 0.193 | 0.046 | 0.278 |
| Consistency Light | 0.076 | 0.478 | 0.631 | 0.373 | 0.671 | 0.443 | 0.744 |
| MSP | 0.188 | 0.353 | 0.466 | 0.361 | 0.646 | **0.624** | 0.223 |
| CoCoA$_{MSP}$ | **0.251** | 0.542 | **0.718** | **0.406** | 0.693 | 0.613 | 0.433 |
| PPL | 0.178 | 0.501 | 0.619 | 0.349 | 0.660 | **0.624** | 0.750 |
| CoCoA$_{PPL}$ | 0.211 | **0.576** | 0.692 | 0.400 | 0.700 | 0.613 | **0.805** |
| MTE | 0.168 | 0.431 | 0.487 | 0.295 | 0.643 | 0.612 | 0.703 |
| CoCoA$_{MTE}$ | 0.212 | 0.546 | 0.641 | 0.369 | **0.702** | 0.591 | 0.796 |

Table 20: Detailed experimental results for Gemma 3 12B-Base model. Model response obtained by most probable sample decoding

| Method | Dataset | | | | | | |
|---|---|---|---|---|---|---|---|
| | XSUM | WMT14FrEn | WMT19DeEn | CoQA | TriviaQA | MMLU | GSM8k |
| MCSE | 0.120 | 0.290 | 0.361 | 0.271 | 0.544 | 0.487 | 0.493 |
| MCNSE | 0.126 | 0.366 | 0.439 | 0.308 | 0.597 | 0.497 | 0.464 |
| Semantic Entropy | 0.121 | 0.267 | 0.396 | 0.307 | 0.611 | 0.550 | 0.468 |
| DegMat | 0.195 | 0.222 | 0.318 | **0.399** | 0.684 | 0.571 | 0.373 |
| EigValLaplacian | 0.197 | 0.170 | 0.254 | 0.367 | 0.653 | 0.522 | 0.319 |
| SAR | 0.204 | 0.409 | 0.464 | 0.372 | 0.680 | 0.591 | 0.508 |
| P(True) | 0.030 | 0.043 | 0.018 | 0.014 | 0.204 | 0.079 | 0.313 |
| Consistency Light | 0.192 | 0.311 | 0.419 | 0.368 | 0.690 | 0.421 | 0.467 |
| MSP | 0.222 | 0.390 | 0.491 | 0.240 | 0.330 | 0.542 | 0.441 |
| CoCoA$_{MSP}$ | **0.282** | 0.452 | **0.598** | 0.347 | 0.624 | 0.574 | 0.498 |
| PPL | 0.187 | 0.400 | 0.526 | 0.183 | 0.344 | 0.542 | 0.468 |
| CoCoA$_{PPL}$ | 0.270 | **0.476** | 0.587 | 0.337 | 0.638 | 0.574 | 0.557 |
| MTE | 0.139 | 0.425 | 0.515 | 0.251 | 0.628 | **0.623** | 0.498 |
| CoCoA$_{MTE}$ | 0.230 | 0.451 | 0.528 | 0.361 | **0.712** | 0.610 | **0.575** |

Table 21: Detailed experimental results for Gemma 3 12B-Base model. Model response obtained by MBR decoding

