# OpenReview forum: "CoCoA: A Minimum Bayes Risk Framework Bridging Confidence and Consistency for Uncertainty Quantification in LLMs"
_NeurIPS.cc/2025/Conference — NeurIPS 2025 poster_

### Official Review · Reviewer_aWz6 · 2025-06-29

**Clarity:** 2
**Significance:** 2
**Originality:** 2
**Rating:** 4
**Confidence:** 3

**Summary:**

This paper introduces CoCoA (Confidence and Consistency-based Approaches), a new framing of methods for uncertainty quantification (UQ) in large language models (LLMs), grounded in the Minimum Bayes Risk (MBR) framework. The goal is to unify information-based (model confidence) and consistency-based (semantic agreement between sampled outputs) UQ methods into a principled decision-making risk-based formulation. The authors also propose CoCoA Light, a learned approximation of the full CoCoA score that avoids repeated sampling using an additional held out set. They evaluate their methods on several datasets and compared with a few baseline methods.

**Questions:**

Please refers to my comments in the weakness section.

**Ethical Concerns:**

["NO or VERY MINOR ethics concerns only"]

**Final Justification:**

All details in the rebuttal history

**Limitations:**

The authors mentioned the limitations but not much on the societal impact. I will encourage the authors to discuss this aspect in their rebuttal.

**Paper Formatting Concerns:**

N / A

**Quality:**

1

**Strengths And Weaknesses:**

**Strengths**

1. Principled unification: The use of MBR as a unifying lens for both confidence and consistency is theoretically grounded (although reading line 47-49 and reference [9, 10] does not offer me the same hint, I wonder if the authors could elaborate for me). CoCoA makes this combination interpretable as a multiplicative interaction of model uncertainty and semantic inconsistency.

2. Practical algorithm: CoCoA Light reduces the sampling cost by learning to approximate the consistency-based term from representations, making it more applicable for real-world applications.

3. Empirical performance: Across diverse tasks and multiple open-source LLMs, CoCoA and CoCoA Light outperform good baselines.

4. In general well-written and easy to follow.

**Weaknesses**

1. Limited novelty compared to prior works: While the MBR framing is clean, many recent UQ methods already combine confidence and consistency (e.g., SAR, Semantic Entropy). The main novelty lies in formalizing their combination via MBR. I would like to ask the authors to explain a bit more their motivation from [9] and [10], and what is the novelty of this formulation.

2. Lack of theoretical justification: The main section for the CoCoA seems to build up a combination of model confidence and semantic inconsistency directly, without explaining the intuition and theoretical justification.

3. Similarity function bottleneck: Results depend heavily on the semantic similarity function used. The paper acknowledges this but could further explore when and which score should be used for certain tasks and why (which is really helpful if we do not have an additional held out set, e.g. open-ended generation tasks).

4. Measurement using PRR: While CoCoA improves PRR, it's less clear how well it calibrates model uncertainty. My understanding is this metric measures the ranking but not model uncertainty directly.

5. Error bars missing: I am wondering why the variance is not reported in this paper.

---

> ### Author Rebuttal · Authors · 2025-07-31
>
> We thank the reviewer for their valuable feedback and address the concerns raised below.
>
> >Limited novelty compared to prior works: While the MBR framing is clean, many recent UQ methods already combine confidence and consistency (e.g., SAR, Semantic Entropy). The main novelty lies in formalizing their combination via MBR. I would like to ask the authors to explain a bit more their motivation from [9] and [10], and what is the novelty of this formulation.
>
> We acknowledge that there are existing methods which blend confidence and consistency. However, we argue that our method proposes several novel contributions:
>
> While existing approaches have explored combining confidence and consistency and they did serve as an inspiration for our work, they tend to aggregate both confidence and consistency signals over all the sampled outputs, which reduces sensitivity to the expected quality of the particular model output (https://arxiv.org/pdf/2305.19187).
>
> This observation informs our design of CoCoA, which formalizes uncertainty estimation through a Minimum Bayes Risk (MBR) framework, interpreting uncertainty as expected risk based on both confidence and semantic consistency via explicit utility modeling. Within the MBR framework we introduced, semantic similarity and consistency are not used to reweight model confidence scores, but rather a separate signal integrated into the utility function.
>
> To further improve efficiency, we introduce CoCoA Light-a supervised approximation of the consistency component-which enables uncertainty estimation without the need for repeated sampling at inference time, which is another limitation of existing hybrid approaches.
>
> >Lack of theoretical justification: The main section for the CoCoA seems to build up a combination of model confidence and semantic inconsistency directly, without explaining the intuition and theoretical justification.
>
>
> While we don’t have the full theoretical treatment for the choice of the loss function for CoCoA (which is often a heuristic choice in MBR framework), we can consider a case study of the default confidence choice (MSP) that leads to the strongest performance.
>
> In this case, $U(y_*) = - \log p(y_* | x) \cdot \frac{1}{M} \sum_i (1 - s(y_*, y_i)$
>
> Equivalently, $U(y_*) = - \log [ p(y_* | x)^{\frac{1}{M} \sum_i (1 - s(y_*, y_i))}] = - \log [ p(y_* | x)^{ 1- \frac{1}{M} \sum_i s(y_*, y_i)}]$.
>
> This formula is essentially a criterion to select the maximum probability sequence but tempered by a semantic consistency of this sequence with other samples. From the MBR perspective, it is a natural criterion that balances choosing the sequence with high probability and simultaneously from a part of the distribution with highly concentrated meaning (high average similarity).
>
>
> >Measurement using PRR: While CoCoA improves PRR, it's less clear how well it calibrates model uncertainty. My understanding is this metric measures the ranking but not model uncertainty directly.
>
> While calibration is indeed a very important topic in UQ, it requires uncertainty scores to be interpretable as probabilities and the output correctness measure to be binary. Neither is true for a large portion of our experimental setup. Most UQ methods output unbounded values which can not be interpreted as probabilities, while many quality metrics are continuous, especially in NMT and summarization tasks. So straightforward computation of ECE is not possible.
>
> However, we adopted an approach to calibration proposed in (https://arxiv.org/abs/2406.15627) and fitted an isotonic regression to pairs of raw ue scores and response quality, and report MSE between ground truth quality and prediction of the regression here:
>
> | Metric | Llama QA | Llama NMT | Llama SUM | Mistral QA | Mistral NMT | Mistral SUM | Falcon QA | Falcon NMT | Falcon SUM |
> |:--|:--:|:--:|:--:|:--:|:--:|:--:|:--:|:--:|:--:|
> | MCSE | 0.177 | 0.005 | 0.074 | 0.159 | **0.004** | 0.099 | 0.140 | 0.005 | 0.059 |
> | MCNSE | 0.179 | 0.005 | 0.074 | 0.160 | **0.004** | 0.098 | 0.140 | **0.004** | 0.059 |
> | Semantic Entropy | 0.172 | 0.005 | 0.074 | 0.154 | **0.004** | 0.100 | 0.136 | 0.005 | 0.059 |
> | DegMat | 0.167 | 0.005 | 0.073 | 0.153 | **0.004** | 0.097 | 0.132 | 0.005 | 0.059 |
> | EigValLaplacian | 0.171 | 0.005 | 0.073 | 0.156 | **0.004** | 0.097 | 0.135 | 0.005 | 0.058 |
> | SAR | 0.164 | **0.004** | 0.073 | 0.150 | **0.004** | 0.098 | 0.131 | **0.004** | 0.059 |
> | Dissimilarity | 0.161 | **0.004** | 0.074 | 0.149 | **0.004** | 0.098 | 0.130 | **0.004** | 0.057 |
> | MSP | 0.165 | **0.004** | 0.066 | 0.151 | **0.004** | **0.088** | 0.131 | 0.005 | 0.058 |
> | CoCoA_MSP | **0.158** ↑ | **0.004** ↑ | **0.065** ↑ | **0.147** ↑ | **0.004** ↑ | **0.088** ↑ | 0.127 ↑ | **0.004** ↑ | **0.056** ↑ |
> | Perplexity | 0.170 | 0.005 | 0.066 | 0.155 | **0.004** | 0.089 | 0.135 | **0.004** | 0.058 |
> | CoCoA_Perplexity | 0.160 ↑ | **0.004** ↑ | **0.065** ↑ | 0.148 ↑ | **0.004** ↑ | 0.089 ↑ | 0.126 ↑ | **0.004** ↑ | 0.057 ↑ |
> | MeanTokenEntropy | 0.175 | 0.005 | 0.067 | 0.156 | **0.004** | **0.088** | 0.135 | **0.004** | 0.058 |
> | CoCoA_MeanTokenEntropy | 0.161 ↑ | **0.004** ↑ | **0.065** ↑ | 0.148 ↑ | **0.004** ↑ | 0.090 ↑ | **0.125** ↑ | **0.004** ↑ | 0.057 ↑ |
>
> *Results for Evaluated Sequence – Greedy Sample: Mean MSE between quality and calibrated UQ score across datasets for each task. The best performing method is in bold, and the second-best is underlined. Arrows indicate improvement in CoCoA over the base version.*
>
> CoCoA family methods still retain their edge over the baselines in most of the task-model pairs in this setting.
>
> >Similarity function bottleneck: Results depend heavily on the semantic similarity function used. The paper acknowledges this but could further explore when and which score should be used for certain tasks and why (which is really helpful if we do not have an additional held out set, e.g. open-ended generation tasks).
>
> This is indeed an important point, we thank the reviewer for bringing this up. Indeed, the choice of the similarity function is the important step here. We hypothesize that for general natural language generation with outputs of limited length (1-2 sentences) any capable NLI/CrossEncoder model will suffice. For longer generations, models that perform similarity estimation on the chunks with further aggregation over the chunks (like AlignScore for example) can be a better choice. For special cases, like code generation, a domain-specific similarity function would probably be needed.
>
> >Error bars missing: I am wondering why the variance is not reported in this paper.
>
> Thank you for pointing this out. We only ran a single instance of the experiment. Each run requires generating 2,000 outputs from the model, followed by extensive quality and other downstream evaluations. Due to the significant computational cost, repeating the experiment enough times to produce statistically meaningful error bars was not feasible within our resource constraints. We acknowledge this as a limitation and plan to address it in future work, resources permitting.

---

> > ### Comment · Reviewer_aWz6 · 2025-08-01
> > **Reply to Rebuttal**
> >
> > Thanks for your rebuttal and some of my concerns are addressed, however, I still think the current score seems to be fair for me. Here are the reasons:
> >
> > > "Limited novelty compared to prior works: While the MBR framing is clean, many recent UQ methods already combine confidence and consistency (e.g., SAR, Semantic Entropy). The main novelty lies in formalizing their combination via MBR. I would like to ask the authors to explain a bit more their motivation from [9] and [10], and what is the novelty of this formulation."
> >
> > I am not convinced but rather think MBR can be consider as a new angle of combing confidence and consistency score. However, I agree with the following contribution: *"we introduce CoCoA Light-a supervised approximation of the consistency component-which enables uncertainty estimation without the need for repeated sampling at inference time, which is another limitation of existing hybrid approaches."*
> >
> > > "Lack of theoretical justification: The main section for the CoCoA seems to build up a combination of model confidence and semantic inconsistency directly, without explaining the intuition and theoretical justification."
> >
> > Thank for your explanation, it makes a lot of sense! Although I think theoretical formulation is missing, perhaps this is something the authors could work on to combine with MBR view?
> >
> > > "Measurement using PRR: While CoCoA improves PRR, it's less clear how well it calibrates model uncertainty. My understanding is this metric measures the ranking but not model uncertainty directly."
> >
> > I am not entirely convinced by this method of regression uq to response quality beyond binary variable (I know, for example, the nature paper who introduce semantic entropy used something similar, but for binary decisions). And once again, the changes in this Table is barely minimum (are they actually significant at all?)
> >
> > > "Similarity function bottleneck: Results depend heavily on the semantic similarity function used. The paper acknowledges this but could further explore when and which score should be used for certain tasks and why (which is really helpful if we do not have an additional held out set, e.g. open-ended generation tasks)."
> >
> > Again, I think this is very critical bits of the paper which is missing, the utility function seems like the core building blocks of this proposed framework.
> >
> > > "Error bars missing: I am wondering why the variance is not reported in this paper."
> >
> > I would suggest the authors to add a section of discussion and limitations on the computational constrains if the results is presented based on over 2,000 samples. And maybe worthwhile to take one task and study the effect of sample size with respect to variance and uq results.
> >
> > Also, I would like to request **Reviewer zvVx** to kindly provide some justification on why given the score of 6 and everything (Quality, Clarity, Significance, Originality) a score of 4. I have checked on other reviewers' comments and think we share similar concerns.

---

> > > ### Author Response · Authors · 2025-08-02
> > >
> > > > I am not convinced but rather think MBR can be consider as a new angle of combing confidence and consistency score...
> > >
> > > First, it is important to discuss why existing methods combining token probabilities and semantic similarity of sampled outputs (SAR or Semantic Entropy) sometimes underperform in practice compared to simple baselines like MSP or PPL ([https://aclanthology.org/2025.tacl-1.11/](https://aclanthology.org/2025.tacl-1.11/)). We observed that LLM output probabilities can be unstable (e.g., Romeo and Juliet vs. Romeo & Juliet). As a result, approaches that integrate confidence and consistency by reweighting probabilities of the outputs based on the semantic similarities can also introduce a lot of noise into the estimation.
> > >
> > > In contrast, our MBR-based framework avoids this issue by focusing on confidence and consistency of an evaluated sequence. Rather than using semantic similarity to reweight probabilities, in the utility function we treat confidence and consistency as uncertainty signals, enabling more stable integration of two components. We also argue that confidence can be measured in various ways, such as maximum sequence probability, mean token entropy, or perplexity and each can be easily incorporated into our framework (see, also, new experimental results on claim-level uncertainty below). This modularity is a unique feature of our MBR-based framework, unlike existing methods such as SAR or Semantic Entropy, which rely on fixed formulations and do not offer the same flexibility in how confidence is defined.
> > >
> > > > Although I think theoretical formulation is missing...
> > >
> > > Our initial motivation when starting this work was to find the theoretically grounded way to combine confidence and consistency. We had Semantic Entropy and SAR as starting points. Unfortunately, as discussed above, any Bayesian-type combinations that aggregate probabilities of different generations (including SE and SAR theirselves) didn’t really show a stable performance across tasks. However, focus on the confidence in a single generation led to CoCoA which has extremely stable performance across tasks and models. We agree that explanation via MBR is partial, and we think that full theoretical justification is of interest. But, unfortunately, our (strong) efforts in this direction were not fully successful and we leave full theoretical justification to the future work.
> > >
> > > > I am not entirely convinced by this method of regression uq...
> > >
> > > While we agree that calibration is an important topic to be addressed, we stress that we focus on error detection, and consider the question of calibration to be somewhat out of scope for this submission, hence the original choice of evaluation metrics. We should note that the quality of probabilistic forecasts can be decomposed into calibration and sharpness [5], thus making calibration only a part of the prediction error. Thus, even perfectly calibrated models can be bad in prediction, which makes calibration only a secondary metric for our study.
> > >
> > > We also want to emphasize, that traditional calibration metrics **are not applicable to unbounded UQ scores**. Thus at the request of the esteemed reviewer, we used a known approach (see [1], [2]) to calibrate unbounded scores wrt to expected output quality.
> > >
> > > Regarding the paper [3] which we believe is the one being referred to here, we note that:
> > >
> > > - Firstly, this paper uses a rejection-based metric (referred to as AURAC in the paper) to compare performance of the methods under consideration, consistent with our work.
> > >
> > > - Secondly, for other performance metrics, the authors resort to thresholding the continuous correctness scores, which is known [4] to be quite non-robust wrt the choice of threshold value. These kinds of scores come from classification, and we believe are a poor fit to the free-generation setting of LLMs, especially when longer generations are common.
> > >
> > > We agree that in some tasks (NMT most notably), the resulting MSE between calibrated values and expected quality is very similar **for all considered baselines and proposed methods**, which is another reason to consider error detection as a better proxy for UQ performance in the setting of LLMs.
> > >
> > > [1] [https://direct.mit.edu/tacl/article/doi/10.1162/tacl_a_00737/128713/Benchmarking-Uncertainty-Quantification-Methods](https://direct.mit.edu/tacl/article/doi/10.1162/tacl_a_00737/128713/Benchmarking-Uncertainty-Quantification-Methods)
> > >
> > > [2] https://aclanthology.org/2024.emnlp-main.18.pdf
> > >
> > > [3] [https://www.nature.com/articles/s41586-024-07421-0](https://www.nature.com/articles/s41586-024-07421-0)
> > >
> > > [4] [https://openreview.net/forum?id=hbIpwrdfE1](https://openreview.net/forum?id=hbIpwrdfE1)
> > >
> > > [5] https://sites.stat.washington.edu/raftery/Research/PDF/Gneiting2007jrssb.pdf

---

> > > > ### Author Response · Authors · 2025-08-02
> > > >
> > > > > Error bars missing: I am wondering why the variance is not reported in this paper...
> > > >
> > > > We have computed the standard error of PRR scores for the greedy generation case over 50 bootstrap samples and report mean PRR score across samples along with standard error of the mean. The results confirm that all improvements over baselines are highly statistically significant.
> > > >
> > > > | Method           | llama8b/QA                      | llama8b/NMT                     | llama8b/SUM                     | mistral7b/QA                    | mistral7b/NMT                   | mistral7b/SUM                   | falcon7b/QA                     | falcon7b/NMT                    | falcon7b/SUM                    |
> > > > |:-----------------|:--------------------------------|:--------------------------------|:--------------------------------|:--------------------------------|:--------------------------------|:--------------------------------|:--------------------------------|:--------------------------------|:--------------------------------|
> > > > | MonteCarlo SE    | 0.310 ± 0.003     | 0.322 ± 0.004     | 0.036 ± 0.003     | 0.389 ± 0.003     | 0.305 ± 0.003     | 0.008 ± 0.003     | 0.413 ± 0.003     | 0.315 ± 0.003     | 0.160 ± 0.004     |
> > > > | MonteCarlo NSE   | 0.310 ± 0.003     | 0.389 ± 0.003     | 0.021 ± 0.003     | 0.385 ± 0.004     | 0.414 ± 0.003     | 0.011 ± 0.004     | 0.406 ± 0.003     | 0.419 ± 0.003     | 0.105 ± 0.004     |
> > > > | SemanticEntropy  | 0.356 ± 0.003     | 0.342 ± 0.004     | 0.037 ± 0.003     | 0.425 ± 0.003     | 0.329 ± 0.004     | 0.010 ± 0.003     | 0.438 ± 0.003     | 0.347 ± 0.003     | 0.166 ± 0.004     |
> > > > | DegMat           | 0.407 ± 0.003     | 0.301 ± 0.004     | 0.083 ± 0.004     | 0.427 ± 0.003     | 0.307 ± 0.003     | 0.146 ± 0.004     | 0.483 ± 0.003     | 0.354 ± 0.003     | 0.204 ± 0.004     |
> > > > | EigVal Laplacian | 0.375 ± 0.003     | 0.238 ± 0.004     | 0.080 ± 0.004     | 0.396 ± 0.003     | 0.269 ± 0.003     | 0.140 ± 0.004     | 0.458 ± 0.003     | 0.313 ± 0.003     | 0.204 ± 0.004     |
> > > > | SAR              | 0.416 ± 0.003     | 0.452 ± 0.003     | 0.074 ± 0.004     | 0.464 ± 0.003     | 0.436 ± 0.004     | 0.099 ± 0.004     | 0.482 ± 0.003     | 0.456 ± 0.003     | 0.142 ± 0.004     |
> > > > | MSP              | 0.409 ± 0.003     | 0.397 ± 0.004     | 0.322 ± 0.003     | 0.476 ± 0.003     | 0.384 ± 0.004     | 0.287 ± 0.003     | 0.476 ± 0.003     | 0.354 ± 0.003     | 0.201 ± 0.004     |
> > > > | CoCoA MSP        | 0.451 ± 0.003     | **0.518** ± 0.003 | 0.370 ± 0.003     | **0.511** ± 0.003 | **0.496** ± 0.003 | **0.332** ± 0.004 | 0.512 ± 0.003     | 0.507 ± 0.003     | **0.259** ± 0.004 |
> > > > | PPL              | 0.382 ± 0.003     | 0.383 ± 0.003     | 0.366 ± 0.003     | 0.428 ± 0.004     | 0.427 ± 0.004     | 0.205 ± 0.004     | 0.457 ± 0.003     | 0.449 ± 0.003     | 0.156 ± 0.004     |
> > > > | CoCoA PPL        | **0.456** ± 0.003 | 0.478 ± 0.003     | **0.382** ± 0.004 | 0.498 ± 0.003     | 0.471 ± 0.003     | 0.289 ± 0.004     | 0.523 ± 0.003     | **0.508** ± 0.003 | 0.231 ± 0.004     |
> > > > | MTE              | 0.354 ± 0.003     | 0.378 ± 0.004     | 0.354 ± 0.003     | 0.420 ± 0.004     | 0.438 ± 0.003     | 0.182 ± 0.004     | 0.456 ± 0.003     | 0.471 ± 0.003     | 0.154 ± 0.004     |
> > > > | CoCoA MTE        | 0.448 ± 0.003     | 0.475 ± 0.003     | 0.375 ± 0.004     | 0.495 ± 0.003     | 0.468 ± 0.003     | 0.291 ± 0.003     | **0.527** ± 0.003 | 0.508 ± 0.003     | 0.230 ± 0.004     |

---

> > > > > ### Author Response · Authors · 2025-08-02
> > > > >
> > > > > > Similarity function bottleneck: Results depend heavily on the semantic similarity...
> > > > >
> > > > > We agree that the choice of semantic similarity function plays an important  role in our framework and deserves careful consideration. In response to this, we would like to emphasize that we already have provided an empirical study of various similarity functions in Appendix C.1. The study considers:
> > > > >
> > > > > - CrossEncoder, a strong semantic similarity model adopted in prior work (https://aclanthology.org/2024.acl-long.276.pdf). It is specifically tailored for semantic similarity measurement and has demonstrated strong performance for textual similarity tasks (https://aclanthology.org/D19-1410.pdf)
> > > > >
> > > > > - NLI-based similarity, which has also been utilized in recent works (https://iclr.cc/virtual/2025/poster/31486)
> > > > >
> > > > > - Rouge L, which is a simple lexical measure.
> > > > >
> > > > > - AlignScore, which models information alignment via a transformer trained over diverse tasks.
> > > > >
> > > > > The results show that performance of method does not vary drastically across different similarity functions, indicating a relative robustness. We default to CrossEncoder due to its use in the literature and strong performance across tasks.
> > > > >
> > > > > When it comes to the question of which score should be used for an optimal performance, we observe that:
> > > > >
> > > > > - For shorter generations any well-trained CrossEncoder or NLI-based similarity model should suffice.
> > > > >
> > > > > - For longer outputs like document summarization, chunk-based measures like AlignScore can be a better alternative.
> > > > >
> > > > > Our ablation offers initial guidance, and future work could further systematize selection of a similarity function.
> > > > >
> > > > > > Also, I would like to request Reviewer zvVx to kindly provide some justification...
> > > > >
> > > > > We definitely leave the word to Reviewer zvVx, but within our group we have a simple and clear answer why CoCoA deserves publication in the current form: its empirical performance is extremely robust across models and tasks. Just recently we have extended the evaluation by:
> > > > >
> > > > > 1. Considering the larger model Gemma 3 12B: the results are consistently good for CoCoA-based approaches.
> > > > >
> > > > > | Method                              | QA        | ATS       | NMT       |
> > > > > |:------------------------------------|:----------|:----------|:----------|
> > > > > | CocoaMSP                            | 0.604     | _0.292_   | **0.632** |
> > > > > | CocoaMTE                            | _0.626_   | 0.288     | _0.628_   |
> > > > > | CocoaPPL                            | **0.636** | **0.296** | 0.627     |
> > > > > | DegMat                              | 0.503     | 0.095     | 0.312     |
> > > > > | EigVal Laplacian                    | 0.487     | 0.095     | 0.302     |
> > > > > | MSP          | 0.523     | 0.194     | 0.472     |
> > > > > | MTE                    | 0.525     | 0.189     | 0.593     |
> > > > > | MonteCarlo NSE | 0.446     | 0.044     | 0.451     |
> > > > > | MonteCarlo SE           | 0.480     | 0.012     | 0.340     |
> > > > > | PPL                          | 0.551     | 0.217     | 0.557     |
> > > > > | SAR                                 | 0.500     | 0.089     | 0.483     |
> > > > > | SemanticEntropy                     | 0.496     | 0.013     | 0.346     |
> > > > >
> > > > >
> > > > > 2. Claim-level uncertainty quantification similarly to the recent work: https://aclanthology.org/2024.findings-acl.558.pdf
> > > > >
> > > > > We combined their strong CCP method with consistency (estimated via claim Frequency score) within CoCoA and it gave a sizable performance benefits across several models.
> > > > >
> > > > > | UQ Method               | Gemma-3-12b-it PR-AUC | Gemma-3-12b-it PRR | Llama-3.1-8B PR-AUC | Llama-3.1-8B PRR | Llama-3.1-8B-Instruct PR-AUC | Llama-3.1-8B-Instruct PRR |
> > > > > |-------------------------|----------------------:|-------------------:|--------------------:|-----------------:|-------------------------------:|--------------------------:|
> > > > > | Maximum Probability     | 0.149 | 0.122 | 0.213 | 0.088 | 0.224 | 0.278 |
> > > > > | Perplexity              | 0.140 | 0.100 | 0.192 | 0.073 | 0.219 | 0.274 |
> > > > > | Max Token Entropy       | 0.164 | 0.155 | 0.184 | 0.049 | 0.253 | 0.296 |
> > > > > | P(True)                 | 0.196 | 0.119 | 0.233 | 0.135 | 0.240 | 0.326 |
> > > > > | CCP                     | 0.206 | 0.288 | 0.326 | 0.271 | 0.326 | 0.352 |
> > > > > | Attention Score         | 0.129 | 0.052 | 0.204 | 0.087 | 0.142 | 0.027 |
> > > > > | Focus                   | 0.161 | 0.145 | 0.199 | 0.082 | 0.226 | 0.279 |
> > > > > | Frequency Scoring       | 0.215 | 0.337 | 0.292 | 0.309 | 0.308 | 0.401 |
> > > > > | TokenSAR                | 0.153 | 0.134 | 0.211 | 0.089 | 0.222 | 0.279 |
> > > > > | CoCoA (Freq. × CCP)     | **0.230** | **0.341** | **0.365** | **0.350** | **0.395** | **0.449**
> > > > >
> > > > >
> > > > > Such a strong performance combined with simplicity of the method makes CoCoA a current default choice if one aims to maximize the UQ performance.
> > > > >
> > > > >
> > > > > **All things considered, we believe that we provided copious clarifications and conducted all additional investigations that were requested and some more, which confirmed outstanding robustness of empirical results. We thus kindly ask the reviewer to adjust their final score accordingly.**

---

> > > > ### Comment · Reviewer_aWz6 · 2025-08-05
> > > > **Reply to Authors**
> > > >
> > > > I would like to thank the authors again for their detailed rebuttal, here are my comments
> > > >
> > > > > I am not convinced but rather think MBR can be consider as a new angle of combing confidence and consistency score...
> > > >
> > > > I like this new story and think this is much better motivated that the current version, please consider revise the paper with this version.
> > > >
> > > > > Although I think theoretical formulation is missing..
> > > >
> > > > Agree on its empirical performance, but again I think this paper lack on the theoretical justification. A way to make this story better is to related to the motivation of how MBR is developed (see previous question).
> > > >
> > > > > I am not entirely convinced by this method of regression uq...
> > > >
> > > > I don't think the authors answered my question well, I am suggesting that in this [paper](https://arxiv.org/abs/2406.15627) regressing uq score with response quality does not seems to make sense other than binary categorical values. - which is related to but not directly refers to calibration.
> > > >
> > > > > Error bars missing: I am wondering why the variance is not reported in this paper...
> > > >
> > > > Thanks for the error bars, I am wondering how do you select the right scoring function if you do not have a hold out set? It seems like score influence the results quite a lot. (And are you sure these are significant?)
> > > >
> > > > > Similarity function bottleneck: Results depend heavily on the semantic similarity...
> > > >
> > > > Sounds fair, would have been better to explore more scoring functions though.
> > > >
> > > > > Reviewer zvVx
> > > >
> > > > I have almost everything I need from the authors, I think this a generally a good paper, but could be either 3 or 4. Still not sure why Reviewer zvVx give a score of 6. I would like to see his response before I update the final score. Thank you!

---

> > > > > ### Author Response · Authors · 2025-08-07
> > > > >
> > > > > >> I am not convinced but rather think MBR can be consider as a new angle of combing confidence and consistency score...
> > > > >    I like this new story and think this is much better motivated that the current version, please consider revise the paper with this version.
> > > > >
> > > > > We will surely revise the paper accordingly and thank the reviewer for bringing up this discussion.
> > > > >
> > > > > >> Although I think theoretical formulation is missing.
> > > > > > Agree on its empirical performance, but again I think this paper lack on the theoretical justification. A way to make this story better is to related to the motivation of how MBR is developed (see previous question).
> > > > >
> > > > > We will certainly add the updated theoretical discussion to the paper, if accepted. We want to underline, we provided a motivation for the particular choice of risk combination (see previous responses) which the reviewer has described as "making a lot of sense", and explained the key conceptual differences with current SOTA methods of UQ.
> > > > >
> > > > > > I don't think the authors answered my question well, I am suggesting that in this [paper](https://arxiv.org/abs/2406.15627) regressing uq score with response quality does not seems to make sense other than binary categorical values. - which is related to but not directly refers to calibration.
> > > > >
> > > > > Can you elaborate on why this approach is not suitable for continuous output scores? As we see it, it was proposed specifically to accommodate for both unbounded UE scores and continuous response correctness. The similar approach was concurrently proposed by another paper as well (https://aclanthology.org/2024.emnlp-main.18.pdf), thus we believe it is well-grounded in the recent literature on the subject.
> > > > >
> > > > > > Thanks for the error bars, I am wondering how do you select the right scoring function if you do not have a hold out set? It seems like score influence the results quite a lot. (And are you sure these are significant?)
> > > > >
> > > > > We selected the similarity score function based on its use in modern well-performing methods like https://arxiv.org/abs/2307.01379 as well as based on empirical performance. We stress that we use the same similarity function **across all experiments** (we do not change the function from task to task) and thus we did not need to use a held-out set specifically for this choice.
> > > > >
> > > > > > Sounds fair, would have been better to explore more scoring functions though.
> > > > >
> > > > > We have tested 4 scoring functions spanning simple lexical approach like Rouge-L, NLI, semantic similarity and factuality-based similarity functions. Can you elaborate on which kinds of similarity measurement that we missed, may best augment our existing results from your perspective?

---

> > > > > > ### Comment · Reviewer_aWz6 · 2025-08-08
> > > > > > **Reply to Authors**
> > > > > >
> > > > > > > Can you elaborate on why this approach is not suitable for continuous output scores? As we see it, it was proposed specifically to accommodate for both unbounded UE scores and continuous response correctness. The similar approach was concurrently proposed by another paper as well (https://aclanthology.org/2024.emnlp-main.18.pdf), thus we believe it is well-grounded in the recent literature on the subject.
> > > > > >
> > > > > > I have consulted with a colleague regarding this point, and I am satisfied that the method seem to be reasonable. I am happy to leave this point here.
> > > > > >
> > > > > > > Sounds fair, would have been better to explore more scoring functions though.
> > > > > >
> > > > > > I think using 4 different scoring functions is okay, but I just feel that expanding the range of scores could provide a more comprehensive perspective.
> > > > > >
> > > > > > To summarise, based on the rebuttal history, I will raise my score to 4. I believe this paper could pass the acceptance threshold with revisions.

---

> > > > > > > ### Author Response · Authors · 2025-08-09
> > > > > > >
> > > > > > > We are deeply grateful to the reviewer for the active engagement during the rebuttal process. The concerns that were raised helped us significantly widen our experimental setup and take a deeper look into the theoretical interpretation of the method we propose. All the additional results and theoretical considerations will be added to the camera-ready version of the paper, if accepted.

---

### Official Review · Reviewer_yPFe · 2025-07-03

**Clarity:** 3
**Significance:** 1
**Originality:** 3
**Rating:** 4
**Confidence:** 3

**Summary:**

This paper introduces CoCoA, which is a framework that generalizes several prior works efforts in direction of semantic clustering from a standpoint of uncertainty estimation as risk prediction.
The decomposition consists of Consistency and Confidence scores.
The former corresponds to the plurality of possible predictions, while the latter corresponds to confidence of the model in the 'main predicted' sequence.
The authors further propose a computationally cheaper version of the measure, which is achieved by learning an estimate for Consistency score.
The method shows competitive performance on a variety of tasks.

**Questions:**

1. Would introducing weighing for the two terms into the AdditiveCoCoA improve its performance for specific tasks?
2. How would the evaluation picture change if the PRR rejection cut-off was changed from it's 50% value?
3. What is the theoretical justification for multiplicative CoCoA combination rather than additive?

**Ethical Concerns:**

["NO or VERY MINOR ethics concerns only"]

**Final Justification:**

The paper was initially lacking some details and had a gap in between the theoretical and empirical part. The authors made some steps to bridge that gap during the rebuttal. I hope that all the promised material ends up being present in the final revision.

**Limitations:**

yes

**Quality:**

3

**Strengths And Weaknesses:**

#### Strengths
1. **Overall paper quality:** the paper is well structured and written.
2. **Originality:** CoCoA generalizes several prior works that deal with semantic invariances in uncertainty estimation for NLP. It advances the connections between uncertainty and prediction risks in the NLP domain.
3. **Evaluation:** the authors perform extensive evaluation on datasets and models standard to the subfield.

#### Weaknesses
1. **Introducing a Framework Rather than a Specifc Method**: as is acknowledged by the authors in the limitations, the different versions of CoCoA excel at different tasks.
An entire framework of possible options is therefore presented that can accommodate many prior methods and has multiple design choices.
The method empirically speaking is largely a combination of two families of prior work methods, each performing better in different scenarios.
This erodes the methods empirical utility.
2. **Treatment of risk combinations**: even though the authors attempt to derive their measure from a theoretically grounded standpoint of prediction risk, the resulting CoCoA measures are still quite arbitrary.
For instance, the authors do not provide a theoretical explanation for multiplicative combination of the confidence and consistency and instead settle for picking one with greater empirical performance.
If the Confidence and Consistency parts both correspond to some sort of risk, I would imagine them being additive rather than multiplicative.
3. **Evaluation Nuances**:
    - the authors discuss the advantages of PRR over AUROC (even though its not the focus of the paper), yet acknowledge that they have to cut the PRR evaluation at 50% rejection to avoid artifacts.
    - in appendix G table 16 shows a less confident picture of the CoCoA performance when using the AUROC+Judge instead of PRR + Rouge/AlignScore.

---

> ### Author Rebuttal · Authors · 2025-07-31
>
> We appreciate the reviewer’s feedback and address their concerns below.
>
> >Introducing a Framework Rather than a Specifc Method: as is acknowledged by the authors in the limitations, the different versions of CoCoA excel at different tasks. An entire framework of possible options is therefore presented that can accommodate many prior methods and has multiple design choices. The method empirically speaking is largely a combination of two families of prior work methods, each performing better in different scenarios. This erodes the methods empirical utility.
>
> While it is true that our method combines two families of prior approaches, this is not a weakness but an intentional design choice. In the general case, a maximum sequence probability (MSP)-based combination yields strong, reliable performance. The framework makes it flexible: when time and computational resources are available, it can be further optimized to better suit the specifics of a task. This makes it not only effective in general settings but also adaptable to specialized use cases. In this sense, we view the introduction of a framework not as an erosion of empirical utility, but as a practical strength - offering a solid default with room for task-specific refinement.
>
> >Treatment of risk combinations: even though the authors attempt to derive their measure from a theoretically grounded standpoint of prediction risk, the resulting CoCoA measures are still quite arbitrary. For instance, the authors do not provide a theoretical explanation for multiplicative combination of the confidence and consistency and instead settle for picking one with greater empirical performance. If the Confidence and Consistency parts both correspond to some sort of risk, I would imagine them being additive rather than multiplicative.
>
> In Appendix C.2, we compare our multiplicative formulation to additive and full-matrix consistency variants:
> Additive combination often allows one signal to dominate, especially since some information-theoretic confidence scores (e.g., entropy) are unbounded while consistency is bounded in [0, 1]. This imbalance undermines the contribution of consistency and practically requires proper empirical selection of relative contribution of each source. This selection would be task-dependent and would generalize poorly to the OOD settings.
> On the other hand, since consistency is bounded between 0 and 1, it serves as a natural scaling factor on confidence. This enforces mutual alignment between the two signals without requiring learned weights or tuning.
>
> While additive risk formulations are common in theory, in practice this setting involves differently scaled signals, and the multiplicative form has shown stronger empirical performance with a clear intuitive justification. That said, we agree that deeper theoretical analysis is a promising direction for future work.
>
> >Evaluation Nuances: the authors discuss the advantages of PRR over AUROC (even though its not the focus of the paper), yet acknowledge that they have to cut the PRR evaluation at 50% rejection to avoid artifacts.
>
> The choice to cut off at 50% was deliberate, as it rarely makes sense to reject all of the answers in practical settings. By focusing on the 50% most uncertain cases, we target the region where rejection is most relevant and actionable. This provides a more realistic and interpretable measure of performance in scenarios where selective prediction is actually useful.
>
> Furthermore, due to diminishing sample size, the expected quality of the output on the remaining dataset is a heteroscedastic value wrt rejection rate, and extreme rejection can lead to very noisy estimates.
>
> That being said, we report PRR up to 100% rejection here, and CoCoA-family methods retain their edge over the baselines even in this setup.
>
> | Metric | Llama QA | Llama NMT | Llama SUM | Mistral QA | Mistral NMT | Mistral SUM | Falcon QA | Falcon NMT | Falcon SUM |
> |---|---:|---:|---:|---:|---:|---:|---:|---:|---:|
> | MCSE | 0.486 | 0.439 | 0.031 | 0.550 | 0.424 | 0.006 | 0.613 | 0.440 | 0.284 |
> | MCNSE | 0.468 | 0.449 | 0.087 | 0.541 | 0.465 | 0.037 | 0.599 | 0.478 | 0.211 |
> | Semantic Entropy | 0.528 | 0.474 | 0.029 | 0.581 | 0.462 | 0.014 | 0.631 | 0.478 | 0.301 |
> | DegMat | 0.539 | 0.374 | 0.165 | 0.566 | 0.377 | 0.224 | 0.642 | 0.426 | 0.363 |
> | EigValLaplacian | 0.515 | 0.360 | 0.158 | 0.546 | 0.367 | 0.214 | 0.627 | 0.414 | 0.363 |
> | SAR | 0.578 | 0.501 | 0.155 | 0.603 | 0.503 | 0.170 | 0.667 | 0.508 | 0.269 |
> | Consistency | 0.586 | 0.427 | 0.105 | 0.604 | 0.426 | 0.144 | 0.656 | 0.450 | **0.383** |
> | MSP | 0.544 | 0.495 | **0.584** | 0.552 | 0.494 | **0.559** | 0.637 | 0.487 | 0.290 |
> | `CoCoA_MSP` | **0.607** ↑ | **0.575** ↑ | 0.560 ↑ | **0.623** ↑ | **0.575** ↑ | 0.530 ↑ | 0.688 ↑ | **0.582** ↑ | 0.382 ↑ |
> | Perplexity | 0.491 | 0.434 | 0.543 | 0.518 | 0.463 | 0.520 | 0.601 | 0.493 | 0.229 |
> | `CoCoA_Perplexity` | 0.591 ↑ | 0.497 ↑ | 0.538 ↑ | 0.602 ↑ | 0.518 ↑ | 0.535 ↑ | 0.691 ↑ | 0.534 ↑ | 0.344 ↑ |
> | MeanTokenEntropy | 0.450 | 0.417 | 0.537 | 0.507 | 0.452 | 0.506 | 0.592 | 0.492 | 0.216 |
> | `CoCoA_MeanTokenEntropy` | 0.586 ↑ | 0.486 ↑ | 0.528 ↑ | 0.606 ↑ | 0.506 ↑ | 0.520 ↑ | **0.693** ↑ | 0.526 ↑ | 0.345 ↑ |
>
> *Results for Evaluated Sequence – Greedy Sample: Mean PRR up to full rejection across datasets for each task. The best performing method is in bold, and the second-best is underscored. Arrows indicate improvement in CoCoA over the base version.*
>
> > Would introducing weighing for the two terms into the AdditiveCoCoA improve its performance for specific tasks?
>
> Yes, introducing learned or tuned weights could improve the performance of AdditiveCoCoA for specific tasks. However, adding weights would effectively turn the method into a supervised or task-specific approach, as these weights would need to be tuned on downstream validation sets to balance the contributions of each term. This tuning may yield better empirical performance but sacrifices the task-agnostic and unsupervised nature of the original CoCoA formulation.
> In contrast, our multiplicative formulation avoids this trade-off by enforcing mutual agreement between the two signals without requiring tuning or additional parameters.
>
> >How would the evaluation picture change if the PRR rejection cut-off was changed from it's 50% value?
>
> We report PRR up to 100% rejection here, and CoCoA-family methods retain their edge over the baselines even in this setup.
>
> >What is the theoretical justification for multiplicative CoCoA combination rather than additive?
>
> The key justification lies in how consistency and confidence are treated as signals of uncertainty within the Minimum Bayes Risk (MBR) framework. Consistency is bounded between 0 and 1, while confidence metrics such as perplexity (PPL) or mean token entropy (MTE) are unbounded. As a result, an additive combination would require careful supervision and tuning of weights to ensure a balanced contribution from each signal. Without proper tuning, one signal could dominate or be underrepresented, reducing the robustness of the final score. By contrast, the multiplicative combination used in CoCoA avoids this issue, as it inherently normalizes the interaction between signals, preserving their relative informativeness without requiring task-specific weight calibration.

---

> > ### Comment · Reviewer_yPFe · 2025-08-04
> >
> > I appreciate the authors' detailed rebuttal as it provides a wealth of valuable additional information.
> > At the same time, my key concerns regarding the interplay between the theoretical justification and applied performance remain:
> >
> > > The key justification lies in how consistency and confidence are treated as signals of uncertainty within the Minimum Bayes Risk (MBR) framework. Consistency is bounded between 0 and 1, while confidence metrics such as perplexity (PPL) or mean token entropy (MTE) are unbounded. As a result, an additive combination would require careful supervision and tuning of weights to ensure a balanced contribution from each signal. Without proper tuning, one signal could dominate or be underrepresented, reducing the robustness of the final score. By contrast, the multiplicative combination used in CoCoA avoids this issue, as it inherently normalizes the interaction between signals, preserving their relative informativeness without requiring task-specific weight calibration.
> >
> > > While additive risk formulations are common in theory, in practice this setting involves differently scaled signals, and the multiplicative form has shown stronger empirical performance with a clear intuitive justification. That said, we agree that deeper theoretical analysis is a promising direction for future work.
> >
> > Indeed, the rationale of balancing the signals from the two approaches is reasonable from an applied perspective.
> > However, this interferes with the framework's theoretical foundation: since the choices and adjustments to the formulation are made to suit empirical needs, where do we draw the boundary between the theoretical justification for combining risks and a purely empirical interpolation between the two families of methods?
> >
> > Regarding the CoCoA-MSP, the top-performing variant in the tables - does it use the logarithmic $U_{MSP}$ or just the likelihood?
> > In Appendix D, the authors specify the logarithmic variant as preferable (though there is a small factual error in line 961, as $-\log p(y \mid x)$ inverts the order compared to $p(y \mid x)$ because of the negative sign).
> > The two variables are also correlated, since, in the corner case, if e.g. MSP is 1, the consistency should also be 1.
> > Perhaps likelihood in range [0,1] and consistency in range [0,1] would lead to a better signal match in the additive setting (rather than log likelihood + log consistency ) ?
> > This needs additional exploration.
> >
> > At the same time, I do not fully understand why the difference in scales would be such a significant issue for the additive combination in the case of CoCoA-Light, since it already incorporates a small trained model whose output could be normalized to match the distribution of the confidence term.
> > For this reason, I have requested an interpolation analysis of the additive CoCoA, as it would presumably be relatively straightforward to compute and could make an important contribution to the analysis.
> >
> > I am convinced that these additional investigations are important to bridging the gap between the paper's otherwise sound theory and its empirical analysis.

---

> ### Author Response · Authors · 2025-08-07
>
> > Regarding the CoCoA-MSP, the top-performing variant in the tables - $U_{MSP}$ does it use the logarithmic  or just the likelihood?
>
> It uses logarithmic value for the confidence:  $U_{conf} = -\log(p(y \mid x))$.
>
> > though there is a small factual error in line 961, as $-\log(p(y\mid x))$ inverts the order compared to $p(y^*|x)$ because of the negative sign
>
> Indeed, but likelihood formulation uses $1 - p(y \mid x)$ as a $U_{conf}$ so the order is kept between logarithmic and likelihood formulations. We agree that this is confusing, as intuitively confidence implies that higher values represent higher confidence. Following [https://arxiv.org/abs/2305.19187] we use terms uncertainty/confidence to distinguish between types of uncertainty related to the generation process in general, and uncertainty of the particular output $y$. To reduce confusion in this matter, we should probably reverse the consistency term as well, to align the whole expression with the intuitive understanding of confidence. We will do this in the camera-ready version (if accepted), thank you for highlighting this issue.
>
> > Perhaps likelihood in range [0,1] and consistency in range [0,1] would lead to a better signal match in the additive setting (rather than log likelihood + log consistency )
>
> Thank you for this important observation. Here we present the results of additive formulation with likelihood instead of logarithm for confidence term:
>
> | Method                       | llama8b/QA | llama8b/NMT | llama8b/SUM |
> | :--------------------------- | :--------- | :---------- | :---------- |
> | CoCoA MSP                    | 0.451      | **0.519**   | 0.378       |
> | CoCoA PPL                    | 0.454      | 0.481       | **0.387**   |
> | CoCoA MTE                    | 0.447      | 0.478       | 0.380       |
> | Additive CoCoA MSP (1.0)     | 0.424      | 0.403       | 0.330       |
> | Additive CoCoA PPL (1.0)     | 0.455      | 0.467       | 0.368       |
> | Additive CoCoA MTE (1.0)      | 0.426        | 0.443         | 0.374         |
> | Additive ProbCoCoA MSP (1.0) | 0.449      | 0.475       | 0.035       |
> | Additive ProbCoCoA PPL (1.0) | **0.459**  | 0.476       | 0.343       |
>
> While additive formulation with normalized likelihood gets close to the the performance of multiplicative CoCoA MSP, its more unstable, and falls short on NMT and SUM. At the same time it requires confidence to be a bounded likelihood, reducing the generality of the multiplicative formulation, where confidence term can be of any nature. For example, using token entropy in additive setting catastrophically reduces performance, while multiplicative formulation retains performance close to other choices of the confidence term.
>
> > At the same time, I do not fully understand why the difference in scales would be such a significant issue for the additive combination in the case of CoCoA-Light, since it already incorporates a small trained model whose output could be normalized to match the distribution of the confidence term.
>
> Indeed, but the distribution of the confidence term can be task-specific. For example,  generations of different lengths have vastly different likelihoods. Learning this distribution would make the method less generally applicable. For example here we provide an additive formulation with a scaling factor for consistency term selected for best performance on QA tasks:
>
> | Method                        | llama8b/QA   | llama8b/NMT   | llama8b/SUM   |
> |:------------------------------|:-------------|:--------------|:--------------|
> | CoCoA MSP                     | 0.451        | **0.519**         | 0.378         |
> | CoCoA PPL                     | 0.454        | 0.481         | **0.387**     |
> | CoCoA MTE                     | 0.447        | 0.478         | 0.380         |
> | Additive CoCoA MSP (2.0)      | 0.429        | 0.408         | 0.332         |
> | Additive CoCoA PPL (1.0)      | 0.455        | 0.467         | 0.368         |
> | Additive CoCoA MTE (50.0)     | 0.439        | 0.432         | 0.055         |
> | Additive ProbCoCoA MSP (0.55) | 0.447        | 0.487         | 0.044         |
> | Additive ProbCoCoA PPL (0.9)  | **0.458**        | 0.474         | 0.352         |
>
> While having strong performance in-domain (QA), it falls short on summarization.

---

> > ### Author Response · Authors · 2025-08-07
> >
> > We also have tried an additive formulation of the form $r(y, y' \mid x) = u(y \mid x) + u(y \mid x)(1 - s(y,y'))$, where $u(y \mid x)$ in the second term acts like an adaptive scaling factor for the consistency:
> >
> > | Method                        | llama8b/QA   | llama8b/NMT   | llama8b/SUM   |
> > |:------------------------------|:-------------|:--------------|:--------------|
> > | CoCoA MSP                     | 0.451        | **0.519**     | 0.378         |
> > | CoCoA PPL                     | **0.454**    | 0.481         | **0.387**     |
> > | CoCoA MTE                     | 0.447        | 0.478         | 0.380         |
> > | Adaptive CoCoA MSP            | 0.430        | 0.429         | 0.353         |
> > | Adaptive CoCoA PPL            | 0.418        | 0.424         | 0.384         |
> > | Adaptive CoCoA MTE            | 0.409        | 0.430         | 0.376         |
> > | Adaptive ProbCoCoA MSP        | 0.449        | 0.478         | 0.041         |
> > | Adaptive ProbCoCoA PPL        | 0.423        | 0.430         | **0.387**     |
> >
> > Sadly, this form of additive risk falls short of the multiplicative CoCoA variations as well.

---

> > > ### Comment · Reviewer_yPFe · 2025-08-07
> > >
> > > Thank you for providing the additional evaluation. Truly quite a few things to consider.
> > > The latest table represents, basically, an interpolation between the multiplicative CoCoA and a Confidence Score?
> > >
> > > With respect to the rest of the tables, it does appear that the Additive CoCoA works best with the length normalized confidence metrics, i.e. PPL (also not sure what is  ProbCoCoA PPL, since PPL is a log space quantity unlike MSP), which makes sense in the context of stability.
> > >
> > > These are all interesting details and observations, and I hope that the authors will discuss those in their paper.

---

> > > > ### Author Response · Authors · 2025-08-07
> > > >
> > > > > The latest table represents, basically, an interpolation between the multiplicative CoCoA and a Confidence Score?
> > > >
> > > > It can be viewed in that way, yes. We completely agree that an additive formulation of risk would be more grounded in theory, and keeping in mind the issue of different levels of signal in the confidence and consistency terms, we tried to used the confidence term itself to serve as an adaptive scaling factor on consistency. This allows to disentangle the signal from the confidence term from the signal from the consistency term. However, the empirical results still clearly show that multiplicative risk is the one to go for optimal performance across different tasks.
> > > >
> > > > > also not sure what is ProbCoCoA PPL, since PPL is a log space quantity unlike MSP
> > > >
> > > > ProbCoCoA PPL just uses the exponent of the log-perplexity. Indeed it's not really a probability, as it can easily be larger than 1. We added that to cover as much ground as possible with this ablation.
> > > >
> > > > > These are all interesting details and observations, and I hope that the authors will discuss those in their paper.
> > > >
> > > > We will surely expand our ablations with the results obtained during this discussions and augment our theoretical sections with some of the thoughts that were discussed here as well, all in the camera-ready version, if accepted.
> > > >
> > > > We deeply thank the reviewer for the push towards additional empirical confirmation of the method and expanding the ablation substantially. We believe that we have exaustively answered reviewer's concerns during the discussion and kindly ask to revise the final score.

---

### Official Review · Reviewer_Awk2 · 2025-07-03

**Clarity:** 2
**Significance:** 2
**Originality:** 2
**Rating:** 4
**Confidence:** 4

**Summary:**

This paper introduces CoCoA, a new framework for uncertainty quantification (UQ) in large language models. The method combines two key signals: the model's internal confidence (derived from token probabilities) and the semantic consistency of its outputs across multiple sampled generations. The authors frame this combination within a minimum Bayes risk framework. To address the computational cost of multiple generations, a more efficient variant, CoCoA Light, is proposed, which uses a lightweight predictor to estimate semantic inconsistency. The authors conduct experiments on various tasks, including question answering, summarization, and machine translation, showing that CoCoA outperforms existing UQ baselines in failure prediction.

**Questions:**

1.  How would you adapt the CoCoA framework for use with closed-source, black-box models (e.g., via the GPT-4 API) where direct access to token probabilities is not available?
2.  The paper omits comparisons with verbalized confidence methods. Could you elaborate on the potential advantages or disadvantages of CoCoA relative to this line of work?
3.  While the experiments show that a multiplicative combination works well, have you explored other aggregation functions (e.g., attention-based weighting)?
4.  How the reasoning-models like QwQ-32B perform on the CoCoA framework?

**Ethical Concerns:**

["NO or VERY MINOR ethics concerns only"]

**Final Justification:**

My concerns are addressed.

**Limitations:**

yes

**Paper Formatting Concerns:**

nan

**Quality:**

3

**Strengths And Weaknesses:**

**Strengths:**

1.  The paper provides a clear and well-grounded framework that unifies two important sources of uncertainty—model confidence and output consistency. The connection to the minimum Bayes risk framework provides a solid theoretical foundation.
2.  The experiments are comprehensive, covering multiple tasks, datasets, and models. CoCoA demonstrates significant and consistent improvements over a variety of established UQ baselines on failure detection metrics like PRR and AUROC. And the CoCoA framework is presented as a plug-in module that can enhance different information-based uncertainty estimators, highlighting its flexibility and potential for broad applicability.

**Weaknesses:**

1.  **Limited Novelty:** The core idea of combining confidence and consistency feels somewhat incremental. The paper could be strengthened by a deeper investigation into the relationship and interaction between these two types of uncertainty, beyond the proposed multiplicative combination. The current insight into *why* this combination is so effective is not fully explored.
2.  **Missing Baselines:** The evaluation does not include comparisons with "verbalized confidence" methods[1]. [1] demonstrates that the logits of the output of LLMs suffers from over-confident issue due to RLHF process. [2] proposes sampling and consistency-based confidence calibration methods to measure the intrinsic uncertainty of LLMs. These methods, which prompt the model to express its own uncertainty, are a relevant and important class of UQ techniques and would serve as a crucial point of comparison.
3.  **Incomplete Evaluation Metrics:** The evaluation relies primarily on failure detection metrics (PRR, AUROC). While important, these do not measure confidence calibration. Including metrics like the Expected Calibration Error (ECE) would provide a more complete picture of the UQ quality.
4.  **Limited Model Scale:** The experiments are conducted on models up to 8B parameters. The claims would be more convincing if the method's effectiveness were also demonstrated on larger, state-of-the-art models (e.g., Llama 3 70B, even with quantization).

[1] Katherine Tian, Eric Mitchell, Allan Zhou, Archit Sharma, Rafael Rafailov, Huaxiu Yao, Chelsea Finn, and Christopher D Manning. Just ask for calibration: Strategies for eliciting calibrated confidence scores from language models fine-tuned with human feedback. In EMNLP, 2023.

[2] Miao Xiong, Zhiyuan Hu, Xinyang Lu, Yifei Li, Jie Fu, Junxian He, and Bryan Hooi. Can LLMs Express Their Uncertainty? An Empirical Evaluation of Confidence Elicitation in LLMs. In ICLR, 2023.

---

> ### Author Rebuttal · Authors · 2025-07-31
>
> We thank the reviewer for the insightful comments. Below, we would like to address the concerns raised by reviewer:
>
> >Limited Novelty: The core idea of combining confidence and consistency feels somewhat incremental. The paper could be strengthened by a deeper investigation into the relationship and interaction between these two types of uncertainty, beyond the proposed multiplicative combination. The current insight into why this combination is so effective is not fully explored.
>
>
> **On limited novelty**:
> While our work builds upon established concepts of confidence and consistency, our contribution is threefold.
>
> (1) We focus on selective generation, where the decision to output a response is based on the confidence score of a specific candidate, rather than modeling uncertainty over a potential input. Specifically, we frame our approach as a variant of Minimum Bayes Risk Decoding, which defines a utility function that naturally favors responses that are both confident (according to the model) and consistent (with other sampled candidates).
>
> (2) We show that our MBR-based framework is flexible and performs well across various uncertainty measures.
>
> (3) We demonstrate that the consistency component can be learned, significantly reducing the computational cost associated with extensive sampling.
>
> **On discussing potential reasons for the best performance of multiplicative formulation**:
>
> We agree that more discussion in the combination strategy is valuable. In Appendix C.2, we compare our multiplicative approach to additive and full-matrix variants. The additive form often lets one signal dominate (e.g., consistency gets overwhelmed by high-entropy outputs). Full-matrix averaging dilutes pairwise relationships by incorporating irrelevant comparisons. The multiplicative form works best because it naturally favours an output if it is both confident and semantically consistent. This enforces alignment between both criteria without introducing extra parameters or reweighting.
>
> > Missing Baselines: The evaluation does not include comparisons with "verbalized confidence" methods[1]
>
> We do not focus on verbalized uncertainty methods, as prior work (https://openreview.net/pdf?id=gjeQKFxFpZ, https://aclanthology.org/2025.tacl-1.11/  ) has shown that LLMs tend to express high confidence regardless of actual correctness - particularly in smaller models, like 7-8b models used in our experiments. In such models these methods often yield poorly calibrated scores that do not reflect true reliability.
>
> > Incomplete Evaluation Metrics: The evaluation relies primarily on failure detection metrics (PRR, AUROC). While important, these do not measure confidence calibration. Including metrics like the Expected Calibration Error (ECE) would provide a more complete picture of the UQ quality.
>
> While calibration is an important aspect of uncertainty quantification (UQ), it assumes that uncertainty scores are interpretable as probabilities and that output correctness can be measured in binary terms. Neither of these assumptions holds for much of our experimental setup. Most UQ methods produce unbounded scores that lack a probabilistic interpretation, and many of the quality metrics we use - particularly in tasks like neural machine translation (NMT) and summarization - are continuous rather than binary. As a result, standard calibration metrics such as Expected Calibration Error (ECE) are not directly applicable.
>
> However, we adopted an approach to calibration proposed in (https://aclanthology.org/2025.tacl-1.11/) and fitted an isotonic regression to pairs of raw UQ scores and response quality values, and report here MSE between ground truth quality and prediction of the regression here:
>
> | Metric | Llama QA | Llama NMT | Llama SUM | Mistral QA | Mistral NMT | Mistral SUM | Falcon QA | Falcon NMT | Falcon SUM |
> |:--|:--:|:--:|:--:|:--:|:--:|:--:|:--:|:--:|:--:|
> | MCSE | 0.177 | 0.005 | 0.074 | 0.159 | **0.004** | 0.099 | 0.140 | 0.005 | 0.059 |
> | MCNSE | 0.179 | 0.005 | 0.074 | 0.160 | **0.004** | 0.098 | 0.140 | **0.004** | 0.059 |
> | Semantic Entropy | 0.172 | 0.005 | 0.074 | 0.154 | **0.004** | 0.100 | 0.136 | 0.005 | 0.059 |
> | DegMat | 0.167 | 0.005 | 0.073 | 0.153 | **0.004** | 0.097 | 0.132 | 0.005 | 0.059 |
> | EigValLaplacian | 0.171 | 0.005 | 0.073 | 0.156 | **0.004** | 0.097 | 0.135 | 0.005 | 0.058 |
> | SAR | 0.164 | **0.004** | 0.073 | 0.150 | **0.004** | 0.098 | 0.131 | **0.004** | 0.059 |
> | Dissimilarity | 0.161 | **0.004** | 0.074 | 0.149 | **0.004** | 0.098 | 0.130 | **0.004** | 0.057 |
> | MSP | 0.165 | **0.004** | 0.066 | 0.151 | **0.004** | **0.088** | 0.131 | 0.005 | 0.058 |
> | CoCoA_MSP | **0.158** ↑ | **0.004** ↑ | **0.065** ↑ | **0.147** ↑ | **0.004** ↑ | **0.088** ↑ | 0.127 ↑ | **0.004** ↑ | **0.056** ↑ |
> | Perplexity | 0.170 | 0.005 | 0.066 | 0.155 | **0.004** | 0.089 | 0.135 | **0.004** | 0.058 |
> | CoCoA_Perplexity | 0.160 ↑ | **0.004** ↑ | **0.065** ↑ | 0.148 ↑ | **0.004** ↑ | 0.089 ↑ | 0.126 ↑ | **0.004** ↑ | 0.057 ↑ |
> | MeanTokenEntropy | 0.175 | 0.005 | 0.067 | 0.156 | **0.004** | **0.088** | 0.135 | **0.004** | 0.058 |
> | CoCoA_MeanTokenEntropy | 0.161 ↑ | **0.004** ↑ | **0.065** ↑ | 0.148 ↑ | **0.004** ↑ | 0.090 ↑ | **0.125** ↑ | **0.004** ↑ | 0.057 ↑ |
>
>
> *Results for Evaluated Sequence – Greedy Sample: Mean MSE between quality and calibrated UQ score across datasets for each task, lower values mean better calibration. The best performing method is in bold, and the second-best is underlined. Arrows indicate improvement in CoCoA over the base version.*
>
>
> CoCoA family methods still retain their edge over the baselines in most of the task-model pairs in this setting.
>
> >Limited Model Scale: The experiments are conducted on models up to 8B parameters. The claims would be more convincing if the method's effectiveness were also demonstrated on larger, state-of-the-art models (e.g., Llama 3 70B, even with quantization).
>
> We appreciate the reviewer’s point regarding model size. Due to limited compute resources, our experiments focus on models up to 8B parameters (Similar to https://aclanthology.org/2025.tacl-1.11/, https://neurips.cc/virtual/2024/poster/97746, https://arxiv.org/abs/2503.05318, https://arxiv.org/abs/2406.04306 ).
>
> Our method does not rely on model-specific tuning and is compatible with quantized or cached generation setups, making it feasible to apply to larger models in future work. We agree that testing on 70B-scale models like LLaMA 3 would further strengthen our claims, and we plan to explore this direction in the future.
>
> >How would you adapt the CoCoA framework for use with closed-source, black-box models (e.g., via the GPT-4 API) where direct access to token probabilities is not available?
>
> While CoCoA relies on confidence estimates such as token-level probabilities, adaptation to black-box models like GPT-4 is still feasible:
> GPT-4 and other APIs expose log probabilities for generated outputs. This allows the computation of confidence-based metrics like Maximum Sequence Probability (MSP).
> In cases where logprobs are unavailable, they can be estimated empirically from the sampled responses, or verbalized confidence can be used instead, which should perform well enough for larger models which are often used in a black-box setting.
>
> >The paper omits comparisons with verbalized confidence methods. Could you elaborate on the potential advantages or disadvantages of CoCoA relative to this line of work?
>
> We do not focus on verbalized uncertainty methods, as prior work (https://openreview.net/pdf?id=gjeQKFxFpZ, https://aclanthology.org/2025.tacl-1.11/) has shown that LLMs tend to express high confidence regardless of actual correctness - particularly in smaller models. In such models these methods often yield poorly calibrated scores that do not reflect true reliability.
>
> > While the experiments show that a multiplicative combination works well, have you explored other aggregation functions (e.g., attention-based weighting)?
>
> In Appendix C.2, we evaluated alternatives to the multiplicative combination-including additive scoring and full-sample averaging. The underperformance of these alternatives stems from their inability to balance both confidence and consistency signals effectively.
>
> While we did not experiment with attention‑based weighting of token logprobs (if that’s what is being referred to in the question), we stress that any confidence measure can be used in the CoCoA (CCP, TokenSAR, attention-based re-weighting, etc). We show that even using very basic approaches like perplexity and sequence probability can improve over the baseline methods.
>
> >How the reasoning-models like QwQ-32B perform on the CoCoA framework?
>
> We have not evaluated CoCoA on reasoning models such as QwQ-32B, so we cannot make claims about their performance. However, if a long reasoning chain is considered as a part of the output, we hypothesize that more complicated measures of confidence and consistency are required. The reason for that is that over long sequences, simple aggregation of log probabilities can be drowned in noise and simple NLI similarity is not well-defined. In general, uncertainty quantification for reasoning setup and long-form outputs in general requires special considerations that we are planning to tackle in further work on CoCoA, but which are out of scope for this particular submission.

---

> > ### Comment · Reviewer_Awk2 · 2025-08-05
> >
> > I appreciate the authors' detailed rebuttal, which has addressed many of my concerns. However, some points remain insufficiently convincing:
> >
> > **Lack of verbalized confidence baselines**: If verbalized confidence methods do not perform well on smaller models, supporting evidence should be provided. Additionally, considering my previous question regarding larger models, a comparison with models of increased size would be valuable, where the verbalized confidence baselines may be good.
> >
> > **Handling unbounded uncertainty values**: While applying a softmax function might seem straightforward to address unbounded uncertainty values, in real-world scenarios requiring an **explicit** confidence/uncertainty value within a [0,100%] range for decision-making, reporting the Expected Calibration Error (ECE) metric becomes important.
> >
> > Therefore, I choose to maintain my current borderline score.

---

> > > ### Author Response · Authors · 2025-08-07
> > >
> > > We sincerely appreciate your engagement and your acknowledgment that our rebuttal addressed many of your concerns. We would like to further clarify the remaining points you raised.
> > >
> > > > **Lack of verbalized confidence baselines**
> > >
> > > Several studies have examined how model size affects the ability to express verbalized uncertainty, with Kadavath et al. [1] and Xiong et al. [2] both finding that calibration and failure-prediction performance improve as model size increases and smaller models struggle with a task. Vashurin et al. [3] benchmark several verbalized-confidence methods and report that their performance for 7–8 B models is significantly worse than baselines like MSP or PPL.
> > >
> > > To address concerns about model size in relation to verbalized confidence baselines, we included P(True) [1] among the baselines for the new experiments with larger Gemma-12B model.
> > > Unfortunately, we did not have the capacity to compare CoCoA with verbalized methods across all settings explored in the paper, the results for this configuration are as follows:
> > >
> > >
> > > | Method          | qa        | ats       | nmt       |
> > > | :-------------- | :-------- | :-------- | :-------- |
> > > | CocoaMSP        | 0.604     | _0.292_   | **0.632** |
> > > | CocoaMTE        | _0.626_   | 0.288     | _0.628_   |
> > > | CocoaPPL        | **0.636** | **0.296** | 0.627     |
> > > | DegMat          | 0.503     | 0.095     | 0.312     |
> > > | EigValLaplacian | 0.487     | 0.095     | 0.302     |
> > > | MSP             | 0.523     | 0.194     | 0.472     |
> > > | MTE             | 0.525     | 0.189     | 0.593     |
> > > | MC-NSE          | 0.446     | 0.044     | 0.451     |
> > > | MC-SE           | 0.480     | 0.012     | 0.340     |
> > > | PTrue           | 0.133     | 0.056     | 0.032     |
> > > | PPL             | 0.551     | 0.217     | 0.557     |
> > > | SAR             | 0.500     | 0.089     | 0.483     |
> > > | SemanticEntropy | 0.496     | 0.013     | 0.346     |
> > >
> > > As can be seen, (1) Cocoa methods perform well, consistent with results from smaller models. (2) The evaluated verbalized uncertainty method demonstrates weaker performance.
> > >
> > > > real-world scenarios requiring an **explicit** confidence/uncertainty value within a [0,100%] range for decision-making
> > >
> > > We agree that in real-world decision-making scenarios, providing scores in an interpretable range (0–100%) is indeed important. We emphasize that the isotonic regression–calibrated scores presented in our rebuttal naturally fall within this range (they do transform the scores into the same range as the quality score, usually 0–100%) and can therefore be directly communicated to end users.
> > >
> > > However, it is important to note that many existing SOTA-methods like Semantic Entropy or SAR, also provide scores that are not bounded to a specific range. In selective generation tasks, the primary concern is the method's performance on the selection objective itself; scores can subsequently be scaled or fitted to a desired range for presentation to the end user.
> > >
> > > When it comes to ECE , while we agree that calibration is an important topic, our focus in this work is on error detection, and we therefore consider calibration to be somewhat out of scope for this submission-hence our original choice of evaluation metrics. It is worth noting that the quality of probabilistic forecasts can be decomposed into calibration and sharpness [4], making calibration only one component of prediction error. Even perfectly calibrated models can perform poorly at prediction, which is why we treat calibration as a secondary metric in our study.
> > >
> > >
> > > Traditional calibration metrics like ECE are, since in all but two tasks, evaluation is based on continuous quality scores rather than binary outcomes.
> > > Thus at the request of the reviewers, we used a known approach (see [5], [6]) to calibrate unbounded scores w.r.t. to expected output quality which we provided in our response. Specifically, we fit an isotonic regression model to map raw scores to observed output quality, and then report the mean squared error (MSE) between the calibrated scores (ranges from 0 to 1) and the actual quality of the text (ranges from 0 1). These allows us to evaluate how close are the UE scores to the actual quality of the text, which is a more relevant metric for our task than traditional ECE.
> > >
> > > [1] https://arxiv.org/abs/2207.05221
> > >
> > > [2] https://openreview.net/forum?id=gjeQKFxFpZ
> > >
> > > [3] https://aclanthology.org/2025.tacl-1.11/
> > >
> > > [4] https://sites.stat.washington.edu/raftery/Research/PDF/Gneiting2007jrssb.pdf
> > >
> > > [5] https://direct.mit.edu/tacl/article/doi/10.1162/tacl_a_00737/128713/Benchmarking-Uncertainty-Quantification-Methods
> > >
> > > [6] https://aclanthology.org/2024.emnlp-main.18.pdf
> > >
> > > We hope that these clarifications help resolve the remaining concerns. Given the additional context and evidence provided, we respectfully ask you to reconsider your borderline evaluation.

---

> > > > ### Comment · Reviewer_Awk2 · 2025-08-07
> > > >
> > > > Thanks for the authors' response. Most of my concerns are addressed. I will accordingly raise my score.

---

### Official Review · Reviewer_zvVx · 2025-07-05

**Clarity:** 4
**Significance:** 4
**Originality:** 4
**Rating:** 6
**Confidence:** 4

**Summary:**

The authors present a new framework called CoCoA for uncertainty quantification (UQ) of large language models (LLMs). The method is based on minimum Bayes risk (MBR). They start by showing how many existing “confidence” (information-based) and “consistency” (sampling-based) scores are different approximations of the same MBR objective.

Then they introduce CoCoA - a method for MBR-derived scores by multiplying a sequence-level confidence term with semantic-inconsistency term. They also present CoCoA-Light, that learns to predict the consistency term from the model’s own hidden states, so it needs only a single greedy decode at test time. This can avoid the requirement to generate multiple samples for UQ.

The authors conduct experiments on 3 tasks (question answering, abstractive summarisation and machine translation) with 3 open-weight 7-8 B models (mistral, llama, falcon) and show that CoCoA and CoCoA-Light do better than baselines across PRR, AUROC and other metrics.

Overall this is a very good submission.

**Questions:**

1. How will you use this approach for large black box models provided by the likes of OpenAI, Claude and Gemini. We can generate consistency scores. Can the confidence be approximated somewhat similar to what you have tried with CoCoA light? Assuming it is allowed under their terms, can small models be learned from such large black box models to provide confidence scores and used in this CoCoA setting?

2. In Figure 1, you have illustrated the problem with model confidence. You tend to alleviate this problem by using it together with consistence measures. But is it possible that if the models are large and the generated text is small, the consistency based methods will also miss the variations in the possible outputs?

What I mean is most of the samples look like romeo and juliet, and very few look like romeo & juliet or romeo and juliet,. This might happen if the samples are few and based on temperature sampling. In such a scenario how does MBR or CoCoA help?

**Ethical Concerns:**

["NO or VERY MINOR ethics concerns only"]

**Final Justification:**

I see that other reviewers have not considered this work as such a strong submission like I have. But I went through the discussions and I'm quite familiar with recent literature on this topic. I think the main framework formalizing the combining of consistency and confidence methods and the Cocoa lite variant are very strong contributions of this work. Hence I'm comfortable keeping my score of 6.

**Limitations:**

Yes, the authors have listed 3 limitations which are quite thoughtful.

**Quality:**

4

**Strengths And Weaknesses:**

Strengths:

S1.	The paper presents a clear theoretical unification.  While [10] had introduced uncertainty aware MBR, this paper re-derives disparate UQ scores as MBR risks and clarifies when and why they work.

S2.	The solution presented seems relatively simple but effective metric.  The components of CoCoA are already familiar to the research community. So this method seems relatively easy to implement. I assume these methods are available in LM-Polygraph library, which the authors also seems to have used.

S3.	CoCoA-Light is very interesting. One of the main complaints about sampling techniques is the cost. Hence the solution provided in this paper to approximate using two MLPs and hence avoid sampling is very interesting.

S4.	The paper does a very good job with the experiments. The authors experiment on 3 tasks (QA, summarization and translation), use 3 small sized models, use PRR besides other metrics. They also present useful ablations. Reporting AUROC in addition to PRR and performing experiments with other metrics (for PRR) shows thoroughness of experiments.

S5.	The results of their experiments support the claims of the paper. CoCoA ppl performing better than baseline perplexity etc gives lot of confidence in this proposed approach.

Weaknesses:

W1.	There is some dependency on an external similarity model. They mention task and domain dependency in the limitations section. I wonder if the length of the generated text is also crucial. I might have missed it, but it would be good to comment on the effect of length of the generated output on CoCoA.

W2.	QA (especially long form), summarization and machine translation are somewhat similar tasks. Adding GSM8k is useful. It would be good to see how these methods work on code generation tasks.

W3.	More details on the CoCoA light training will be useful. What is the size of the training set?

---

> ### Author Rebuttal · Authors · 2025-07-31
>
> We thank the reviewer for their thoughtful comments and the time they invested. Please find below our responses to the concerns raised.
>
> >W1. There is some dependency on an external similarity model. They mention task and domain dependency in the limitations section. I wonder if the length of the generated text is also crucial. I might have missed it, but it would be good to comment on the effect of length of the generated output on CoCoA.
>
> We absolutely agree that expected length of the generated sequence should be a major factor when selecting both particular confidence and consistency estimates. One has to be mindful whether this length falls into the domain that the similarity model was trained on. Most modern similarity models are trained on rather short sequences, so longer outputs might require special handling. On the other hand, from the confidence standpoint, some sort of importance weighting for individual token contribution might be needed.
>
> >W2. QA (especially long form), summarization and machine translation are somewhat similar tasks. Adding GSM8k is useful. It would be good to see how these methods work on code generation tasks.
>
> We fully agree that code generation is an exciting domain to test UQ capabilities of LLMs. It would require a careful selection of the similarity measure for consistency estimation so we consider this to be out of scope for this submission. However, we definitely will pursue this in our future work.
>
> >W3. More details on the CoCoA light training will be useful. What is the size of the training set?
>
> Training set sizes:
>
>
> | Dataset        | LLaMA | Falcon | Mistral |
> |----------------|-------|--------|---------|
> | CoQA           | 10,000 | 10,000 | 10,000  |
> | GSM8K          | 3,000  | 3,000  | 2,500   |
> | MMLU           | 1,461  | 1,461  | 1,461   |
> | TriviaQA       | 10,000 | 10,000 | 10,000  |
> | WMT14 Fr-En    | 6,000  | 6,000  | 6,000   |
> | WMT19 De-En    | 6,000  | 6,000  | 6,000   |
> | XSum           | 7,500  | 5,000  | 6,500   |
>
> The average training set among datasets was around 5k examples. The architecture used was a simple MLP with one hidden layer. We have observed that optimal behavior was reached after only a couple of epochs, so probably the train set size can be further reduced with longer train time. We also highlight that these training examples don’t need to be labelled for correctness, only consistency among samples should be computed for each.
>
> >How will you use this approach for large black box models provided by the likes of OpenAI, Claude and Gemini. We can generate consistency scores. Can the confidence be approximated somewhat similar to what you have tried with CoCoA light? Assuming it is allowed under their terms, can small models be learned from such large black box models to provide confidence scores and used in this CoCoA setting?
>
> Some black box models give probabilities for top-1 tokens, which allows us to estimate confidence directly. For models which do not expose such information in their API, one could use empirical probability estimation from the relative frequencies of the outputs in the samples, or opt for verbalized approaches for confidence estimation, which are calibrated better for large models.
>
> >In Figure 1, you have illustrated the problem with model confidence. You tend to alleviate this problem by using it together with consistence measures. But is it possible that if the models are large and the generated text is small, the consistency based methods will also miss the variations in the possible outputs?
> What I mean is most of the samples look like romeo and juliet, and very few look like romeo & juliet or romeo and juliet,. This might happen if the samples are few and based on temperature sampling. In such a scenario how does MBR or CoCoA help?
>
> https://arxiv.org/abs/2406.15627 shows that for very large models, pure consistency estimation methods still perform well. This suggests that CoCoA would be a strong choice in such settings as well. We do agree, however, that the artifacts highlighted in Figure 1 are more prevalent in the outputs of smaller models. Conducting benchmarking with larger models would indeed be a valuable direction for future research.

---

> > ### Comment · Reviewer_zvVx · 2025-08-06
> > **Thanks for these additional details**
> >
> > Like I mentioned in the review as well, I think this is a strong submission. Thanks for these additional details. I'll follow the discussions in other reviews and pitch in if relevant.

---

### Note · Authors · 2025-08-13

Dear Area Chair and Reviewers,

We appreciate reviewers constructive feedback and have addressed all points in the rebuttal; the additional clarifications and analyses suggested can be readily incorporated into the camera-ready version. Below, we summarize the strengths noted and our resolutions to key concerns.

Strengths highlighted by reviewers:
- Unified confidence-consistency framework for UQ: MBR-based formulation integrating model confidence and output consistency into a single multiplicative term [zvVx, aWz6, yPFe, Awk2].
- Strong empirical performance and evaluation: Consistent gains across QA, summarization, and translation, with comprehensive metrics, baselines, and ablations [zvVx, Awk2, yPFe, aWz6].
- Practical approach: CoCoA-Light reduces sampling cost via representation-based approximation while remaining simple to implement [zvVx, aWz6].

Resolutions to key concerns:

>Similarity function selection [zvVx, aWz6]:

Performance robust across functions (Appendix C.1.); default to CrossEncoder for strong results and prevalence. Discussion of selection will be expanded in the camera-ready version.

>Calibration performance [aWz6, Awk2]:

We report MSE between quality and calibrated UQ scores via isotonic regression; reviewers agreed with this choice. Results will be included in the camera-ready version.

>Multiplicative formulation justification [yPFe, aWz6, Awk2]:

Multiplicative variant outperformed additive and full-matrix (Appendix C.2); further discussion and theoretical motivation will be added.

We will also add a verbalized baseline and larger model results (Awk2, aWz6), CoCoA-Light training details (zvVx), and an expanded PRR metric discussion (yPFe).

We thank the Area Chair and reviewers for their thoughtful feedback and engagement, which will strengthen the final version of this work.

---

### Decision · Program_Chairs · 2025-09-17

**Decision:**

Accept (poster)

**Comment:**

# Summary
This paper introduces CoCoA, a new framework for uncertainty quantification (UQ) in large language models. The method combines two key signals: the model's internal confidence (derived from token probabilities) and the semantic consistency of its outputs across multiple sampled generations. The authors frame this combination within a minimum Bayes risk framework. To address the computational cost of multiple generations, a more efficient variant, CoCoA Light, is proposed, which uses a lightweight predictor to estimate semantic inconsistency. The authors conduct experiments on various tasks, including question answering, summarization, and machine translation, showing that CoCoA outperforms existing UQ baselines in failure prediction.

# Strengths
* The paper is well structured and written.
* The paper provides a clear and well-grounded framework that unifies two important sources of uncertainty—model confidence and output consistency. The connection to the minimum Bayes risk framework provides a solid theoretical foundation.
* The experiments are comprehensive, covering multiple tasks, datasets, and models.
* CoCoA Light reduces the sampling cost by learning to approximate the consistency-based term from representations, making it more applicable for real-world applications.

# Weaknesses
* The core idea of combining confidence and consistency feels somewhat incremental. While the MBR framing is clean, many recent UQ methods already combine confidence and consistency (e.g., SAR, Semantic Entropy). The main novelty lies in formalizing their combination via MBR.

# Overall

Following the author-reviewer discussion period, no major weaknesses or concerns remain. The reviewers generally agree that the paper is well-written and supported by extensive experiments across numerous tasks, datasets, and models. In particular, they view CoCoA Light as an important and novel contribution to the research community.

A slight reservation was noted regarding the novelty compared to previous work. While combining confidence and consistency methods is not novel per se, framing it as a combination via MBR is new. Additionally, CoCoA Light is recognized as a novel and efficient contribution.